# Revealing Positive and Negative Role Models to Help People Make Good Decisions

## Abstract

We consider a setting where agents take action by following their role models in a social network, and study strategies for a social planner to help agents by revealing whether the role models are positive or negative. Specifically, agents observe a local neighborhood of possible role models they can emulate, but do not know their true labels. Revealing a positive label encourages emulation, while revealing a negative one redirects agents toward alternative options. The social planner observes all labels, but operates under a limited disclosure budget that it selectively allocates to maximize social welfare (the expected number of agents who emulate adjacent positive role models). We consider both algorithms and hardness results for welfare maximization, and provide a sample-complexity guarantee when the planner observes a sampled subset of agents. We also consider fairness guarantees when agents belong to different groups. It is a technical challenge that the ability to reveal negative role models breaks submodularity. We thus introduce a proxy welfare function that remains submodular even when revealed targets include negative ones. When each agent has at most a constant number of negative target neighbors, we use this proxy to achieve a constant-factor approximation to the true optimal welfare gain. When agents belong to different groups, we also show that each group's welfare gain is within a constant factor of the optimum achievable if the full budget were allocated to that group. Beyond this basic model, we also propose an intervention model that directly connects high-risk agents to positive role models, and a coverage radius model that expands the visibility of selected positive role models. Lastly, we conduct extensive experiments on four real-world datasets to support our theoretical results and assess the effectiveness of the proposed algorithms.

## 1 Introduction

Consider a government agency that wants to provide a booklet with information about how to correctly file your taxes. The booklet will not be able to cover every possible question that any taxpayer may have, but it should cover common tax situations that arise. For these situations, the booklet may include positive examples showing what to do and/or negative examples showing what not to do. For example, IRS Publication 970[1] gives a large number of examples showing what to report for various types of fellowships, scholarships, and grants, as well as warnings about common mistakes. Given a limited budget of how many examples can be included in the booklet, the government would like to include those that will be most helpful. In this case, it is natural to model a taxpayer facing a given scenario as following a positive example if a relevant one is present in the booklet, avoiding an incorrect action if it is described as incorrect in the booklet, and otherwise choosing randomly from among the reasonable options they have.

This work formalizes the setting as an unweighted bipartite graph with taxpayers (agents) on the left and strategies (targets) on the right. Edges connect taxpayers to strategies within their collection. Each strategy is classified as *positive* if it reflects desirable decision-making patterns and as *negative* if it reflects those that the agents should avoid. Initially, agents do not know which of their adjacent strategies are positive or negative. In this case, the booklet is the subset of targets whose labels are revealed by the social planner.

---

[1]"Tax Benefits for Education" https://www.irs.gov/pub/irs-pdf/p970.pdf

Our goal is to study how a tax-compliance body (i.e., a social planner) with a limited budget can help agents identify and emulate or imitate positive targets, thereby improving their decision quality, by revealing the labels of a limited number of targets. Revealing positive targets causes agents to emulate their behavior, while revealing negative ones indicates decisions that should be avoided, but does not suggest which choices are good. Consequently, in the standard model, the planner's objective is to reveal labels of a budgeted subset of targets to maximize *social welfare*, defined as the total probability that agents emulate adjacent positive targets, given the revealed subset. We extend the standard model to consider a setting where the planner identifies agents most likely to emulate negative targets and uses the intervention budget to connect them directly to positive ones, ensuring these agents emulate a positive target. In the tax filing setting, this could involve identifying taxpayers prone to bad strategies and pairing them with positive ones to follow.

**Contributions.** While motivated by a tax filing example, the proposed models are more broadly applicable to other domains in which agents rely on role models, options or exemplars in their neighborhood to make decisions. Appendix A provides additional examples. Below are our main contributions.

1. In the standard model, we show that when the social planner reveals negative targets, the social welfare function remains monotone, but may become supermodular (Section 2.1). Consequently, the approximation guarantees of the classic polynomial-time, budget-constrained greedy algorithm can deteriorate to as low as $\frac{2}{\sqrt{n}+2}$, where $n$ denotes the number of agents, when both positive and negative targets can be revealed. To address this limitation, we introduce a proxy welfare function that remains submodular even when revealed targets include negative ones. When all agents have at most $c$ negative target neighbors, this proxy achieves a constant-factor approximation to the true welfare (gain) (Section 3.2).

2. In a setting where agents are divided into $w$ groups (for constant $w$), we show that running the classic greedy algorithm on each group $a$ with budget $\lceil K/w \rceil$ guarantees that $c$-bounded agents in $a$ achieve a welfare gain of $\Omega(\text{OPT}_a^K)$, i.e., within a constant factor of the optimum achievable if the full budget $K$ were allocated to that group. When agents are not $c$-bounded, we show that this guarantee may fail to hold (Section 3.3).

3. To assess the potential for stronger algorithmic results and demonstrate that the existing guarantees are essentially tight, we establish NP-Hardness for the social welfare maximization problem when the planner can reveal only positive targets, only negative targets, or both (Section 3.4). Then, in Section 3.5, we study a learning-theoretic variant of the standard model in which the left-hand side of the bipartite graph is replaced by a probability distribution $D$ over agents. We establish a sample-complexity guarantee for the setting where the social planner observes the neighborhood of each agent sampled from $D$.

4. In cases where agents either have poor neighborhoods or are unaware of the adjacent positive targets, the standard model fails to help these agents make good decisions. To address these limitations, Section 4 introduces two intervention mechanisms: connecting poorly positioned agents to positive targets before or after revealing a set of atmost $K$ welfare-maximizing targets (Section 4.1), and increasing the visibility of positive targets so that neighboring agents can observe them (Section 4.2).

5. Finally, we conduct extensive semi-synthetic experiments using bipartite graphs generated from four real-world datasets: Adult, Student Performance (Mathematics and Portuguese), and Garment Workers Productivity. We empirically evaluate and compare several greedy strategies under the standard model, examine gains from targeted intervention, and assess the performance of Algorithm 1 in the learning setting[2] (Section 5).

## 1.1 Related Work

**Personalized recourse.** The growing reliance on machine learning (ML) models to make high-stakes decisions (e.g., for hiring and loan approvals) raises urgent questions about transparency and the provision

---

[2]Our code will be available after review

of guidance that enables individuals to improve their outcomes. This concern motivated extensive work on personalized recourse, typically operationalized through single-agent (Karimi et al., 2022; Verma et al., 2024) or multi-agent (Pedapati et al., 2020; Kanamori et al., 2022; Ley et al., 2023; Carrizosa et al., 2024; Naggita et al., 2025; Kavouras et al., 2026) frameworks. These approaches assume access to the agents' initial feature states and action spaces and identify the minimum cost set of actions that lead to a desirable prediction. In contrast, the social planner in our setting lacks such information and instead releases limited signals, namely whether adjacent targets are positive or negative influences, enabling agents to improve outcomes by emulating adjacent positive targets.

**Strategic learning.** Our line of inquiry is closely related to strategic learning under manipulation graphs (Zhang & Conitzer, 2021; Lechner & Urner, 2022; Ahmadi et al., 2022; 2024; Cohen et al., 2024a; Attias et al., 2025), where each agent's reaction set is shaped by its local neighborhood. The key distinction is that the social planner in our setting does not control the labeling function and is limited to revealing only a small subset of labels, a setup analogous to strategic learning with restricted label queries (Balcan & Beyhaghi, 2025). Our work is also related to research on strategic learning under imitative strategic behavior and partial information release, in which agents strategically modify their features by observing or imitating the strategies of social targets (Heidari et al., 2019; Raab & Liu, 2021; Zhang et al., 2022; Xie et al., 2024) or of historical feature-prediction pairs (Ghalme et al., 2021; Bechavod et al., 2022; Cohen et al., 2024b). Unlike these studies, we do not assume agents know which targets to emulate; rather, we study how selectively revealing labels for a subset of targets can guide agents toward better decisions. Lastly, our work is tangentially related to prior research that leverages counterfactual explanations to help agents respond optimally in strategic classification settings (Tsirtsis & Gomez-Rodriguez, 2020; Xie et al., 2025); however, unlike these works, ours is a multi-agent approach that steers agents toward positive targets to emulate rather than recommending feature-level actions for individual agents.

**Influence maximization.** Our objective is analogous to influence maximization, which aims to iteratively identify the most influential nodes to shape agents' behaviors (Richardson & Domingos, 2002; Kempe et al., 2003; Kamarthi et al., 2020). Traditional methods typically assume a monotone, submodular objective and apply greedy strategies to maximize influence across multi-step diffusion processes (Du et al., 2017; Li et al., 2018; 2023). The importance of submodularity extends beyond influence maximization; there is a rich literature on learning submodular functions from data (Balcan & Harvey, 2018; 2011). Although submodularity often enables efficient optimization and learning guarantees, recursive diffusion-based methods remain computationally demanding. By contrast, we study one-step models on bipartite graphs, where the objective may remain monotone but become supermodular. In this setting, greedy strategies can perform poorly, but our approach avoids the complexity of multi-step diffusion.

## 2 Problem Formulation

We model the setting as an unweighted bipartite graph $\mathcal{G} = (\mathcal{X} \cup \mathcal{T}, E)$ (Figure 1), where $\mathcal{X}$ is the set of $n$ agents (left-hand nodes), $\mathcal{T}$ is the set of $m$ targets (right-hand nodes), and $E$ contains edges between each agent and the targets it can emulate. Let $f : \mathcal{T} \to \{-1, +1\}$ be the true target labeling function only known to the social planner. For an agent $x \in \mathcal{X}$, let $N(x) = \{t \in \mathcal{T} : (x, t) \in E\}$ denote its neighborhood, and $\delta_x^+ = |\{t \in N(x) : f(t) = +1\}|$ and $\delta_x^- = |\{t \in N(x) : f(t) = -1\}|$ be the number of positive and negative neighbors, respectively. Of the $m = m^+ + m^-$ targets, $m^+$ are positive, representing desirable behaviors agents should emulate, and $m^-$ are negative, representing behaviors agents should avoid. We next describe how each agent chooses a target to emulate from their neighborhood and how the social planner selects a subset of targets whose information (i.e., labels) is revealed.

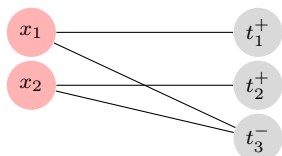

**Figure 1:** An unweighted bipartite graph in which the LHS nodes $\{x_1, x_2\}$ represent agents, the RHS nodes $\{t_1, t_2, t_3\}$ represent targets, and edges connect agents to targets they can emulate. Initially, agents do not know whether a target is positive or negative.

**Agents' choice of who to emulate.** Agents do not observe targets' labels and rely only on those revealed in the set $S \subseteq \mathcal{T}$. For an agent $x$, let $P_x^+(S) = \{t \in N(x) \cap S : f(t) = +1\}$ denote the adjacent targets revealed as positive, and $P_x^-(S) = \{t \in N(x) \cap S : f(t) = -1\}$ denote those revealed as negative. If no adjacent targets are revealed, meaning either $S = \emptyset$ or $N(x) \cap S = \emptyset$, the agent emulates a target selected uniformly at random from $N(x)$. Otherwise, the agent assigns zero probability to adjacent targets revealed as negative. If any adjacent targets are revealed as positive, it selects uniformly among them; if none are positive, it selects uniformly among the adjacent unlabeled targets. If all targets in $N(x)$ are revealed as negative, the probability that agent $x$ emulates a positive target is 0. Accordingly, we define the total probability mass that agent $x$ assigns to positively labeled target neighbors given the revealed set $S$ as follows:

$$Q^S(x) = \begin{cases} 1 & \text{if } |P_x^+(S)| > 0, \\ \dfrac{\delta_x^+}{\delta_x^+ + (\delta_x^- - |P_x^-(S)|)} & \text{if } |P_x^+(S)| = 0 \text{ and } N(x) > 0, \\ 0 & \text{otherwise.} \end{cases}$$

**The social planner information reveal.** Assume the social planner has full knowledge of the graph, including all agents and the finite set of targets, and observes each target's true label through the labeling function $f : \mathcal{T} \to \{+1, -1\}$. Further, the social planner is aware of the aforementioned process by which agents choose which target in their neighborhood to emulate. Given a target reveal budget $K \in \mathbb{N}$, the objective of the social planner is to reveal the labels of the subset of targets[3] $S \subseteq \mathcal{T}$ with $|S| \leq K$ that maximizes the probability that the agents choose to emulate positively labeled targets. That is, the social planner aims to find

$$S^\star = \underset{S : |S| \leq K}{\arg\max} \ F(S).$$

where the social welfare function $F$ is defined as:

$$F(S) = \sum_{x \in \mathcal{X}} Q^S(x) \tag{1}$$

The gain in social welfare from revealing $S$ is defined as the difference between the social welfare under $S$ and the social welfare under the empty set:

$$G(S) = F(S) - F(\emptyset) \tag{2}$$

The marginal gain of revealing a target $t \in \mathcal{T} \setminus S$ given a revealed set $S \subseteq \mathcal{T}$ is defined as:

$$\Delta_t(S) = F(S \cup \{t\}) - F(S) \tag{3}$$

### 2.1 Monotonicity and Submodularity of the Social Welfare Function

In this section, we first demonstrate that the social welfare function is a monotonically increasing function, and then explore the conditions under which it is submodular.

**Proposition 1.** *The social welfare function is a monotonically increasing function. That is, for any set of revealed targets $A \subseteq \mathcal{T}$, $F(A \cup \{t\}) \geq F(A)$ for all $t \in \mathcal{T} \setminus A$.*

Intuitively, revealing an additional target cannot reduce an agent's probability of selecting a positive target from its neighborhood. The proof is provided in Appendix B.1 for completeness.

We now analyze the submodularity of the social welfare function when the social planner can reveal only positive targets and when revealed targets include negative ones.

**Definition 2.1** (Submodularity). *A function $F : 2^\mathcal{T} \to \mathbb{R}_{\geq 0}$ is submodular if the marginal gain (Eqn. 3) of adding a revealed target $t$ to a smaller revealed target set $A \subseteq B$ is at least as large as the marginal gain of adding it to a larger revealed target set $B$. That is, for every $A, B \subseteq \mathcal{T}$ where $A \subseteq B$, and every $t \in \mathcal{T} \setminus B$, we have*

$$F(A \cup \{t\}) - F(A) \geq F(B \cup \{t\}) - F(B).$$

---

[3]When clear from context, "reveal a subset of targets" denotes "reveals the labels of a subset of targets"

Proposition 2 establishes that when the social planner is restricted to revealing only positive targets, the social welfare function $F(S)$ is monotone and submodular.

**Proposition 2.** *When the social planner is restricted to only revealing positive targets, then $F : 2^{\mathcal{T}^+} \to \mathbb{R}_{\geq 0}$ is submodular.*

The function $F$ is monotone (Proposition 1), and the marginal gain from revealing a positive target is at least as large when the current revealed target set is small as when it is large and more such targets are already known. Formal proof in Appendix B.2.

Proposition 3 shows that when the revealed targets include negative ones, $F$ remains monotone, but might not necessarily be submodular.

**Proposition 3.** *When the targets the social planner reveals include negative ones, then the social welfare function might not necessarily be submodular.*

*Proof sketch.* Consider an agent adjacent to at least two positive and two negative targets. Revealing an additional positive target results in 0 marginal gain, whereas revealing an additional negative target before any adjacent positive target is revealed results in increasing marginal gain because the probability of emulating an adjacent positive target increases with increase in the number of revealed adjacent negative targets. Full proof in Appendix B.3. $\qquad\square$

## 3 The Standard Model

We begin this section with a simple and intuitive greedy algorithm that selects up to $K$ targets to maximize social welfare (Section 3.1). We then examine how the target disclosure policy affects the algorithm's approximation guarantees. In particular, Section 3.2 shows that when disclosure is restricted to positive targets, submodularity is preserved, allowing the greedy algorithm to attain the classic $(1-1/e)$-approximation guarantee. By contrast, once negative targets can be revealed, submodularity may fail, causing the algorithm's performance to deteriorate arbitrarily. To address this challenge, we introduce a proxy welfare function that approximates the true social welfare (gain) while preserving submodularity under any disclosure policy. Section 3.2.3 shows that when each agent has at most $c$ negative-target neighbors, the proxy objective yields a constant-factor approximation to the optimal true welfare (gain). In settings where agents belong to multiple groups, Section 3.3 shows that the proxy also guarantees that each group's welfare is within a constant factor of the maximum welfare achievable if the entire budget $K$ were allocated to that group.

Finally, Section 3.4 shows that our algorithmic guarantees are essentially tight. That is, when disclosure is restricted to positive targets, no polynomial-time algorithm can achieve a substantially better worst-case approximation guarantee, and when both positive and negative targets may be disclosed, there is no approximation solution. We then conclude the analysis of the standard model by providing a learning-theoretic version of the model and establishing sample-complexity bounds for social welfare maximization when the planner has access only to a sampled subset of agents (Section 3.5).

---

**Algorithm 1** Greedy Target Reveal

---

1: **Input:** Graph $\mathcal{G} = (\mathcal{X} \cup \mathcal{T}, E)$, targets $\mathcal{T}'$, labels $\{f(t)\}_{t \in \mathcal{T}'}$, budget $K$, initial set $S' = \emptyset$
2: **Output:** Solution set $S_{\mathrm{g}} \subseteq \mathcal{T}'$ with $|S_{\mathrm{g}}| \leq K$, and social welfare $F(S_{\mathrm{g}})$
3: $S_{\mathrm{g}} \leftarrow S'$
4: **while** $|S_{\mathrm{g}}| \leq K$ **do**
5: $\quad t^\star \leftarrow \arg\max_{t \in \mathcal{T}' \setminus S_{\mathrm{g}}} \left( F(S_{\mathrm{g}} \cup \{t\}) - F(S_{\mathrm{g}}) \right)$
6: $\quad$ **if** $F(S_{\mathrm{g}} \cup \{t^\star\}) = F(S_{\mathrm{g}})$ **then**
7: $\quad\quad$ **break**
8: $\quad S_{\mathrm{g}} \leftarrow S_{\mathrm{g}} \cup \{t^\star\}$
9: **return** $(S_{\mathrm{g}},\ F(S_{\mathrm{g}}))$

---

### 3.1 The Greedy Algorithm

The main result of this section is a greedy algorithm that selects up to $K$ targets to maximize social welfare (Algorithm 1). Proposition 4 in Appendix C.1 analyzes Algorithm 1's complexity, and Appendix C contains the omitted proofs and additional greedy variants.

**Overview of Algorithm 1.** At each iteration, Algorithm 1 reveals the target $t^\star \in \mathcal{T} \setminus S_g$ that yields the highest marginal gain $(F(S_g \cup \{t^\star\}) - F(S_g))$. The process repeats until no unrevealed target yields a positive marginal gain or when the budget is exhausted.

Unless otherwise stated, the initial revealed target set is empty $S' = \emptyset$. Algorithm 1 is run with target set $\mathcal{T}' = \mathcal{T}^+ = \{t \in \mathcal{T} \mid f(t) = +1\}$ when target reveal is restricted to only positive targets, with $\mathcal{T}' = \mathcal{T}^- = \{t \in \mathcal{T} \mid f(t) = -1\}$ when restricted to negative targets, and with $\mathcal{T}' = \mathcal{T}$ when no restriction is imposed.

### 3.2 Approximation Guarantees

The composition of the optimal solution set depends on the graph and may include only positive targets (Figure 2a), only negative targets (Figure 2b), or both positive and negative targets (Figure 2c). Additionally, since the information disclosure policy affects the submodularity of the social welfare function (Section 2.1), in this section, we examine the approximation guarantees of the classic greedy algorithm (Algorithm 1) under three disclosure regimes: revealing only positive targets (Section 3.2.1), revealing only negative targets (Section 3.2.2), and unrestricted disclosure (Section 3.2.3).

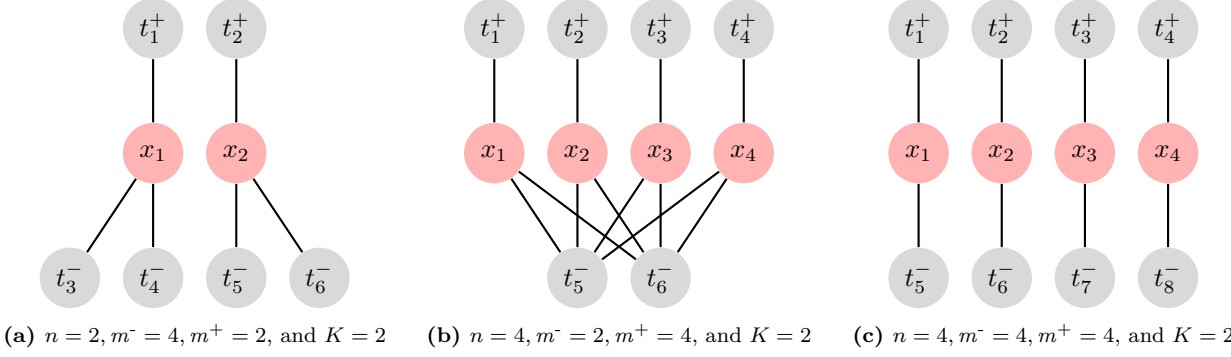

**(a)** $n = 2, m^- = 4, m^+ = 2,$ and $K = 2$    **(b)** $n = 4, m^- = 2, m^+ = 4,$ and $K = 2$    **(c)** $n = 4, m^- = 4, m^+ = 4,$ and $K = 2$

**Figure 2:** Even with the same budget, the composition of the optimal target set varies with graph structure.

#### 3.2.1 The social planner is restricted to revealing only positive targets

By Proposition 2, restricting the planner to positive targets makes $F$ monotone and submodular, implying that Algorithm 1 achieves a $(1 - 1/e)$-approximation (Theorem 1).

**Theorem 1.** *When the social planner can only reveal positive targets, Algorithm 1 achieves an $(1 - 1/e)$-approximation for the $\max\limits_{\substack{S:\, |S| \leq K \\ f(t)=1 \ \forall t \in S}} F(S)$ problem. That is, $F(S_g) \geq (1 - 1/e)\, F(S^\star)$, where $S_g$ is the greedy solution and $S^\star$ is the optimal solution.*

This guarantee follows from the classical result of Nemhauser et al. (1978).

Note that when the social planner is restricted to revealing only positive targets, the welfare gain (Eqn. 2) is monotone and submodular because $F(\emptyset)$ is a constant and $F(S)$ is monotone and submodular. Consequently, the approximation guarantee in Theorem 1 extends to this setting. That is, $G(S_g) \geq (1 - 1/e)\, G(S^\star)$ where $S_g$ is the greedy solution and $S^\star$ is the optimal solution.

### 3.2.2 The social planner can only reveal negative targets

By Proposition 3, restricting the social planner to revealing only negative targets preserves monotonicity but not submodularity of $F$. We construct an example where the approximation ratio of Algorithm 1 can be strictly below $3/\sqrt{2n}$, where $n$ is the number of agents (Theorem 6 in Appendix C.3).

### 3.2.3 The social planner can reveal both positive and negative targets

Assume the social planner may reveal both positive and negative targets. By Proposition 3, including negative targets in the revealed set preserves monotonicity but not the submodularity of true social welfare function $F$. We construct an example where the approximation ratio of Algorithm 1 can be strictly below $2/(\sqrt{n} + 2)$, where $n$ is the number of agents (Appendix Theorem 7).

Now assume each agent has at most $c$ negative target neighbors for some constant $c \geq 1$. That is, we assume $\delta_x^- \leq c$ for all $x \in \mathcal{X}$. We call such an agent $c$-bounded. Definition 3.1 introduces the proxy social welfare function, with proxy welfare gains defined in Definition 3.2.

**Definition 3.1** (Proxy social welfare function).

$$F_p(S) = \sum_{x \in \mathcal{X}} Q_p^S(x), \qquad \text{where} \tag{4}$$

$$Q_p^S(x) = \begin{cases} 1, & \text{if } |P_x^+(S)| > 0, \\ \dfrac{\delta_x^+}{|N(x)| - \mathbf{1}_{\{|P_x^-(S)| \geq 1\}}} \left(1 + \dfrac{\max\{0, |P_x^-(S)| - 1\}}{|N(x)|}\right) & \text{if } |P_x^+(S)| = 0 \text{ and } |N(x)| > 0, \\ 0, & \text{otherwise.} \end{cases}$$

*is the proxy total probability mass assigned by agent $x$ to positively labeled targets in its neighborhood given the revealed set $S$. In particular, if we reveal an agent's negative neighbor, we only increase $Q_p^S(x)$ by the amount that revealing the first negative neighbor helped.*

**Definition 3.2** (Proxy welfare gain). *The proxy social welfare gain from revealing $S$ is defined as the difference between the proxy welfare under $S$ and the welfare under the empty set:*

$$G_p(S) = F_p(S) - F(\emptyset) \tag{5}$$

By definition (Eqn. 1 and Defn. 3.2), the proxy welfare (gain) is less than or equal to the true welfare (gain). What we show is that if agents are $c$-bounded for some constant $c$, then in fact the proxy welfare (gain) is within a constant factor of the true welfare (gain). That is, $F_p(S) \leq F(S) \leq cF_p(S)$ and $G_p(S) \leq G(S) \leq cG_p(S)$ (Appendix Lemma 2). Moreover, when the social planner can reveal both positive and negative targets, the proxy social welfare function is submodular, and the proxy is submodular on the gain (Appendix Lemmas 3 and 4). Consequently, *proxy-greedy*, defined as running the classical greedy algorithm on the proxy welfare function rather than the true welfare, achieves a constant-factor approximation to the true social welfare gain (Theorem 2). Note that for any solution set, the proxy achieves $(1/c)$-factor approximation to the true optimal welfare (Appendix Remark 5).

**Theorem 2.** *When revealed targets may include negative ones, and all agents are $c$-bounded for some constant $c \geq 1$, the proxy-greedy algorithm achieves a constant-factor approximation to the true social welfare gain. That is, if $S_p$ denotes the set revealed by proxy-greedy and $S^\star$ an optimal set under the true social welfare function, then $G(S_p) \geq \frac{1-1/e}{c} G(S^\star)$.*

*Proof.* Proof in Appendix C.3.3 □

### 3.3 Simultaneous Approximate Optimality

In this section, we establish fairness guarantees by showing that there exists a solution set that is simultaneously approximately optimal for all groups, and also examine how the inclusion of negative targets in the

revealed target set affects this per-group approximation guarantee. That is, using Algorithm 1, along with the true and proxy social welfare functions and the $c$-boundedness property, we assess the fairness of the greedy algorithm in a grouped setting.

Suppose agents are divided into $w$ groups $A_1, \ldots, A_w$, and the social planner has a total reveal budget of $K$. For each group $a \in [w]$, define

$$\text{OPT}_a^k = \max_{\substack{S \subseteq \mathcal{T} \\ |S| \leq k}} \left( \sum_{x \in A_a} Q^S(x) - \sum_{x \in A_a} Q^\emptyset(x) \right), \qquad 1 \leq k \leq K. \tag{6}$$

as the maximum social welfare gain for group $a$ using a budget of $k$, assuming we focus solely on that group. Throughout this section, we assume that the number of groups $w$ is a constant and that the total target reveal budget is $K \geq w$. A revealed target set is *simultaneously approximately optimal for all groups* (i.e., satisfies all groups), if every group $a \in [w]$ receives social welfare gain proportional to $\text{OPT}_a^K$. This goal is natural because no group can achieve more than $O(\text{OPT}_a^K)$ social welfare, even if it were allocated the entire budget. We now characterize conditions under which this fairness guarantee can be met.

For arbitrary graphs, if the social planner can only reveal positive targets and allocates a budget of $K_a = \lceil K/w \rceil$ to each group, then all groups can be satisfied (Appendix Theorem 8). In contrast, when negative targets may be revealed, there exist graph structures where no solution satisfies all groups (Appendix Remark 8). On the other hand, when the revealed targets may include negative ones, but all agents are $c$-bounded, then running the proxy-greedy algorithm separately on each group with a budget of $K_a$ guarantees that each group $a$ is helped by $\Omega(\text{OPT}_a^K)$ (Appendix Corollary 9).

### 3.4 NP-Hardness Results

In this section, we show that when the social planner is restricted to positive targets and the social welfare function is monotone and submodular, hardness follows from a reduction from the max-$K$-cover problem. When the planner is restricted to only revealing negative targets, we prove hardness of the maximization problem via a reduction from the $K$-clique problem in a $\theta$-regular graph.

**Theorem 3.** *Given a set of agents $x_1, \ldots, x_n$ and a set of targets $\mathcal{T}$, when the social planner can only reveal positive targets $\mathcal{T}^+ = \{t \in \mathcal{T} \mid f(t) = +1\}$, the problem of finding $S_g \subseteq \mathcal{T}^+$ of size at most $K$ ($|S_g| \leq K$) that maximizes social welfare $F(S_g)$ is NP-hard. Also, unless $P = NP$, the problem of finding a positive target subset $S_g \subseteq \mathcal{T}^+$ of size $K$ that maximizes social welfare cannot be approximated within a factor better than $1 - 1/e$.*

*Proof sketch.* We prove NP-hardness by a polynomial-time reduction from max-$K$-cover. Given a universe $U = \{e_1, \ldots, e_n\}$ and sets $\mathcal{C} = \{C_1, \ldots, C_m\}$, we create an instance of our problem by creating one agent for each element and one positive target for each set, connecting an agent to the target if and only if it is in the set. To ensure all agents have the same initial welfare, each agent is also connected to a number of private negative targets equal to its positive degree, ensuring that before any positive target is revealed, every agent contributes $1/2$, so $F(\emptyset) = n/2$. Revealing a positive target raises the contribution of all adjacent agents from $1/2$ to $1$, implying that for any $S \subseteq \mathcal{T}^+$ with $|S| \leq K$, $F(S) = \frac{n}{2} + \frac{1}{2}\left|\bigcup_{t_j \in S} C_j\right|$. Hence, achieving welfare at least $W = \frac{n}{2} + \frac{\mathcal{E}}{2}$ is equivalent to covering at least $\mathcal{E}$ elements with at most $K$ sets, establishing NP-hardness. Moreover, since the welfare gain beyond the baseline is exactly proportional to the achieved coverage, any approximation for maximizing social welfare induces an approximation of the same factor for max-$K$-cover. Therefore, unless $P = NP$, no polynomial-time algorithm can approximate the problem within a factor better than $1 - 1/e$. Formal proof is included in Appendix C.5.1. $\qquad\square$

**Remark 1.** *If the social planner can reveal both positive and negative targets, the problem of revealing a subset $S_g \subseteq \mathcal{T}$ with $|S_g| \leq K$ that maximizes social welfare remains NP-hard. This follows directly from Theorem 3 where even though both positive and negative targets can be revealed, revealing negative targets results in less social welfare than revealing positive ones, so the social planner's optimal strategy reduces to the revealing only positive targets. Hence, permitting both types of targets does not change the NP-hardness of the problem.*

**Theorem 4.** *Given a graph $\mathcal{G} = (\mathcal{X} \cup \mathcal{T}, E)$ with $n = |\mathcal{X}|$ agents and target set $\mathcal{T}$, suppose the social planner can reveal only negative targets $\mathcal{T}^- = \{t \in \mathcal{T} \mid f(t) = -1\}$. Then finding a subset $S_g \subseteq \mathcal{T}^-$ with $|S_g| \leq K$ that maximizes social welfare is NP-hard.*

*Proof sketch.* We prove NP-hardness by a polynomial-time reduction from the $K$-clique problem in a $\theta$-regular graph. Given an instance $\mathcal{G}_c = (V, E_c)$, we create an instance of our problem by creating by constructing a bipartite instance with one negative target $t_v^-$ per vertex $v \in V$, one positive target $t_{uv}^+$ per edge $\{u, v\} \in E_c$, and one agent $x_{uv}$ per edge, with neighborhood $N(x_{uv}) = \{t_u^-, t_v^-, t_{uv}^+\}$. With no revealed targets, each agent contributes $1/3$ to welfare. Revealing a single adjacent negative target increases that agent's contribution by $1/6$, while revealing both adjacent negatives increases it by $2/3$. Setting the budget to $K$ and the threshold $W$ to $\frac{n}{3} + \frac{K\theta}{6} + \frac{1}{3}\binom{K}{2}$, a set $S \subseteq \mathcal{T}^-$ of size $K$ achieves social welfare of at least $W$ if and only if the corresponding vertices form a $K$-clique since non-clique pairs fail to realize the $\binom{K}{2}$ agents that gain the full $2/3$ increase in probability for emulating a positive target. Thus, deciding whether such a set $S$ exists is NP-hard, and because a candidate solution can be verified in polynomial time, the decision problem is NP-complete, which implies NP-hardness of the optimization problem. Formal proof is included in Appendix C.5.2. $\qquad\square$

## 3.5 Learning Setting

Consider a setting where the left-hand side of the bipartite graph $\mathcal{G} = (\mathcal{X} \cup \mathcal{T}, E)$ is replaced by a probability distribution $D$ over agents. The social planner draws agents i.i.d. from $D$ and for each agent, the planner observes its neighborhood (adjacent targets) and the probability of emulating a positive target. The planner's goal is to reveal a subset of targets $S \subseteq \mathcal{T}$ whose social welfare deviates from the true value (Eq. 7) by at most $\varepsilon$.

$$F(S) = \mathbb{E}_{x \sim D}\left[Q^S(x)\right] \tag{7}$$

Given a budget $K$ and a sample graph $\mathcal{G} = (\mathcal{X} \cup \mathcal{T}, E)$ with $\mathcal{X}$ agents drawn i.i.d. from $D$, the social planner runs Algorithm 1 on this train graph $\mathcal{G}$ and returns the revealed target set $S_g \subseteq \mathcal{T}$ with $|S_g| \leq K$ as its hypothesis. For a new agent $x_i$, $Q^{S_g}(x_i)$ estimates the agent's probability of emulating a positive target, and the performance of the hypothesis is measured as social welfare per agent. Theorem 5 gives a sufficient sample size to ensure, with high probability, that the welfare returned by the hypothesis is within $\varepsilon$ of the true value.

**Theorem 5.** *Let $S_g \subseteq \mathcal{T}$ be the target set revealed by Algorithm 1 on graph $\mathcal{G} = (\mathcal{X} \cup \mathcal{T}, E)$, where $\mathcal{X}$ is a set of agents sampled independently from $\mathcal{D}$, $|\mathcal{T}| = m$, and the target reveal budget is $K$. There exists a universal constant $C > 0$, such that for any $\varepsilon > 0, \delta \leq 1$, if*

$$|\mathcal{X}| \geq C((\varepsilon^2)^{-1}(K \log m + \log(1/\delta))),$$

*then with probability at least $1 - \delta$ the social welfare of the revealed target set $S_g$ differs from its true value by at most $\varepsilon$.*

*Proof.* See Appendix F $\qquad\square$

# 4 Extensions of the Standard Model

The standard model (Section 3) operates under the assumption that agents can observe neighboring targets but cannot distinguish positive targets from negative ones. While this captures many relevant settings, it overlooks an important practical constraint: even when a planner reveals a welfare-maximizing set of targets, some agents may still have a low probability of emulating a positive target because their local neighborhoods contain (zero)few positive targets or because no positive targets are directly observable to them. To address this limitation, we extend the standard model to allow budget-constrained interventions that go beyond the passive disclosure of target information. First, the targeted intervention models (Section 4.1) enable the planner to directly connect agents prone to emulating negative targets with positive targets. Second, the coverage radius model (Section 4.2) increases the visibility of positive targets, expanding the set of agents who can observe and potentially emulate them.

### 4.1 The Targeted Interventions Model

The targeted intervention model focuses on settings in which the social planner identifies high-risk agents, namely those most likely to emulate a negative target, and directly connects them to positive targets. For example, suppose that the school counselor knows that some students don't have successful STEM mentors, and the counselor intervenes by directly assigning a STEM mentor to some of these students.

We study two approaches of targeted intervention modeling: one applied before running the standard greedy algorithm with budget $K$ (pre-reveal), and one applied after it (post-reveal). Below, we provide an overview of both the pre- and post-reveal targeted intervention approaches. For completeness, Appendix D provides the full algorithmic details of both approaches, and Appendix G.4 empirically compares their intervention gains while examining how performance varies with the intervention and target-reveal budgets $(B, K)$.

**Overview of the pre- and post-reveal intervention algorithms.** Let $B$ denote the intervention budget, and let $S_o$ be the set of targets revealed either before or after executing the classical greedy algorithm (Algorithm 1) with a $K$ target reveal budget. Let $\mathcal{X}_{\mathrm{hr}} \subseteq \mathcal{X}$ denote the set of at most $B$ high-risk agents selected for intervention by directly connecting them to a positive target.

Intervening on agent $x \in \mathcal{X}_{\mathrm{hr}}$ raises its social welfare from $Q^{S_o}(x)$ to 1. If $Q^{S_o}(x) = 1$, then the intervention was redundant. The closer $Q^{S_o}(x)$ is to 0, the larger the intervention gain (i.e., the difference between social welfare from pre- or post-reveal intervention and Algorithm 1). In *pre-reveal intervention* (Appendix Algorithm 6), total social welfare equals the welfare from applying Algorithm 1 to the updated graph after removing the intervened-on agents and their edges, plus the welfare from intervening on the high-risk agents $Q' \leq B$. In *post-reveal intervention* (Appendix Algorithm 7), total social welfare equals the welfare returned by Algorithm 1 on the full graph, plus the welfare from intervening on high-risk agents $Q' = \sum_{x \in \mathcal{X}_{\mathrm{hr}}} \left(1 - Q^{S_o}(x)\right)$.

### 4.2 The Coverage Radius Model

In this section, we study a setting in which some agents are unaware of nearby positive targets, and a social planner intervenes by increasing their visibility so that agents can observe and learn from them. Consider a school counselor who knows which alumni have followed successful career paths. Some of the students the counselor advises can potentially learn from many of these alumni, but they are unaware of most of them. Therefore, the counselor intervenes by highlighting a subset of successful alumni, for example, through newsletters or targeted emails, thus increasing the students' awareness of these role models.

Formally, consider a geometric bipartite graph $\mathcal{G}^+ = (\mathcal{X} \cup \mathcal{T}^+, E)$ with a set of $d$-featured agents $\mathcal{X} = \{x_1, \ldots, x_n\} \subset \mathbb{R}^d$ on the left-hand side and positive targets $\mathcal{T}^+ = \{t_1, \ldots, t_m\} \subset \mathbb{R}^d$ on the right. Each target is labeled positive, and an (unobserved) edge exists between an agent and a target if their Euclidean distance is at most $r$.

To make adjacent positive targets visible to agents so the agents can emulate them, the social planner could either expand agents' visibility or increase the reach of targets. Since the former is trivial, the coverage radius model focuses on interventions from the targets' perspective.

Each target $t_i$ is assigned a radius $r_i \geq 0$, and an agent $x_j$ is reached if $\|t_i - x_j\|_2 \leq r_i$ for some $i \in [m]$. Initially, $r_i = 0$ for all $i \in [m]$, and a total radius budget $R$ constrains the intervention. The objective of the social planner is to maximize the number of agents reached:

$$\max_{r_1, \ldots, r_m \geq 0} \quad \sum_{j=1}^{n} \mathbf{1}(\exists i \in [m] : \|t_i - x_j\|_2 \leq r_i)$$
$$\text{s.t.} \quad \sum_{i=1}^{m} r_i \leq R. \tag{8}$$

Algorithm 8 in Appendix E presents a greedy approach to this problem, and Appendix G.5 demonstrates its effectiveness on real-world datasets.

## 5 Experiments

We conduct extensive experiments to evaluate the performance of greedy strategies in practical settings under the standard model without information disclosure restrictions, and to assess the proposed algorithms under alternative model settings using semi-synthetic geometric bipartite graphs generated from the Adult, Student Performance (Mathematics and Portuguese), and Garment Workers Productivity datasets. Details on the datasets and preprocessing procedures are provided in Appendix G.1.1.

**Generation of geometric bipartite graphs.** We generate the geometric bipartite graph from two feature sets ($\mathcal{X}\mathrm{LHS} \in \mathbb{R}^{n \times \rho}$ and $\mathcal{X}\mathrm{RHS} \in \mathbb{R}^{m^\star \times \rho}$) extracted from a given dataset, where $\rho$ denotes the number of features, $n$ the number of agents, and $m^\star$ the number of *all* targets. First, we compute the pairwise distances between the agents ($\mathcal{X}_{\mathrm{LHS}}$) and the targets ($\mathcal{X}_{\mathrm{RHS}}$). That is, $D_{ij} = \|x_i - x_j\|_2, \quad i \in [n], \ j \in [m^\star]$. For each agent $i$, its neighborhood $\mathcal{N}(i)$ is defined either by the $k$NN method, where a target $j \in \mathcal{N}(i)$ iff it is among the $k_{\max} \geq 1$ closest targets to $i$ according to $D_{ij}$, or by a distance threshold method, where target $j \in \mathcal{N}(i)$ iff $D_{ij} \leq \ell$. The edge set is then given by $E = \{(i,j) : j \in \mathcal{N}(i)\}$. Now, together with the *used* targets $\mathcal{T} = \bigcup_{i=1}^{n} \mathcal{N}(i) \in \mathbb{R}^{m \times \rho}$ and their labels $f(j) = y_{\mathrm{RHS}}[j]$ for $j \in \mathcal{T}$, the bipartite graph is given by $\mathcal{G} = (\mathcal{X}_{\mathrm{LHS}} \cup \mathcal{T}, E)$. See Appendix G.1.2 for more details on the generated graphs.

**Algorithms, parameters, and evaluation metrics.** In the single-group standard model setting, we compare social welfare without budget constraints $F(S_{\mathrm{full}})$ and with zero budget $F(S_{\mathrm{o}})$ to budgeted strategies: random selection, classic greedy, heuristic greedy, and bruteforce search. In the grouped setting, we compare average group social welfare (gain) (total group welfare (gain) divided by group size) achieved by the Algorithm 1 when applied to (i) the full graph and (ii) male and female bipartite subgraphs constructed from the Adult, Math, and Portuguese datasets.

Under the targeted intervention model, for varying target-reveal ($K$) and intervention ($B$) budgets, we evaluate the intervention gains achieved by the pre- and post-reveal intervention. These gains are respectively defined as $\Delta_F(\mathrm{ig}, \mathrm{g}) = F(S_{\mathrm{ig}}) - F(S_g)$, and $\Delta_F(\mathrm{gi}, \mathrm{g}) = F(S_{\mathrm{gi}}) - F(S_g)$, where $F(S_g)$, $F(S_{\mathrm{ig}})$, and $F(S_{\mathrm{gi}})$ denote the welfare returned by Algorithms 1, 6, and 7, respectively.

In the learning setting, we report training (tr) and testing (ts) performance averaged over 100 independent train-test splits with different random seeds. Here, we report the tr and ts performance results using two metrics: $\mathrm{Perf}_2$, where 100 denotes success on all helpable agents (those with both positive and negative target neighbors), including all sampled agents; and $\mathrm{Perf}_3$, where 100 denotes success on all helpable agents, excluding unhelpable ones. Full details on algorithms, parameters, and evaluation metrics used are included in Appendices G.1.3 and G.1.4.

Empirical results for various settings are reported below and in Appendices G.2–G.6.

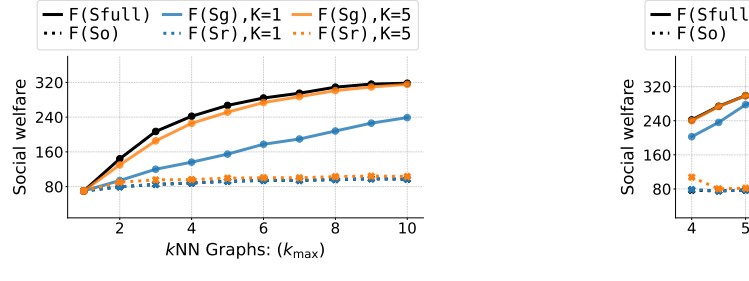

**(a)** The $k$NN generated graphs: $F(S_{\mathrm{g}})$ vs. $F(S_{\mathrm{r}})$  **(b)** The threshold generated graphs: $F(S_{\mathrm{g}})$ vs. $F(S_{\mathrm{r}})$

**Figure 3:** Comparative social welfare returned by random $F(S_{\mathrm{r}})$ and classical greedy $F(S_{\mathrm{g}})$ algorithms on **Adult** dataset for $K = \{1, 5\}$. Black lines mark the maximum $F(S_{\mathrm{full}})$ and minimum $F(S_{\mathrm{o}})$ welfare. Classic greedy consistently outperforms random when executed at the same target reveal budget $K$.

## 5.1 Empirical Results under the Standard Model

**One group setting.** The performance of the budgeted strategies heavily depends on the network structure. When connectivity is low, overall social welfare remains very small, regardless of the algorithm or budget used (see Appendix Tables 3 and 4 where $\ell \leq 5.0$, and corresponding results in Figures 9 and 10, subfigures (**f,g**), when $\ell \leq 5.0$). As connectivity increases, particularly in threshold-generated graphs (Appendix Tables 2–5), the social welfare achieved by classic greedy often matches the maximum achievable ($F(S_{\text{full}})$) (Figure 3b; Appendix Figures 9–11, subfigures **e–h**), because more positive targets are connected to nearly all helpable agents. Additionally, when executed at the same $K$, Algorithm 1 consistently outperforms random (Figure 3a; Appendix Figures 9a–d). Even with high budgets and connectivity, random selection can yield comparably very low social welfare (cf. Figure 3). These results and those in Appendix G.2 suggest that although greedy may have weaker theoretical guarantees without information disclosure constraints, it performs well in practice, likely because the graphs are typically well-connected and balanced.

**Fairness in a grouped setting.** With low graph connectivity and before revealing any targets (i.e., $K = 0$), the female group generally has lower average social welfare than the male group (Appendix Figures 12a–c and 13a–c). As connectivity and the budget increase, the average group social welfare (gain) increases and is closely similar across groups (Appendix Figures 12d–f and 13d–f), both in the case where the greedy is run exclusively on a specific group at $K/2$ (Appendix Figure 13) and when it's run on the whole graph at a budget of $K$ (Appendix Figure 12). See Appendix G.3 for more empirical results on fairness under the standard model.

## 5.2 Empirical Results under the Targeted Interventions Model

Overall, intervention gains are upper-bounded by the intervention budget $B$ and are larger with a smaller target reveal budget $K$ (Figure 4; Appendix Figures 14 and 15) and in graphs where many agents lack positive neighbors (e.g., Appendix Figures 14a and 14c). When classic greedy is already optimal and most agents have positive neighbors, intervention becomes redundant or underutilized, leading to little or no intervention gains (Appendix Figures 14 and 15 (subfigures (**b,d,f,h**))). Post-reveal interventions always yield positive intervention gains that are also usually at least as large as those from pre-reveal interventions (Figure 4; Appendix Figures 14 and 15 (subfigures (**a,c,e,g**))). As shown in Figure 4, pre-reveal intervention can sometimes yield negative intervention gains because removing high-risk agents and their edges early may distort the graph, causing Algorithm 1 to reveal a target set with lower social welfare than it would have otherwise, especially when high-risk agents already had high probabilities of positive emulation. See Appendix G.4 for more empirical results under targeted intervention.

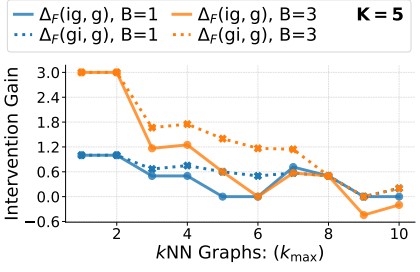

**Figure 4:** Comparison of pre-reveal ($\Delta_F(ig, g)$) and post-reveal ($\Delta_F(gi, g)$)) intervention gains, for $K = 5$ with $B = \{1, 3\}$ across $k$NN graphs on the **Productivity** dataset. Gains increase with decrease in $K$, and pre-reveal gains may be negative.

## 5.3 Empirical Results under the Learning Setting

The evaluation of Algorithm 1 over 100 randomized train-test splits shows that the average training performance generally matches or slightly exceeds testing performance across all datasets and metrics (Figure 5 and Appendix Figures 17 and 18). As graph connectivity increases, especially in threshold-generated graphs, training and testing performances converge, and the impact of a higher budget diminishes (Appendix Figures 17 and 18 (subfigures (**d–f, j–l**))). In contrast, lower connectivity, particularly in $k$NN graphs, amplifies budget effects, with higher budgets consistently producing better or equal train/test performance (Figure 5 and Appendix Figures 17 and 18 (subfigures (**a–c, g–i**))). Lastly, train/test performance depends on graph

structure, neighbor positivity, and metric; e.g., when $|N(x)| = 1$ for all $x \in \mathcal{X}$, $\text{Perf}_2$ equals the fraction of agents connected to positive targets (Figure 5a), and $\text{Perf}_3 = 0$ reflects number of helpable agents (Figure 5b). See Appendix G.6 for more empirical learning results.

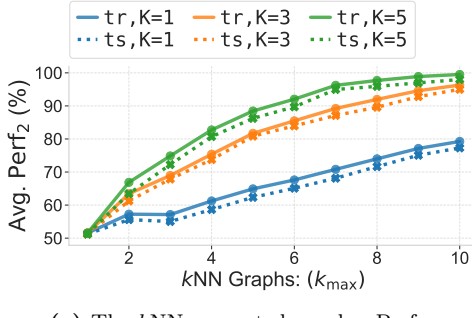

**(a)** The $k$NN generated graphs: $\text{Perf}_2$

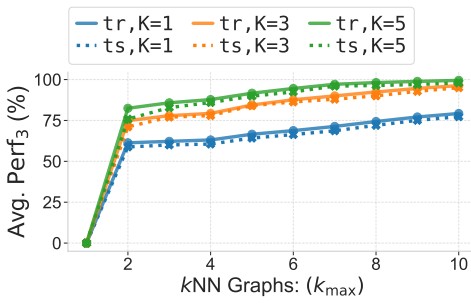

**(b)** The $k$NN generated graphs: $\text{Perf}_3$

**Figure 5:** Analysis of the training (tr) and testing (ts) performance $(\text{Perf}_2, \text{Perf}_3)$ scores when Algorithm 1 is run under budget $K = \{1, 5\}$ on $k$NN graphs (see Appendix Table 3) from the **Math** dataset. When each agent in train/test sets has at most one neighbor, $\text{Perf}_3$ is zero (b). Both $\text{Perf}_2$ and $\text{Perf}_3$ increase with $K$.

## 6   Conclusion

We propose various greedy strategies to help agents make good decisions when they observe targets within their social circles but lack information about the targets' labels. Although theoretical performance guarantees may weaken once negative targets can be in the revealed set because the true social welfare function may become supermodular, empirical evidence suggests that the classic greedy algorithm would likely perform well in practice, as graphs are more likely to be well-connected and balanced. To preserve submodularity, we introduce a proxy welfare function that achieves a constant-factor approximation to the true optimal welfare (gains) when agents are $c$-bounded. When agents are divided into groups, the proxy ensures that each group's welfare gain is within a constant factor of the optimum under full budget allocation. We also study interventions in which a social planner either directly connects high-risk agents to positive targets or increases the visibility of selected targets when agents are otherwise unaware of them. Future work could incorporate weighted edges to model heterogeneous emulation probabilities, extend our modeling setup to strategic classification by treating emulation of positive targets as improvement and negative targets as gaming, and generalize the deterministic graph to a stochastic setting (e.g., bipartite stochastic block models) where targets are positive with probability $p$ and edges from agents to positive and negative targets form with probabilities $q^+$ and $q^-$, respectively. See Appendix H for additional discussion.

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

# A    Additional Motivating Scenarios

**Training videos.**    Consider a company responsible for creating workplace training videos for client firms with the aim of promoting appropriate professional conduct, such as handling sensitive or confidential information.[4] In this setting, the video producer acts as the social planner, and the agents are the employees who watch the training videos and subsequently make decisions in workplace situations. For simplicity, assume that the social planner distributes the same standardized training module to all agents, even though their roles and day-to-day environments may differ slightly. For example, a company's level 4 and 5 employees might receive the same training video on workplace professional conduct. Employees operate in diverse roles and environments, and rely on exemplars such as colleagues when deciding how to act in various unfamiliar workplace situations. Because the social planner can produce only a limited number of dramatized scenarios, they must select those that improve decision-making across a large and heterogeneous workforce. Positive scenarios, such as reporting a suspicious email to technical support, demonstrate desirable conduct. Negative scenarios, such as the consequences of sharing confidential information with a fraudulent sender, discourage similar agents from making similar mistakes. When employees encounter a situation that closely matches one portrayed positively in the training, they follow the demonstrated action. When a similar situation is portrayed negatively, they avoid the depicted choice and instead look to alternative exemplars for guidance. If employees encounter a situation not covered in the training, they select a local exemplar uniformly at random, for instance by imitating how a colleague handled a comparable circumstance.

**Misinformation.**    Consider a setting in which each social media user has a fixed set of news outlets they follow, but they don't know which are legitimate ("good") and which are fake ("bad"). The social media manager seeks to reduce misinformation by increasing the expected number of users who ultimately rely on legitimate outlets, subject to a limited budget for verification labels. To achieve this, the manager assigns visible markers to a selected subset of outlets, indicating whether they are legitimate or fake. If a user observes that at least one of the outlets they follow is labeled as legitimate, they will choose to rely on it for news. However, if the only labeled outlet(s) in their collection are marked as fake, the user remains uncertain about the remaining outlets and chooses one at random from the remaining ones to rely on.

**High school career day.**    Consider a high school that runs a career day program to help senior students make informed decisions about their future careers. In practice, however, students often rely on role models from their own social circles (e.g., family members or community figures) without knowing whether emulating them will lead to positive or negative outcomes. The school aims to steer students toward desirable careers, but can only feature a limited number of role models, which makes the choice of whom to highlight especially important. Featured speakers might include positive examples, such as a physician who motivates students to pursue a career in medicine, as well as cautionary ones, like a former gang member whose story illustrates the long-term negative consequences of criminal involvement. After the event, students emulate a role model from their social circle: avoiding those identified as negative, following a role model identified as positive if any were revealed, and otherwise choosing randomly.

# B    Missing Proofs for Section 2

## B.1    Proof of Proposition 1

*Proof of Proposition 1.* Fix an initial revealed target set $A \subseteq \mathcal{T}$ and let $t \in \mathcal{T} \setminus A$. Compare the social welfare under $A$ with that under $A \cup \{t\}$. Any agent that already selects a positive target with probability 1 under $A$ continues to do so when $t$ is revealed. Agents for whom $t$ provides the first revealed positive target neighbor gain probability of 1 for choosing a positive target to emulate, and no agent's probability decreases.

Therefore, the total probability of choosing a target node under $A \cup \{t\}$ is always at least as large as the total probability under $A$ in every realization of the random process. The same conclusion holds in expectation $F(A \cup \{t\}) \geq F(A)$. This proves that the social welfare function is monotonically increasing since expanding the revealed target set never lowers the probability that agents select positive targets to emulate.    $\square$

---

[4]For example, Vector Solutions: https://www.vectorsolutions.com

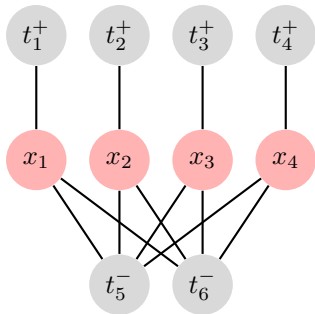

**Figure 6:** A bipartite graph where $n = 4$, $m^- = 2$ and $m^+ = 4$.

## B.2 Proof of Proposition 2

*Proof of Proposition 2.* Let $F(A)$ denote the social welfare generated by the revealed positive target set $A$, and let $A \subseteq B \subseteq \mathcal{T}^+$ where $\mathcal{T}^+ = \{t \in \mathcal{T} \mid f(t) = +1\}$. Submodularity requires that for any $A, B \subseteq \mathcal{T}^+$ and any $t \in \mathcal{T}^+ \setminus B$,

$$F(A \cup \{t\}) - F(A) \geq F(B \cup \{t\}) - F(B).$$

By Proposition 1, $F$ is monotone: $A \subseteq B$ implies $F(A) \leq F(B)$. Revealing a positive target $t$ sets each adjacent agent's probability of emulating a positive target to 1, unless it is already equal to 1 under the current revealed set. For any set $S \subseteq \mathcal{T}^+$, let $\gamma(S)$ be the set of agents whose probability of emulating a positive target equals 1 under $S$. Monotonicity implies that $\gamma(A) \subseteq \gamma(B)$.

Adding a revealed positive target $t$ to the revealed positive target sets $A$ and $B$ yields

$$F(A \cup \{t\}) - F(A) = |\gamma(A \cup \{t\})| - |\gamma(A)| = |\gamma(t) \setminus \gamma(A)|.$$

$$F(B \cup \{t\}) - F(B) = |\gamma(B \cup \{t\})| - |\gamma(B)| = |\gamma(t) \setminus \gamma(B)|.$$

Since $\gamma(A) \subseteq \gamma(B)$,

$$\gamma(t) \setminus \gamma(A) \supseteq \gamma(t) \setminus \gamma(B),$$

because any agent that already had a probability of 1 for emulating a positive target under the larger revealed set $B$ is excluded on the right-hand side. That is, adding a revealed positive target $t$ to a large set $B$ yields lower marginal gain because most agents are already committed to positive targets in $B$, so $t$ might be redundant. Adding $t$ to a small set $A$ instead makes it more likely that additional agents now have a probability of 1 for emulating a positive target. Therefore

$$|\gamma(t) \setminus \gamma(A)| \geq |\gamma(t) \setminus \gamma(B)|.$$

Thus, revealing a positive target $t$ when the revealed positive target set is smaller affects weakly more agents than when the known set is larger. The marginal contribution of revealing a new positive target $t$ declines as the revealed positive target set grows, which establishes submodularity: $F(A \cup \{t\}) - F(A) \geq F(B \cup \{t\}) - F(B)$. □

## B.3 Proof of Proposition 3

Example 1 shows that the social welfare function can become supermodular when the social planner is restricted to revealing negative targets. Furthermore, given the bipartite graph in Figure 7, supermodularity can arise even without this disclosure restriction when the solution set includes negative targets.

**Example 1** (Proof of Proposition 3)**.** *Consider the bipartite graph in Figure 6, with $n = 4$ agents, $m^+ = 4$ positive targets, and $m^- = 2$ negative targets. Each agent is adjacent to a unique positive target and to both the negative targets. Example 1 illustrates that when the social planner is restricted to revealing negative*

*targets in this setting, the social welfare function is not submodular. In particular, it shows that there exists $A, B \subseteq \mathcal{T}^-$ with $\mathcal{T}^- = \{t \in \mathcal{T} \mid f(t) = -1\}$ and $A \subseteq B$, and for some $t \in \mathcal{T}^- \setminus B$ such that*

$$F(A \cup t) - F(A) < F(B \cup t) - F(B).$$

*Given bipartite graph in Figure 6 and restriction to only revealing negative targets. Let the initially revealed negative target set be $A = \emptyset$, resulting in a social welfare of $F(A) = 4/3$. Next, let revealed negative target set $B = \{t_5^-\}$, and corresponding resulting social welfare $F(B) = 2$. Now consider revealing another negative target $t_6^- \in \mathcal{T}^- \setminus B$. The marginal gains become*

$$F(A \cup \{t_6^-\}) - F(A) = 2 - \tfrac{4}{3} = 0.6667, \qquad F(B \cup \{t_6^-\}) - F(B) = 4 - 2 = 2.$$

*Thus the marginal gain is larger when starting from a larger revealed negative target set $B$ than from a smaller one $A$, contradicting the diminishing returns condition required for submodularity.*

## C  Supplementary Material for Section 3

### C.1  Proof of Proposition 4

**Proposition 4.** *Algorithm 1 runs in $O(Kmn\delta)$ time.*

*Proof.* At each iteration, to determine whether to add a target $t \in \mathcal{T}$ to revealed target set, Algorithm 1 computes the resultant marginal gain, which first, involves computing $F(S_g \cup \{t\})$, a sum over $Q^{S_g \cup \{t\}}(x)$ for all $n$ agents $x \in \mathcal{X}$. If each agent has degree of atmost $\delta = |N(x)|$, then the time complexity of computing the social welfare $F(S_g \cup \{t\})$ is $O(n\delta)$. Computing the marginal gain given the previous and the new social welfare is $O(1)$. Repeating this process for $m$ targets across atmost $K$ iterations results in a total time complexity of $O(Kmn\delta)$. $\square$

### C.2  Bruteforce Algorithm

---
**Algorithm 2** Bruteforce Target Reveal

1: **Input:** Bipartite graph $\mathcal{G} = (\mathcal{X} \cup \mathcal{T}, E)$, labels $\{f(t)\}_{t \in \mathcal{T}}$, budget $K$
2: **Output:** Optimal solution set $S^\star \subseteq \mathcal{T}$ with $|S^\star| \leq K$ and social welfare $F(S^\star)$
3: Initialize $S^\star \leftarrow \emptyset$
4: **for all** $S \subseteq \mathcal{T}$ with $|S| \leq K$ **do**
5:     **if** $\big(F(S) > F(S^\star)\big)$ or $\big(F(S) = F(S^\star)$ and $|S| < |S^\star|\big)$ **then**
6:         $S^\star \leftarrow S$
7:         $F(S^\star) \leftarrow F(S)$
8: **return** $(S^\star, F(S^\star))$

---

**Proposition 5.** *The bruteforce algorithm (Algorithm 2) runs in $O(m^K n\delta)$ time.*

*Proof.* Algorithm 2 enumerates all $m^K$ candidate subsets of possible target reveals and selects the one that yields the highest social welfare. Evaluating social welfare for a single subset takes $O(n\delta)$ time, so the overall running time of Algorithm 2 is $O(m^K n\delta)$. $\square$

### C.3  Proofs for Section 3.2

#### C.3.1  Proof of Theorem 6

**Theorem 6.** *When the social planner is restricted to only revealing negative targets, there exists a graph and a budget $K$ for which Algorithm 1 attains an approximation ratio strictly less than $\frac{3}{\sqrt{2n}}$.*

*Proof.* The proof is in Example 2 below. $\square$

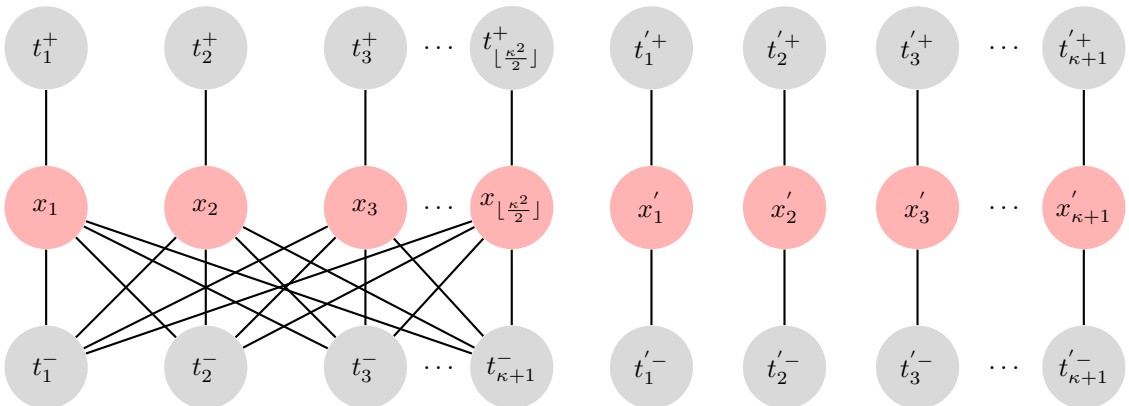

**Figure 7:** A bipartite graph example with statistics: $n = \lfloor \frac{\kappa^2}{2} \rfloor + \kappa + 1, m^- = 2\kappa + 2$, and $m^+ = n$. Algorithm 1 achieves an approximation ratio of $\frac{3}{\sqrt{2n}}$

**Example 2.** *Let $\kappa \in \mathbb{N}$ with $\kappa > 3$, and set the reveal budget to $K := \kappa + 1$. Consider the bipartite graph shown in Figure 7, with $n = \lfloor \frac{\kappa^2}{2} \rfloor + \kappa + 1$ agents, $m^- = 2\kappa + 2$ negative targets, and $m^+ = n$ positive targets. There are two types of agents, referred to as group 1 and group 2. Group 1 consists of $\lfloor \frac{\kappa^2}{2} \rfloor$ agents, each denoted $x_\star$. Each $x_\star$ is connected to a distinct positive target $t_\star^+$ and to $\kappa + 1$ of the negative targets $t_\star^-$. Group 2 consists of $\kappa + 1$ agents, each denoted $x_\star'$, and each is connected to a unique pair consisting of one positive and one negative target $(t_\star'^+, t_\star'^-)$.*

*Initially, the social welfare is $F(\emptyset) = \frac{\lfloor \frac{\kappa^2}{2} \rfloor}{\kappa + 2} + \frac{\kappa + 1}{2}$. At first iteration, we analyze which negative target the greedy algorithm picks: (1) If any one of the negative targets connected to group 1 agents (i.e, any of $t_*^-$), then it achieves a social welfare of $\frac{\lfloor \frac{\kappa^2}{2} \rfloor}{\kappa + 1} + \frac{\kappa + 1}{2}$; (2) If any one of the negative targets connected to group 2 agents (i.e, any of $t_*'^-$), then it achieves a social welfare of $\frac{\lfloor \frac{\kappa^2}{2} \rfloor}{\kappa + 2} + \frac{\kappa}{2} + 1$. The resulting social welfare of case 2 is deceptively higher than that of case 1 because, although it initially appears higher, it can mislead the algorithm towards a suboptimal path.*

*It can be verified that Algorithm 1 reveals, at each iteration, one of the $t_*'^-$ targets, until the budget $K = \kappa + 1$ is fully used. As a result, at $K = \kappa + 1$ budget, $\kappa + 1$ agents can each emulate a positive target with probability 1, and the rest of the $\lfloor \frac{\kappa^2}{2} \rfloor$ agents can each do this at probability $\frac{1}{\kappa + 2}$. Therefore the approximation ratio is*

$$\frac{\frac{\lfloor \frac{\kappa^2}{2} \rfloor}{\kappa + 2} + \kappa + 1}{\lfloor \frac{\kappa^2}{2} \rfloor + \frac{\kappa + 1}{2}} = \frac{\frac{\lfloor \frac{\kappa^2}{2} \rfloor + \kappa^2 + 3\kappa + 2}{\kappa + 2}}{\lfloor \frac{\kappa^2}{2} \rfloor + \frac{\kappa + 1}{2}} < \frac{\kappa^2 \left(3 + \frac{6}{\kappa} + \frac{4}{\kappa^2}\right)}{\kappa^3 \left(1 + \frac{3}{\kappa} + \frac{3}{\kappa^2} + \frac{2}{\kappa^3}\right)} < \frac{3}{\kappa} < \frac{3}{\sqrt{2n}}$$ *which is significantly much worse than $1 - \frac{1}{e}$ for $K = \kappa + 1, \kappa > 3$.*

### C.3.2 Proof of Theorem 7

**Theorem 7.** *When the social planner can reveal both positive and negative targets, there exists a graph and a budget $K$ for which Algorithm 1 attains an approximation ratio strictly less than $2/\sqrt{n} + 2$.*

*Proof.* The proof is in Example 3 below. $\qquad\square$

**Example 3.** *Let $\kappa \in \mathbb{N}$ with $\kappa \geq 3$, and set the reveal budget to $K := \kappa + 1$. Consider the bipartite graph in Figure 8. There are $n = \kappa^2$ agents, each connected to a unique positive target and to all the $m^- = \kappa + 1 = \sqrt{n} + 1$ negative targets.*

*Before any target is revealed, the social welfare is $F(\emptyset) = \frac{\kappa^2}{\kappa + 2}$. Revealing any negative target in the first iteration increases the welfare to $F(\{t_*^-\}) = \frac{\kappa^2}{\kappa + 1}$, giving a marginal gain of $\Delta_{t_*^-}(\emptyset) = \frac{\kappa^2}{\kappa + 1} - \frac{\kappa^2}{\kappa + 2} = \frac{\kappa^2}{(\kappa + 1)(\kappa + 2)}$.*

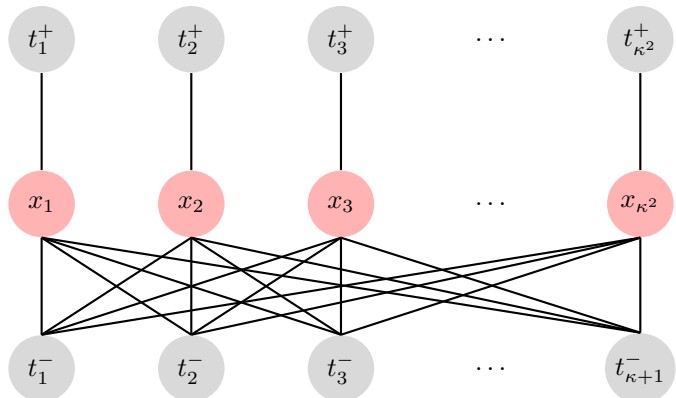

**Figure 8:** A bipartite graph example with statistics: $n = \kappa^2$, $m^- = \kappa + 1$, and $m^+ = n$. Algorithm 1 achieves an approximation ratio of $\frac{2}{\sqrt{n}+2}$.

*In contrast, revealing any positive target yields $F(\{t_*^+\}) = \frac{\kappa^2+\kappa+1}{\kappa+2}$, with marginal gain $\Delta_{t_*^+}(\emptyset) = \frac{\kappa^2+k+1}{\kappa+2} - \frac{\kappa^2}{\kappa+2} = \frac{\kappa+1}{\kappa+2} > \frac{\kappa^2}{(\kappa+1)(\kappa+2)}$.*

*Consequently, Algorithm 1 reveals a positive target in the first iteration and continues to do so in subsequent steps. The problem is that, under the budget $K = \kappa + 1$, revealing the negative targets would achieve the optimal social welfare $F(S^\star) = \kappa^2$. However, the marginal gain from releasing negative targets is not only initially much smaller than that of positive targets but also increases only when additional negative targets are revealed. As a result, the Algorithm 1 never reveals them.*

*At $K = \kappa + 1$ budget, $\kappa + 1$ agents can each emulate a positive target with probability $1$, and the rest of the agents can each do this at probability $\frac{1}{\kappa+2}$. Therefore the approximation ratio is $\frac{\kappa+1+\frac{\kappa^2-(\kappa+1)}{\kappa+2}}{\kappa^2} = \frac{2+\frac{2}{k}+\frac{1}{k^2}}{k+2} = \frac{2}{k+2} + \frac{2}{k(k+2)} + \frac{1}{k^2(k+2)} < \frac{2}{\kappa+2} = \frac{2}{\sqrt{n}+2}$ which is significantly much worse than $1 - \frac{1}{e}$ for $K = \kappa + 1, \kappa > 2$.*

### C.3.3 Proof of Theorem 2

**Lemma 2.** *If all agents are c-bounded, then for any target set $S$, both the true social welfare and gain are approximated by their proxy counterparts within a factor of c. That is, $F_p(S) \le F(S) \le cF_p(S)$ and $G_p(S) \le G(S) \le cG_p(S)$.*

*Proof.* First, we show that the proxy social welfare is always less than or equal to the true social welfare. For a given agent $x$, consider three cases. One, if the agent has no neighbors $N(x) = \emptyset$, then $Q_p^S(x) = Q^S(x) = 0$. Second, if there is at least one revealed positive target neighbor of $x$ in $S$, then $Q_p^S(x) = Q^S(x) = 1$. Otherwise, the proxy probability mass is less than or equal to the true one, since $Q_p^S(x)$ only increases by the amount that revealing the first negative neighbor helped which is atmost the increase in $Q^S(x)$. This then implies that $Q_p^S(x) \le Q^S(x)$. Summing over all agents $x \in \mathcal{X}$ proves that for all $S \subseteq \mathcal{T}$, $F_p(S) \le F(S)$.

Next, we show that $F_p(S) \ge \frac{1}{c}F(S)$. Let $t \in \mathcal{T} \setminus A$ be a target revealed by the social planner such that $S = A \cup \{t\}$. Assume that each agent has at most $c \ge 1$ negative neighbors, $\delta_x^- \le c$ for all $x \in \mathcal{X}$. Then if $t$ is revealed to be positive $|P_x^+(S)| \ge 1$, both the proxy and true social welfare go up by atmost 1, that is, $Q_p^S(x) - Q_p^\emptyset(x) = Q^S(x) - Q^{\emptyset)}(x) = 1 - \frac{\delta_x^+}{|N(x)|}$. If $t$ is negative and $|P_x^+(S)| > 0$, then there will be no effect on both the proxy and true social welfare. Otherwise, if $t$ is negative and $|P_x^+(S)| = 0$ and $|P_x^-(S)| = \delta_x^-$,

the true social welfare increases by

$$Q^S(x) - Q^\emptyset(x) = 1 - \frac{\delta_x^+}{\delta_x^+ + \delta_x^-}$$
$$\geq 1 - \frac{1}{1 + \delta_x^-} \quad (\text{if } \delta_x^+ = 1)$$
$$\geq 1 - \frac{1}{1 + c} \quad (\text{if } \delta_x^- \leq c)$$
$$= \frac{c}{1 + c}$$

and the proxy social welfare increases by

$$Q_p^S(x) - Q_p^\emptyset(x) = \frac{\delta_x^+}{\delta_x^+ + \delta_x^- - 1}\left(1 + \frac{\delta_x^- - 1}{\delta_x^+ + \delta_x^-}\right) - \frac{\delta_x^+}{\delta_x^+ + \delta_x^-}$$
$$\geq \frac{1}{\delta_x^-}\left(\frac{2\delta_x^-}{1 + \delta_x^-}\right) - \frac{1}{1 + \delta_x^-} \quad (\text{if } \delta_x^+ = 1)$$
$$\geq \frac{2}{(1 + c)} - \frac{1}{1 + c} \quad (\text{if } \delta_x^- \leq c)$$
$$= \frac{1}{1 + c}$$

In this case $\frac{Q_p^S(x) - Q_p^\emptyset(x)}{Q^S(x) - Q^\emptyset(x)} \geq \frac{1}{c}$. Therefore, $\left(Q_p^S(x) - Q_p^\emptyset(x)\right) \geq \frac{1}{c}\left(Q^S(x) - Q^\emptyset(x)\right)$. Lastly, since $Q_p^\emptyset(x) = Q^\emptyset(x)$, and if $Q^S(x) > Q^\emptyset(x)$, then summing over all agents $x \in \mathcal{X}$, for all $S$, $F_p(S) \geq \frac{1}{c}F(S)$. Put together, $F_p(S) \leq F(S) \leq cF_p(S)$, and $G_p(S) \leq G(S) \leq cG_p(S)$. $\qquad\square$

**Lemma 3.** *When the social planner can reveal positive and negative targets, the proxy social welfare function is submodular. That is, for every $A, B \subseteq \mathcal{T}$ where $A \subseteq B$, every $t \in \mathcal{T} \setminus B$, $F_p(A \cup \{t\}) - F_p(A) \geq F_p(B \cup \{t\}) - F_p(B)$*

*Proof.* Consider an agent $x$, and two cases where $t$ is adjacent to $x$ and is either positive or negative. If the adjacent target $t$ is positive, then $Q_p^{B \cup \{t\}}(x) - Q_p^B(x) = 1 - Q_p^B(x)$ if there was previously no positive target neighbors of $x$ in $B$, and 0 if other positive target neighbors were already revealed $|P_x^+(B)| > 0$.

In the second case, if target $t$ is adjacent to $x$ and is revealed as negative, then the marginal gain is a constant $Q_p^{B \cup \{t\}}(x) - Q_p^B(x) = \frac{\delta_x^+}{|N(x)|(|N(x)| - 1)}$ defined by the gain from revealing the first negative target.

In both cases, the marginal is non-increasing as the revealed target set grows, because revealing an additional adjacent target of the same label yields zero gain if it's positive and a constant gain if it's negative. Therefore, $Q_p^{A \cup \{t\}}(x) - Q_p^A(x) \geq Q_p^{B \cup \{t\}}(x) - Q_p^B(x)$ and summing over $x \in \mathcal{X}$ and given the sum rule for submodular functions, $F_p(A \cup \{t\}) - F_p(A) \geq F_p(B \cup \{t\}) - F_p(B)$. $\qquad\square$

**Corollary 4.** *When the social planner can reveal positive and negative targets, the proxy is submodular on the gain. That is, for every $A, B \subseteq \mathcal{T}$ where $A \subseteq B$, every $t \in \mathcal{T} \setminus B$, $G_p(A \cup \{t\}) - G_p(A) \geq G_p(B \cup \{t\}) - G_p(B)$*

*Proof.* This follows directly from Lemma 3. Since $G_p(A \cup \{t\}) - G_p(A) = F_p(A \cup \{t\}) - F(\emptyset) - F_p(A) + F(\emptyset)$ and $G_p(B \cup \{t\}) - G_p(B) = F_p(B \cup \{t\}) - F(\emptyset) - F_p(B) + F(\emptyset)$, then the proxy is submodular on the gain. $\qquad\square$

**Remark 5.** *Assume revealed targets may include negative ones. If all agents are c-bounded, then with respect to the true social welfare, $F(\emptyset) \geq \frac{1}{c}F(S^\star)$ where $S^\star$ denotes the optimal revealed set when using the true social welfare function (Eqn. 1). Let $S$ be either $\emptyset$ or any arbitrary solution set. Ignore any agent $x \in \mathcal{X}$ where $\delta_x^+ = 0$ since $Q^S(x) = Q^{S^\star}(x) = 0$. For agents with at least one positive target, $Q^S(x) \geq \frac{1}{c+1}Q^{S^\star}(x)$ since $Q^{S^\star}(x) \leq 1$ and $Q^S(x) \geq \frac{1}{c+1}$. Summing over all such agents yields $F(S) \geq \frac{1}{c+1}F(S^\star)$. For a large c, $F(S) \geq \frac{1}{c}F(S^\star)$. Thus, any solution set (including empty set) achieves a (1/c)-factor approximation to the true optimal welfare.*

*Proof of Theorem 2.* For a target reveal budget of $K$, let $S$ denote the solution returned by Algorithm 1, and let $S^\star$ denote the optimal solution set under the true social welfare function $F$ (Eqn. 1). Let $S_p$ denote the solution returned by proxy-greedy, and $S_p^\star$ the optimal solution set under the proxy social welfare function $F_p$ (Defn. 3.1). Since $S_p^\star$ maximizes $F_p$, then $G_p(S_p^\star) \geq G_p(S^\star)$. Since by Lemma 2 $G_p(S^\star) \geq \frac{1}{c}G(S^\star)$, then $G_p(S_p^\star) \geq G_p(S^\star) \geq \frac{1}{c}G(S^\star)$. By Lemma 3, the proxy welfare function is submodular and therefore proxy-greedy achieves a $(1-1/e)$-approximation to the optimal proxy welfare, i.e., $G_p(S_p) \geq (1-1/e)G_p(S_p^\star)$. Since by Lemma 2, $G(S_p) \geq G_p(S_p)$, then, $G(S_p) \geq G_p(S_p) \geq (1-1/e)F_p(S_p^\star) \geq \frac{1-1/e}{c}G(S^\star)$. Thus, $G(S_p) \geq \frac{1-1/e}{c}G(S^\star)$. $\qquad\square$

### C.4 Proofs for Section 3.3

**Lemma 6.** *If the social welfare function is submodular, then* $\mathrm{OPT}^{K_a} \geq \mathrm{OPT}^K/w$.

*Proof.* Let $S_K^\star = \{t_1, \ldots, t_K\}$ be the optimal revealed target set for the whole graph under budget $K$, so that $\mathrm{OPT}^K = F(S_K^\star) - F(\emptyset)$, and define $S_K^\star$ prefix sets as $S_i = \{t_1, \ldots, t_i\}$ for $i \in [K]$, with $S_0 = \emptyset$. Thus $\mathrm{OPT}^K$ as a sum of successive marginal gains is $\sum_{i=1}^{K}\big(F(S_i) - F(S_{i-1})\big) - F(\emptyset)$. Similarly, for $K_a = \lceil K/w \rceil$, optimal welfare gain is $\mathrm{OPT}^{K_a} = \sum_{i=1}^{K_a}\big(F(S_i) - F(S_{i-1})\big) - F(\emptyset)$. By submodularity of the social welfare function, these marginal gains form a non-increasing sequence, that is, $F(S_j) - F(S_{j-1}) \geq F(S_i) - F(S_{i-1})$ for all $j < i$. Therefore, the average marginal gain over the first $K_a$ targets is at least the average over all $K$ targets. That is, $\frac{1}{K_a}\mathrm{OPT}^{K_a} \geq \frac{1}{K}\mathrm{OPT}^K$. Since $K_a = \lceil K/w \rceil$ then $\mathrm{OPT}^{K_a} \geq \mathrm{OPT}^K/w$. $\qquad\square$

**Corollary 7.** *If the social welfare function is submodular, then* $\mathrm{OPT}_a^{K_a} \geq \mathrm{OPT}_a^K/w$.

*Proof.* Let $\mathrm{OPT}_a^K$ denote the optimal social welfare gain obtained when we focus exclusively on group $a$ with budget $K$, and let $\mathrm{OPT}_a^{K_a}$ denote the optimal welfare gain under a smaller budget $K_a = \lceil K/w \rceil$. Since both welfare gains depend only on the nodes and edges within group $a$, then that graph can itself be viewed as the entire graph. Thus, by Lemma 6, $\mathrm{OPT}_a^{K_a} \geq \mathrm{OPT}_a^K/w$. $\qquad\square$

**Theorem 8.** *For any arbitrary graph, given a limit $K \geq w$ on the reveal budget and restriction to revealing positive targets, Algorithm 1 outputs a solution set that is simultaneously $((1-1/e)/w)$-approximately optimal for each group. That is, for any group $a \in [w]$, the algorithm reveals target set $S_{K_a} \subseteq \mathcal{T}^+$ with $|S_{K_a}| \leq K_a$ such that $G(S_{K_a}) \geq \frac{(1-1/e)}{w}\mathrm{OPT}_a^K$, where $\mathrm{OPT}_a^K$ is the maximum social welfare gain for group $a$ with budget $K$, restricting candidate targets to $\mathcal{T}^+$ and optimizing only over that group.*

*Proof.* Let each group $a$ be assigned a budget $K_a = \lceil K/w \rceil$, such that Algorithm 1 run exclusively on each group $a$ returns a target set $S_{K_a} \subseteq \mathcal{T}^+$ with $|S_{K_a}| \leq K_a$ such that the social welfare gain of the group is given by $G(S_{K_a}) = F(S_{K_a}) - \sum_{x \in A_a} Q^{\emptyset}(x)$. Since the social planner is restricted to only revealing positive targets and the social welfare gain function is monotone and submodular, then for each group $a \in [w]$, $G(S_{K_a}) \geq (1-1/e)\mathrm{OPT}_a^{K_a}$. By Corollary 7, $(1-1/e)\mathrm{OPT}_a^{K_a} \geq \frac{(1-1/e)}{w}\mathrm{OPT}_a^K$. Combining everything, it then follows that for any group $a \in [w]$, $G(S_{K_a}) \geq \frac{(1-1/e)}{w}\mathrm{OPT}_a^K$. $\qquad\square$

**Remark 8.** *When the revealed target set includes negative targets and the social welfare function is monotone but not necessarily submodular (cf. Proposition 3), there may be no solution that substantially benefits multiple groups at once. For example, consider a case with two groups, each defined by a bipartite graph illustrated in Figure 7 in Appendix C.3. In this case, achieving high social welfare may require allocating all or nearly all of the target reveal budget $K$ to a single group.*

**Corollary 9.** *If the revealed targets include negative ones, and all agents are $c$-bounded, then under the proxy-greedy algorithm, each group $a$ receives welfare gain at least $\Omega(\mathrm{OPT}_a^K)$.*

*Proof.* For a given target reveal budget $K$, let $G(S_a)$ denote the welfare gain obtained by running Algorithm 1 exclusively on group $a \in [w]$, and let $\mathrm{OPT}_a^K$ be the corresponding optimal social welfare gain under the true social welfare function $F$ (Eqn. 1). Let $G_p(S_{p,a})$ denote the proxy social welfare gain obtained by running proxy-greedy on that group, and let $\mathrm{OPT}_{p,a}^K$ be the optimal gain under the proxy social welfare function

$F_p$ (Defn. 3.1). By Lemma 2, $G(S_{p,a}) \geq G_p(S_{p,a})$, and by Theorem 2, $G_p(S_{p,a}) \geq (1 - 1/e)\text{OPT}_{p,a}^K \geq \frac{1-1/e}{c}\text{OPT}_a^K$, for any group $a \in [w]$. Put together, $G(S_{p,a}) \geq \frac{1-1/e}{c}\text{OPT}_a^K$. □

**Equal proxy social welfare doesn't imply similar emulation choices.** Consider the example shown in Figure 6, where there are 4 agents, each connected to a unique positive target and 2 common negatives. Here, revealing 2 negative or 2 positive targets results in equal proxy social welfare (8/3), but the two reveals have different implications on agents' emulation choices. Revealing two positive targets ensures that 2/4 agents can emulate a positive target with certainty and the other two can emulate a positive target with probability 1/3 and a negative target with probability 1/3. On the other hand, revealing two negative targets ensures that all the agents can emulate a positive target with probability 2/3 and a negative target with probability 0 since revealing a negative target means agents avoid it or their probability of emulating it is 0. In this case, revealing 2 negative targets results in better emulation choices for all agents, but proxy-greedy might choose to reveal 2 positive targets instead.

### C.5 Proofs for Section 3.4

#### C.5.1 Proof of Theorem 3

*Proof of Theorem 3.* We prove NP-hardness by reducing the max-$K$-cover problem to the problem below.

**Problem 1.** *Consider a bipartite graph $\mathcal{G} = (\mathcal{X} \cup \mathcal{T}, E)$ with agents $\mathcal{X}$ and targets $\mathcal{T}$, where each target $t \in \mathcal{T}$ has a label $f(t) \in \{+1, -1\}$. For each agent $x \in \mathcal{X}$, let their neighborhood be $N(x) = \{t \in \mathcal{T} \mid (x, t) \in E\}$. The goal of the social planner is to find a subset $S \subseteq \mathcal{T}^+$ of at most $K$ positively labeled targets, where $\mathcal{T}^+ = \{t \in \mathcal{T} \mid f(t) = +1\}$ that, when revealed, maximizes social welfare $F(S) = \sum_{x \in \mathcal{X}} Q^S(x)$. The decision problem asks whether there exists such a positive target subset $S$ with $F(S) \geq W$ where $W$ is the welfare threshold.*

We prove the NP-hardness by reducing the max-$K$-cover problem with varied sized sets to Problem 1. In the max-$K$-cover problem, we are given a universe of elements $U = \{e_1, \ldots, e_n\}$, a family of sets $\mathcal{C} = \{C_1, \ldots, C_m\}$ with $C_j \subseteq U$, and a budget of $K$. The goal is to select at most $K$ sets out of $\mathcal{C}$ whose union covers at least $\mathcal{E}$ elements in the universe.

First, we reduce max-$K$-cover to Problem 1 by constructing, in polynomial time, an instance in which selecting a positive target set $S$ with $|S| \leq K$ corresponds exactly to choosing up to $K$ sets in the max-$K$-cover instance, and the resulting social welfare reflects the achieved coverage. For each element $e_i \in U$ we create an agent $x_i$, and for each set $C_j \in \mathcal{C}$, a positive target $t_j$, with an edge $(x_i, t_j)$ if and only if $e_i \in C_j$. To ensure that all agents have a similar initial contribution to social welfare, each agent $x_i$ with neighborhood size $|N(x_i)|$ is additionally connected to $|N(x_i)|$ unique negative targets, distinct across agents. In this construction, before any positive targets are revealed, each agent contributes $\frac{1}{2}$ to social welfare, so $F(\emptyset) = \frac{n}{2}$. Revealing a positive target $t_j$ increases the contribution of every adjacent agent from $\frac{1}{2}$ to 1.

Then, for any revealed set $S \subseteq \mathcal{T}^+$ where $|S_g| \leq K$,

$$F(S) = \frac{n}{2} + \frac{1}{2}\left|\bigcup_{t_j \in S} C_j\right|.$$

Setting the welfare threshold to $W = \frac{n}{2} + \frac{\mathcal{E}}{2}$, the condition $F(S) \geq W$ is equivalent to $\left|\bigcup_{t_j \in S} C_j\right| \geq \mathcal{E}$. Thus, there exists a set of at most $K$ revealed positive targets achieving welfare at least $W$ if and only if there exist at most $K$ sets covering at least $\mathcal{E}$ elements in the original instance. Since $F(S)$ is computable in polynomial time, Problem 1 lies in NP, its decision version is NP-complete, and the corresponding optimization problem of selecting $S_g \subseteq \mathcal{T}^+$ with $|S_g| \leq K$ to maximize social welfare is NP-hard.

Next, we show that the reduction preserves approximation hardness. Since the max-$K$-cover problem is NP-hard to approximate within any factor strictly greater than $1 - \frac{1}{e}$ unless P = NP, any approximation for Problem 1 would yield an approximation of the same quality for max-$K$-cover. In Problem 1, for any positive

target solution set $S$ and an optimal solution set $S^\star$, the construction ensures that $\frac{F(S)-\frac{n}{2}}{F(S^\star)-\frac{n}{2}} = \frac{\left|\bigcup_{t_j \in S} C_j\right|}{\left|\bigcup_{t_j \in S^\star} C_j\right|}$.

Therefore, a polynomial time $\alpha$-approximation for maximizing social welfare in Problem 1 induces a polynomial time $\alpha$-approximation for maximizing coverage in the max-$K$-cover problem. Consequently, if Problem 1 admitted a polynomial-time approximation factor strictly better than $1 - \frac{1}{e}$, then max-$K$-cover would also admit such an approximation, contradicting known hardness results. Hence, unless P = NP, selecting at most $K$ positive targets to maximize social welfare cannot be approximated within a factor better than $1 - \frac{1}{e}$. $\qquad\square$

### C.5.2   Proof of Theorem 4

*Proof of Theorem 4.* We prove NP-hardness by reducing the $K$-clique problem in a graph where all vertices have the same degree $\theta$, to the problem below.

**Problem 2.** *Consider a bipartite graph $\mathcal{G} = (\mathcal{X} \cup \mathcal{T}, E)$ with agents $\mathcal{X}$ and targets $\mathcal{T}$, where each target $t \in \mathcal{T}$ has a label $f(t) \in \{+1, -1\}$. For each agent $x \in \mathcal{X}$, let their neighborhood be $N(x) = \{t \in \mathcal{T} \mid (x,t) \in E\}$. The goal of the social planner is to find a subset $S \subseteq \mathcal{T}^-$ of at most $K$ negatively labeled targets, where $\mathcal{T}^- = \{t \in \mathcal{T} \mid f(t) = -1\}$ that, when revealed, maximizes social welfare $F(S) = \sum_{x \in \mathcal{X}} Q^S(x)$. The decision problem asks whether there exists such a negative target subset $S$ with $F(S) \geq W$ where $W$ is the welfare threshold?*

We prove the NP-hardness by reducing the $K$-clique problem where all vertices have the same degree $\theta$, to Problem 2. In the $K$-clique problem, we are given a graph $\mathcal{G}_c = (V, E_c)$ where $|V| = m$ and $|E_c| = n$. The goal is to find if a clique of size $K$ exists in graph $\mathcal{G}_c$. That is $C \subseteq V$, $|C| \leq K$ such that every pair in $C$ is an edge.

First, we show how to construct an instance of Problem 2 from any instance of the $K$-clique problem in polynomial time, such that revealing a negative target set $S$ with $|S| \leq K$ corresponds exactly to finding a clique of size $K$ in the $K$-clique instance, and the resulting social welfare $F(S)$ is at least the number of edges in the clique. To do so, we create one negative target $t_v^-$ for each vertex $v \in V$, one positive target $t_{uv}^+$ for each edge $\{u, v\} \in E_c$, and one agent $x_{uv}$ for each edge $\{u, v\} \in E_c$. Therefore each agent's neighborhood is defined as $N(x_{uv}) = \{t_u^-, t_v^-, t_{uv}^+\}$, and a negative target with no edge has no agent, because if an agent were connected to it, it would contribute 0 to social welfare. The target reveal budget is set to $K$ and the welfare threshold to $W$.

In this construction, each agent initially, before revealing any negative target (i.e., $S = \emptyset$), contributes $\frac{1}{3}$ to the social welfare, that is, $F(\emptyset) = \frac{n}{3}$. Revealing a negative target $t_u^-$ increases the contribution of each agent connected to it by $\frac{1}{6}$ because their probability of emulating a positive target goes from $\frac{1}{3}$ to $\frac{1}{2}$. Revealing two negative targets $\{t_u^-, t_v^-\}$ increases the contribution of each agent $x_{uv}$ connected to them by $\frac{2}{3}$ since their probability of emulating a positive target in their neighborhood goes from $\frac{1}{3}$ to 1. Every negative target in the revealed target set $S \subseteq \mathcal{T}^-$, $|S| \leq K$ is connected to $K - 1$ other negative targets in $S$ and $\theta - (K-1)$ negative targets outside the target set $S$. For $K$ negative targets inside $S$, there are $K(\theta - (K-1)) = K\theta - 2\binom{K}{2}$ agents with one endpoint in the $S$, and within $S$, there are $\binom{K}{2} = K(K-1)/2$ agents with 2 endpoints in $S$. Hence, the social welfare satisfies

$$F(S) = \frac{n}{3} + \frac{1}{6}\left(K\theta - 2\binom{K}{2}\right) + \frac{2}{3}\binom{K}{2} = \frac{n}{3} + \frac{K\theta}{6} + \frac{1}{3}\binom{K}{2},$$

so that there exists a $K$-clique *iff* there exists $S$ with $|S| = K$ such that $F(S) \geq W$. Consequently, Problem 2 is NP-hard because it can be reduced from the $K$-clique problem. Since any candidate positive target set $S$ can be verified in polynomial time, the decision problem is in NP, and therefore NP-complete. Therefore, the problem of finding a positive target subset $S_g \subseteq \mathcal{T}$ with $|S_g| \leq K$ that maximizes social welfare is NP-hard. $\qquad\square$

---

**Algorithm 3** The $d$-step Lookahead Greedy Target Reveal

---

1: **Input:** Graph $\mathcal{G} = (\mathcal{X} \cup \mathcal{T}, E)$, labels $\{f(t)\}_{t \in \mathcal{T}}$, budget $K$, depth $d$
2: **Output:** Solution set $S_{\mathrm{lg}} \subseteq \mathcal{T}$ with $|S_{\mathrm{lg}}| \leq K$, and social welfare $F(S_{\mathrm{lg}})$
3: Initialize $S_{\mathrm{lg}} \leftarrow \emptyset$
4: **while** $|S_{\mathrm{lg}}| \leq K$ **do**
5: $\quad b \leftarrow \min(d, \ K - |S_{\mathrm{lg}}|)$
6: $\quad S^{iter} \leftarrow \arg \max\limits_{\substack{S \subseteq (\mathcal{T} \setminus S_{\mathrm{lg}}) \\ |S| \leq b}} \left( F(S_{\mathrm{lg}} \cup S) - F(S_{\mathrm{lg}}) \right)$
7: $\quad$ **if** $F(S_{\mathrm{lg}} \cup S^{iter}) > F(S_{\mathrm{lg}})$ **then**
8: $\quad\quad S_{\mathrm{lg}} \leftarrow S_{\mathrm{lg}} \cup S^{iter}$
9: $\quad$ **else**
10: $\quad\quad$ **break**
11: **return** $(S_{\mathrm{lg}}, \ F(S_{\mathrm{lg}}))$

---

### C.6 Alternative Greedy Strategies

Due to the performance limitations of the classic greedy approach for budgeted target selection aimed at maximizing social welfare (as discussed in Section 3.2), this section proposes alternative greedy strategies, examines them, and compares their effectiveness with that of the classic method.

#### C.6.1 The $d$-step Lookahead Greedy Approach

The $d$-step lookahead greedy algorithm (Algorithm 3) generalizes Algorithm 1 by revealing up to $d \in \mathbb{Z}_{\geq 1}$ targets per iteration, chosen to maximize the marginal gain in social welfare.

When $d$ in Algorithm 3 is equivalent to the target reveal budget ($d = K$), Algorithm 3 reduces to bruteforce search (Appendix C.2), which evaluates all $m^K$ possible $K$ sized subsets of targets $\mathcal{T}$ to identify the one that maximizes social welfare.

**Proposition 6.** *The $d$-step lookahead greedy (Algorithm 3) runs in $O\left(\frac{K}{d} m^d n \delta\right)$ time.*

*Proof.* At each iteration, Algorithm 3 reveals a target set of size at most $d$ with the maximum marginal gain, that is $S^{iter} = \arg \max\limits_{\substack{S \subseteq (\mathcal{T} \setminus S_{\mathrm{lg}}) \\ |S| \leq b}} \left( F(S_{\mathrm{lg}} \cup S) - F(S_{\mathrm{lg}}) \right)$. To find this subset, among the possible number of subsets $O(m^d)$, the algorithm evaluates the new social welfare $F(S_{\mathrm{lg}} \cup S)$ for each candidate subset $S$, which takes $O(m^d n \delta)$ time in total. Since each iteration reveals at most $d$ targets, then there are at most $O(\frac{K}{d})$ iterations. Therefore, the total running time is $O(\frac{K}{d} m^d n \delta)$. $\qquad\square$

**Comparison of Classic (Algorithm 1) to Lookahead (Algorithm 3).** While classic greedy algorithm runs in polynomial time ($O(Kmn\delta)$) (Proposition 4), the $d$-step lookahead greedy algorithm incurs a higher computational cost of $O\left(\frac{K}{d} m^d n \delta\right)$.

There exist cases (Proposition 7) where at a relatively low computational cost, the 2-step lookahead significantly outperforms the classic greedy approach. However, as shown in Proposition 8, there exist cases where $d$-step lookahead surpasses Algorithm 1 only when the lookahead depth ($d$) is the equal to the target reveal budget ($K$). Therefore, to try and balance good performance with computational efficiency, we propose the heuristic greedy approach (Appendix C.6.2).

**Proposition 7.** *There exists a graph and a budget $K$ for which a 2-step lookahead algorithm finds the exact optimal solution with low computational overhead, while the classic greedy algorithm attains an approximation ratio strictly less than $\frac{2}{\sqrt{n}+1}$.*

*Proof.* Proof is shown in Example 4 below. $\qquad\square$

**Example 4.** *Let $\kappa \in \mathbb{N}$ with $\kappa \geq 3$, and set the reveal budget to $K := \kappa$. Consider the bipartite graph with $n = \kappa^2$ agents, where each agent is adjacent to a unique positive target and to all $m^- = \kappa$ negative targets.*

*Initially, the social welfare is $F(\emptyset) = \frac{\kappa^2}{\kappa+1}$. Algorithm 1 is indifferent in the first step: revealing either a positive or a negative target yields the same marginal gain. That is, $\left(\frac{\kappa^2}{k} = \frac{\kappa^2+\kappa}{\kappa+1} = \kappa\right) - \frac{\kappa^2}{\kappa+1}$. Once this tie appears, the initial choice dictates the entire trajectory. An initial positive target reveal leads Algorithm 1 to keep selecting positive targets and would require $\kappa^2$ iterations for Algorithm 1 to achieve the optimal social welfare. On the other hand, an initial negative target reveal commits it to negative targets and reaches the optimum in only $\kappa$ iterations.*

*In contrast, Algorithm 3 with $d = 2$ foresees these outcomes and consistently selects negative targets. As a result, it attains the optimal solution for every $\kappa > 1$ consistent with this bipartite graph structure. This shows that a two-step lookahead greedy approach, while remaining comparatively inexpensive to compute, can guarantee an exact solution. Algorithm 1 doesn't. When it commits to positive targets, its approximation ratio can be less than $\frac{2}{\sqrt{n}+1}$.*

**Proposition 8.** *There exists a graph and a budget $K$ for which a $K$-step lookahead algorithm finds the exact optimal solution with high computational overhead, while the classic greedy algorithm attains an approximation ratio strictly less than $1/\sqrt{0.5n}$.*

*Proof.* Proof is shown in Example 5 below. $\qquad\square$

**Example 5.** *Let $\kappa \in \mathbb{N}$ with $\kappa \geq 7$, and set the reveal budget to $K := \kappa + 1$. Consider the bipartite graph with $n = \lfloor \kappa^2/(\kappa - 2) \rfloor + \kappa$ agents, each connected to a unique positive target and to all $\kappa+1$ negative targets.*

*Initially, the social welfare is $(\lfloor \kappa^2/(\kappa - 2) \rfloor + \kappa)/(\kappa + 2)$. Selecting any negative target increases welfare to $(\lfloor \kappa^2/(\kappa - 2) \rfloor + \kappa)/(\kappa + 1)$, whereas selecting a positive target results in larger increase $(2\kappa + 1 + \lfloor \kappa^2/(\kappa - 2) \rfloor)/(\kappa + 2)$. As a result, Algorithm 1 only reveals positive targets, despite the negative ones being better in the long run, capable of achieving the optimal welfare of $\lfloor \kappa^2/(\kappa - 2) \rfloor + \kappa$ at budget $K$.*

*With only positive targets revealed, $\kappa + 1$ agents emulate a positive target with probability one, while the remaining agents do so with probability $1/(\kappa+2)$. This yields an approximation ratio of $\frac{\kappa+1+\frac{\lfloor \frac{\kappa^2}{\kappa-2} \rfloor + \kappa - (\kappa+1)}{\kappa+2}}{\kappa^2} < \frac{1}{\kappa} < 1/\sqrt{0.5n}$.*

*In contrast, Algorithm 3 achieves an exact solution for $\kappa \geq 7$ when its depth $d = K$. While it performs significantly better than the classic greedy algorithm, it incurs substantially higher computational cost.*

### C.6.2 The (Interactive) Heuristic Greedy Approaches

In this section, first, we present the heuristic greedy approach and then then present the interactive heuristic greedy algorithm. Although the proposed heuristics are flexible and can incorporate various algorithms, here we focus on the setting in which the inserted algorithm is the classic greedy method (Algorithm 1).

**The heuristic greedy approach**

**Overview of heuristic greedy algorithm (Algorithm 4).** The heuristic greedy algorithm proceeds as follows. Given a budget $K$, the heuristic greedy algorithm considers all budget splits $\kappa \in \{0, \ldots, K\}$. For each $\kappa$, Algorithm 1 is run in parallel with budget $\kappa$ and restriction to positive targets $\mathcal{T}' = \mathcal{T}^+ = \{t \in \mathcal{T} \mid f(t) = +1\}$, producing $S_+^\kappa$, and with budget $K - \kappa$ and restriction to negative targets $\mathcal{T}' = \mathcal{T}^- = \{t \in \mathcal{T} \mid f(t) = -1\}$, producing $S_-^{K-\kappa}$. The algorithm returns the $S_{\text{hg}}$ that maximizes the social welfare, i.e.,

$$S_{\text{hg}} = \arg \max_{\kappa \in [0,K]} F\left(S_+^{(\kappa)} \cup S_-^{(K-\kappa)}\right)$$

**The heuristic greedy algorithm runs in polynomial time.** Proposition 9 shows that Algorithm 4 runs in $O(K^2 mn\delta)$ time, where $\delta$ is the maximum agent degree, $m$ the number of targets, and $n$ the number of agents.

---

**Algorithm 4** Heuristic Greedy Target Reveal

---

1: **Input:** Graph $\mathcal{G} = (\mathcal{X} \cup \mathcal{T}, E)$, labels $\{f(t)\}_{t \in \mathcal{T}}$, budget $K$
2: **Output:** Solution set $S_{\text{hg}} \subseteq \mathcal{T}$ with $|S_{\text{hg}}| \leq K$, and social welfare $F(S_{\text{hg}})$
3: $\mathcal{R} \leftarrow \emptyset$
4: **for** $\kappa = 0$ to $K$ **do**
5: $\quad (S_+^{(\kappa)}, F_+^{(\kappa)}) \leftarrow \text{GREEDYLABELREVEAL}(\mathcal{G}, \mathcal{T}^+, f, \kappa)$          ▷ Algorithm 1 with $\mathcal{T}' = \mathcal{T}^+$
6: $\quad (S_-^{(\kappa)}, F_-^{(\kappa)}) \leftarrow \text{GREEDYLABELREVEAL}(\mathcal{G}, \mathcal{T}^-, f, K - \kappa)$      ▷ Algorithm 1 with $\mathcal{T}' = \mathcal{T}^-$
7: $\quad S^{(\kappa)} \leftarrow S_+^{(\kappa)} \cup S_-^{(\kappa)}$
8: $\quad F^{(\kappa)} \leftarrow F(S^{(\kappa)})$
9: $\quad \mathcal{R} \leftarrow \mathcal{R} \cup \{(\kappa, S^{(\kappa)}, F^{(\kappa)})\}$
10: $(\kappa_{\text{hg}}, S_{\text{hg}}, F_{\text{hg}}) \leftarrow \arg \max\limits_{(\kappa, S^{(\kappa)}, F^{(\kappa)}) \in \mathcal{R}} F^{(\kappa)}$
11: **return** $(S_{\text{hg}}, F(S_{\text{hg}}))$

---

**Proposition 9.** *The heuristic greedy algorithm (Algorithm 4) runs in $O\left(K^2 mn\delta\right)$ time.*

*Proof.* At each iteration, Algorithm 1 is executed separately on the positive and negative targets. If each of the positive and negative target sets has size of at most $m$, then for any fixed iteration and reveal budget of $\kappa \in \{0, \dots, K\}$, the separate Algorithm 1 calls require $O(\kappa mn\delta) + O((K - \kappa)mn\delta)$ time, equal to $O(Kmn\delta)$. All remaining operations within the loop take constant time. After the $K$ iterations, the total cost becomes $O(K^2 mn\delta)$. To compute among the solutions, the one with the optimal social welfare would take $O(K)$ time. Thus, Algorithm 4 runs in $O(K^2 mn\delta)$ time. $\qquad\square$

**The interactive heuristic greedy approach**

Unlike the heuristic greedy approach which runs Algorithm 1 independently on the positive and negative targets, we analyze the interactive variant that couples the two phases.

The interactive heuristic greedy algorithm first applies Algorithm 1 to either the positive or negative targets for a budget of $\kappa \in \{0, \dots, K\}$. Then the revealed target set produced in this step becomes the initial set $S'$ for a second run of Algorithm 1 on the opposite target set, using the remaining budget $K - \kappa$. Below is the outline of the procedure.

---

**Algorithm 5** Interactive Heuristic Greedy Target Reveal

---

1: **Input:** Graph $\mathcal{G} = (\mathcal{X} \cup \mathcal{T}, E)$, labels $\{f(t)\}_{t \in \mathcal{T}}$, budget $K$
2: **Output:** Solution set $S_{\text{ihg}} \subseteq \mathcal{T}$ with $|S_{\text{ihg}}| \leq K$ and social welfare $F(S_{\text{ihg}})$
3: $\mathcal{R} \leftarrow \emptyset$
4: **for** $\kappa = 0, 1, \dots, K$ **do**
5: $\quad (S_+^{(\kappa)}, F_+^{(\kappa)}) \leftarrow \text{GREEDYLABELREVEAL}(\mathcal{G}, \mathcal{T}^+, f, \kappa)$
6: $\quad (S_-^{(\kappa)}, F_-^{(\kappa)}) \leftarrow \text{GREEDYLABELREVEAL}(\mathcal{G}, \mathcal{T}^-, f, \kappa)$
7: $\quad (S_+^{i(\kappa)}, F_+^{i(\kappa)}) \leftarrow \text{GREEDYLABELREVEAL}(\mathcal{G}, \mathcal{T}^+, f, K - \kappa, S' = S_-^{(\kappa)})$
8: $\quad (S_-^{i(\kappa)}, F_-^{i(\kappa)}) \leftarrow \text{GREEDYLABELREVEAL}(\mathcal{G}, \mathcal{T}^-, f, K - \kappa, S' = S_+^{(\kappa)})$
9: $\quad \mathcal{R} \leftarrow \mathcal{R} \cup \{(\kappa, S_+^{i(\kappa)}, F_+^{i(\kappa)}, S_-^{i(\kappa)}, F_-^{i(\kappa)})\}$
10: $(\kappa^\star, v^\star) \leftarrow \arg \max\limits_{\kappa \in \{0, \dots, K\}, \, v \in \{+, -\}} F_v^{i(\kappa)}$
11: $(S_{\text{ihg}}, F_{\text{ihg}}) \leftarrow (S_{v^\star}^{i(\kappa^\star)}, F_{v^\star}^{i(\kappa^\star)})$
12: **return** $(S_{\text{ihg}}, F(S_{\text{ihg}}))$

---

**Overview of Algorithm 5.** There are two settings we consider. In the first, Algorithm 1 is run on the positive targets before the negative targets. In the second case, the algorithm is run on the negative targets first, followed by a run on the positive targets for the remaining budget.

**Case 1:** Let $S_+^{(\kappa)}$ be the solution of Algorithm 1 when $\mathcal{T}' = \mathcal{T}^+$ at a given budget $\kappa$. Then, given the initialization $S' = S_+^{(\kappa)}$, let $S_-^{i(\kappa)}$ be the remaining part of solution set of Algorithm 1 when $\mathcal{T}' = \mathcal{T}^-$ at a given budget $K - \kappa$ such that $|S_-^{i(\kappa)}| \leq K$. That is,

$$S_-^{i(\kappa)} = \text{GREEDYLABELREVEAL}(\mathcal{G}, \mathcal{T}^-, f, K - \kappa, S' = S_+^{(\kappa)})$$

**Case 2:** Let $S_-^{(\kappa)}$ be the solution of Algorithm 1 when $\mathcal{T}' = \mathcal{T}^-$ at a given budget $\kappa$. Then, given the initialization $S' = S_-^{(\kappa)}$, let $S_+^{i(\kappa)}$ be the remaining part of solution set of Algorithm 1 when $\mathcal{T}' = \mathcal{T}^+$ at a given budget $K - \kappa$ such that $|S_+^{i(\kappa)}| \leq K$. That is,

$$S_+^{i(\kappa)} = \text{GREEDYLABELREVEAL}(\mathcal{G}, \mathcal{T}^+, f, K - \kappa, S' = S_-^{(\kappa)})$$

Run cases 1 and 2 until $\kappa = K$. Then, for each case, find the best solution, that is,

$$\kappa_+ = \arg\max_{\kappa \in \{0,\dots,K\}} F\big(S_+^{i(\kappa)}\big), \quad \kappa_- = \arg\max_{\kappa \in \{0,\dots,K\}} F\big(S_-^{i(\kappa)}\big),$$

Finally

$$S_{\text{ihg}}, F(S_{\text{ihg}}) = \begin{cases} S_-^{i(\kappa_-)}, F(S_-^{i(\kappa_-)}), & \text{if } F\big(S_-^{i(\kappa_-)}\big) > F\big(S_+^{i(\kappa_+)}\big), \\ S_+^{i(\kappa_+)}, F(S_+^{i(\kappa_+)}), & \text{otherwise.} \end{cases}$$

### C.6.3 Comparison of the Greedy Strategies

Similar to Algorithm 1, Algorithms 4 and 5 have polynomial-time complexity, making them significantly more efficient than the $d$-step lookahead greedy algorithm.

As shown in Proposition 10, the heuristic greedy algorithm can outperform the classic greedy approach and achieve performance comparable to that of the $d$-step lookahead algorithm. Nevertheless, there are instances where it performs no better than the classic greedy method, while the $d$-step lookahead algorithm with $d = K$ produces the optimal solution (Proposition 11), albeit at a substantially higher computational cost.

Notably, when revealed targets include negative ones and the social welfare function $F$ maybe supermodular, greedy methods face a trade-off between performance and computational cost. Therefore to asses practical performance, we empirically evaluate the classic, heuristic, and $K$-step lookahead greedy algorithms in a semi-synthetic setting (Section 5).

**Proposition 10.** *There exists graphs and target reveal budget values for which the heuristic greedy algorithm strictly outperforms the classic greedy approach and achieves performance comparable to that of the $d$-step lookahead algorithm.*

*Proof.* In Examples 3, 4, and 5, both the heuristic greedy algorithm and the $d$-step lookahead greedy algorithm recover the optimal solution exactly. In contrast, Algorithm 1 achieves approximation ratios strictly smaller than $\frac{2}{\sqrt{n}+2}$, $\frac{2}{\sqrt{n}+1}$, and $\frac{1}{\sqrt{0.5n}}$, respectively. $\square$

**Proposition 11.** *There exists a graph and a budget $K$ for which the $K$-step lookahead greedy algorithm strictly outperforms the classic and heuristic greedy approaches.*

*Proof.* In Example 2, both the heuristic and the classic greedy algorithms are misled into selecting a suboptimal sequence of targets, yielding a solution with approximation ratio strictly smaller than $\frac{3}{\sqrt{2n}}$. In contrast, the $d$-step lookahead greedy algorithm with $d = K$ anticipates these unfavorable choices and avoids suboptimal trajectories, thereby recovering the optimal solution, though at a substantially higher computational cost. $\square$

**Proposition 12.** *There exists a graph and budget $K$ for which both the classic greedy and interactive heuristic greedy algorithms find an exact solution, whereas heuristic greedy doesn't.*

*Proof.* Proof is in Example 6 below.

**Example 6.** *Consider the bipartite graph in Table 1, with ten agents and nine targets. Five targets are positive, $\{t_0^+, t_1^+, t_2^+, t_3^+, t_5^+\}$, and four are negative, $\{t_6^-, t_7^-, t_8^-, t_9^-\}$. Each row lists an agent along with its neighborhood.*

*For a budget of $K = 3$, the classic greedy and interactive heuristic greedy algorithms both yield an optimal target set $\{t_0^+, t_6^-, t_9^-\}$, where $\{t_0^+\}$ is positive and $\{t_6^-, t_9^-\}$ are negative. In contrast, the heuristic greedy algorithm selects only negative targets $\{t_6^-, t_7^-, t_9^-\}$, achieving an approximation ratio of 0.96.*

$\square$

| agent | neighborhood |
|-------|--------------|
| $x_0$ | $\{t_8^-, t_9^-\}$ |
| $x_1$ | $\{t_3^+, t_6^-, t_9^-\}$ |
| $x_2$ | $\{t_0^+, t_6^-, t_7^-, t_8^-, t_9^-\}$ |
| $x_3$ | $\{t_5^+, t_9^-\}$ |
| $x_4$ | $\{t_7^-\}$ |
| $x_5$ | $\{t_6^-, t_7^-, t_9^-\}$ |
| $x_6$ | $\{t_9^-\}$ |
| $x_7$ | $\{t_1^+, t_2^+, t_6^-, t_7^-\}$ |
| $x_8$ | $\{t_7^-, t_8^-\}$ |
| $x_9$ | $\{t_1^+, t_9^-\}$ |

**Table 1:** A bipartite graph where each agent (first column) is connected to its neighborhood of targets (second column) with labels initially unknown to the agents.

## D   Supplementary Material for Section 4.1

In this section, we present pre- and post-reveal intervention algorithms (i.e., Algorithms 6 and 7, respectively) along with their runtime analysis.

---

**Algorithm 6** Pre-reveal Targeted Intervention

---

1: **Input:**   Graph $\mathcal{G} = (\mathcal{X} \cup \mathcal{T}, E)$, labels $\{f(t)\}_{t \in \mathcal{T}}$, reveal budget $K$, intervene budget $B$
2: **Output:**   Solution set $S_{\text{ig}}$, and pre-reveal intervention social welfare $F(S_{\text{ig}})$
3: $S_o \leftarrow \emptyset$
4: **for** each agent $x \in \mathcal{X}$ **do**
5:     $Q^{S_o}(x) \leftarrow |\{t \in N(x) : f(t) = 1\}| / |N(x)|$
6: $B' \leftarrow \mathbf{1}[\exists t \in \mathcal{T} : f(t) = 1] \cdot \min\big(B, |\{x \in \mathcal{X} : Q^{S_o}(x) < 1\}|\big)$
7: $\mathcal{X}_{\text{hr}} \leftarrow \underset{\mathcal{X}' \subseteq \mathcal{X} \ | \ |\mathcal{X}'| = B'}{\arg\min} \sum_{x \in \mathcal{X}'} Q^{S_o}(x)$
8: $\mathcal{G}' = \mathcal{G}[\mathcal{X} \setminus \mathcal{X}_{\text{hr}}]$
9: $(S_{\text{ig}}, F(S)) \leftarrow \text{GreedyLabelReveal}(\mathcal{G}', \mathcal{T}, f, K)$         ▷ run Algorithm 1
10: $F(S_{\text{ig}}) \leftarrow F(S) + B'$
11: **return** $(S_{\text{ig}}, F(S_{\text{ig}}))$

---

**Runtime analysis for the targeted intervention model algorithms.**   Algorithms 6 and 7 run in $O(Kmn\delta)$ time, where $\delta$ is the maximum agent degree, $m$ the number of targets, and $n$ the number of agents.

**Proposition 13.** *Algorithms 6 and 7 each runs in $O(Kmn\delta)$ time.*

*Proof.* Focusing on the costly procedures in Algorithms 6 and 7, each of them requires running the classic greedy algorithm sub module for at most $n$ agents, taking $O(Kmn\delta)$ time. To identify the high-risk agents,

---

**Algorithm 7** Post-reveal Targeted Intervention

---

1: **Input:** Graph $\mathcal{G} = (\mathcal{X} \cup \mathcal{T}, E)$, labels $\{f(t)\}_{t \in \mathcal{T}}$, reveal budget $K$, intervene budget $B$
2: **Output:** Solution set $S_{\mathrm{gi}}$, and post-reveal intervention social welfare $F(S_{\mathrm{gi}})$
3: $(S_o, F(S_o)) \leftarrow \textsc{GreedyLabelReveal}(\mathcal{G}, \mathcal{T}, f, K)$ ▷ run Algorithm 1
4: **for** each agent $x \in \mathcal{X}$ **do**
5: $\quad Q^{S_o}(x) \leftarrow \begin{cases} 1, & \text{if } |\{t \in N(x) \cap S_o : f(t) = 1\}| > 0, \\ \dfrac{|\{t \in N(x) \setminus S_o : f(t) = 1\}|}{|\{t \in N(x) \setminus S_o\}|}, & \text{if } |N(x) \setminus S_o| > 0, \\ 0, & \text{otherwise.} \end{cases}$
6: $\quad B' \leftarrow \mathbf{1}[\exists t \in \mathcal{T} : f(t) = 1] \cdot \min\big(B, \ |\{x \in \mathcal{X} : \ Q^{S_o}(x) < 1\}|\big)$
7: $\quad \mathcal{X}_{\mathrm{hr}} \leftarrow \underset{\mathcal{X}' \subseteq \mathcal{X} \ | \ |\mathcal{X}'| = B'}{\arg\min} \sum_{x \in \mathcal{X}'} Q^{S_o}(x)$
8: $\quad F(S_{\mathrm{gi}}) \leftarrow F(S_o) + \sum_{x \in \mathcal{X}_{\mathrm{hr}}} (1 - Q^{S_o}(x))$
9: $\quad S_{\mathrm{gi}} \leftarrow S_o$
10: **return** $(S_{\mathrm{gi}}, F(S_{\mathrm{gi}}))$

---

both algorithms first compute $Q^{S_o}(x)$ for every agent, which adds $O(n\delta)$ to the computation time. Then ranking agents by $Q^{S_o}(x)$ value, would take $O(n \log n)$ time. Combining these terms results in a total running time of $O(Kmn\delta + n\delta + n \log n) = O(Kmn\delta)$. $\qquad\square$

## E    Supplementary Material for Section 4.2

**Overview of Algorithm 8.**    The coverage radius algorithm (Algorithm 8) proceeds as follows. Let $c \in \{0,1\}^n$ denote coverage of $n$ agents where 1 means that the agent is covered or reached, and 0 otherwise. For each target $t_i$, compute the distance to the nearest uncovered agent relative to the target's current radius $r_i$ that is, $g_i = \min_{j:c_j=0} \|t_i - x_j\|_2 - r_i$. Then, increase the radius $r_k$ of the target $t_k$ with the smallest cost by that cost $r_k = r_k + g_k$, update the remaining radius budget by subtracting the cost of coverage, and mark corresponding agent(s) as covered. Continue until the radius budget is exhausted.

**Proposition 14.** *Algorithm 8 runs in $O(mn(\log n + d))$ time.*

*Proof.* Algorithm 8 begins by computing all pairwise distances between the $m$ positive targets and $n$ agents, forming the matrix $D \in \mathbb{R}^{m \times n}$. This step requires $O(mnd)$ time. Given these distances, it then sorts, for each positive target, the distances to all $n$ agents, which costs $O(mn \log n)$ overall. The while loop contributes at most $O(mn)$. Combining all parts, the total running time is $O(mnd) + O(mn \log n) + O(mn) = O\big(mn(d + \log n)\big)$. $\qquad\square$

## F    Proof of Theorem 5

*Proof.* Let $x_1, \ldots, x_n \sim D$ be set of agents drawn i.i.d from $D$. For any fixed revealed target set $S \subseteq \mathcal{T}$, we can compute $Q^S(x_i) \in [0,1]$ for each agent. As a result, the empirical social welfare is given as $\hat{F}(S) = \frac{1}{n} \sum_{i=1}^{n} Q^S(x_i)$. Consider that the revealed target set $S \subseteq \mathcal{T}$ has error at least $\varepsilon$ for distribution $D$. That is, for a fixed revealed target set $S$, $|\hat{F}(S) - F(S)| \leq \varepsilon$ with probability at least $1 - \delta$. With a target reveal budget of $K$, each of the subsets $S$ will be of size at most $K$, and therefore, there is atmost $m^K$ potential subsets.

By Hoeffding's inequality, the probability that the revealed target set will have social welfare off by more than $\varepsilon$ can be bounded as follows, $\Pr\big(|\hat{F}(S) - F(S)| \geq \varepsilon\big) \leq 2\exp(-2n\varepsilon^2)$. By union bound over all the possible subsets $m^K$, then $\Pr\big(\exists S \in \mathcal{T} : \ |\hat{F}(S) - F(S)| \geq \varepsilon\big) \leq 2m^K \exp(-2n\varepsilon^2)$. To ensure this probability is at most $\delta$, it suffices that $2m^K \exp(-2n\varepsilon^2) \leq \delta$, which holds whenever $n \geq C\Big(\frac{1}{\varepsilon^2}\big(K \log m + \log \frac{1}{\delta}\big)\Big)$ for

---

**Algorithm 8** Greedy Coverage Radius

---

1: **Input:** Agents $\mathcal{X} \in \mathbb{R}^{n \times \rho}$, targets $\mathcal{T}^+ \in \mathbb{R}^{m \times \rho}$, radius budget $R \in \mathbb{R}_{\geq 0}$
2: **Output:** Radii $r$, Covered agents $c$
3: $D \leftarrow [\, d_{ij} \,]_{i,j} \in \mathbb{R}^{m \times n}$
4: $r_i \leftarrow 0, \ \forall i \in [m], \quad c_j \leftarrow 0, \ \forall j \in [n]$
5: $R' \leftarrow R$                                     $\triangleright$ remaining R budget
6: $(sorted\_dist, sorted\_idx) \leftarrow sort\_along\_rows(D)$
7: $ptr \leftarrow \mathbf{0}_m$
8: **while** $R' > 0$ **do**
9:      $cost_i \leftarrow +\infty, \ \forall i \in [m]$
10:     **for** $i = 1$ to $m$ **do**
11:         **while** $ptr_i < n$ and $c_{sorted\_idx_i[ptr_i]}$ **do**
12:            $ptr_i \leftarrow ptr_i + 1$
13:         **if** $ptr_i < n$ **then**
14:            $cost_i \leftarrow sorted\_dist_i[ptr_i] - r_i$
15:     $t \leftarrow \arg\min_i cost_i$
16:     **if** $cost_t = +\infty$ or $cost_t > R'$ **then**
17:         **break**
18:     $r_t \leftarrow r_t + cost_t$
19:     $R' \leftarrow R' - cost_t$
20:     $a \leftarrow sorted\_idx_t[ptr_t]$
21:     $c_a \leftarrow 1$
22:     $ptr_t \leftarrow ptr_t + 1$
23: **return** $(r, c)$

---

a suitable universal constant $C > 0$. Under this condition, all revealed target sets $S$ of size at most $K$, including $S_g$, have empirical social welfare within $\varepsilon$ of their true value with probability at least $1 - \delta$.    $\square$

## G    Supplementary Material for Section 5

All experimentation, including bipartite graph generation, algorithmic computations and comparative analytics were performed on a CPU-based system with the following specifications: a 2.6-GHz 6-Core Intel Core i7 processor, 16 GB of 2400-MHz DDR4 RAM, and an Intel UHD Graphics 630 GPU with 1536 MB of memory.

### G.1    Experimental Setup

#### G.1.1    Datasets

We utilized four datasets obtained from the UCI Machine Learning Repository. The first was the **Adult** (Adult Income) dataset (Becker & Kohavi, 1996). From this, we selected the following features: *age, workclass, fnlwgt, education, education-num, marital-status, occupation, relationship, race, sex, capital-gain, capital-loss, hours-per-week, native-country,* and *income.* The target variable, "*target*", was defined as 1 if the "*income*" value was was greater than 50*k*; otherwise, it was 0. Afterward, we removed the "*income*" variable from the data features.

The second dataset was **Productivity** (Garment Worker Productivity) (Imran et al., 2020; 2021). During preprocessing, we first removed the variables "*date*" and "*day*". Next, missing values in the "*wip*" column, the only feature with missing data, were imputed with zeros. Outliers in the incentive column were then eliminated. The target variable, "*target*", was defined as a binary indicator: if the difference between "*actual_productivity*" and "*targeted_productivity*" was greater than or equal to zero, the target was set to 1; otherwise to 0. Finally, we excluded "*actual_productivity*" and "*targeted_productivity*" from data features.

The third and fourth datasets were derived from the Student Performance dataset (Cortez, 2008), specifically the **Portuguese** (Student-por) and **Math** (Student-mat) performance subsets. For both datasets, we defined the target variable, "*pass*", as 1 if the sum of the three grade variables (*G1, G2, G3*) was greater than or equal to 35, and 0 otherwise. After defining the target, we removed the grade variables from the feature set.

**Preparation of datasets for graph generation.** For all datasets, we label-encoded categorical variables, removed duplicate rows, and, when necessary, applied subsampling to ensure a maximum of 500 rows. Each dataset was then divided into data features $\mathcal{X}_{\mathrm{orig}}$ and labels $y$, after which the features were standardized and transformed.

To prepare a given dataset for bipartite graph generation, the feature data was randomly partitioned, with 90% of the samples assigned to the left-hand side (LHS), $\mathcal{X}_{\mathrm{LHS}}$ and 10% to the right-hand side (RHS) $\mathcal{X}_{\mathrm{RHS}}$. Labels of the LHS and RHS samples were directly retrieved from $y$. We then remove all positively labeled samples from the LHS and disregard labels for the remaining samples. We retain all the RHS samples and their labels in the experiments.

### G.1.2   Statistics of the Generated Geometric Bipartite Graphs

For each generated bipartite graph, we report the following statistics: the dataset name (name), number of data features ($\rho$), maximum number of nearest targets in an agent's neighborhood ($k_{\mathrm{max}}$) or threshold for distance between targets and agents in an agents' neighborhood ($\ell$), number of agents ($n$), number of targets positive ($m^+$) and negative ($m^-$) targets, average agent degree (avg.LHS), number of agents with all-positive neighborhoods (only+Ns), all-negative neighborhoods (only-Ns), and empty-neighborhoods (emptyNs), and lastly, the number of positive targets connected to all helpable agents ($uni^+$).

For all experiments under the standard and targeted intervention models, the statistical properties of the bipartite graphs remain as described above. In the learning setting, each bipartite graph is split into training and testing sets. The training set contains 70% of the agents along with their associated edges, while the remaining 30% form the testing set. The targets and their labels are kept constant across both sets. For experiments under the coverage radius model, as described above, each dataset was first split into $\mathcal{X}_{\mathrm{LHS}}$, $\mathcal{X}_{\mathrm{RHS}}$, and $y_{\mathrm{RHS}}$. Then, only positive tar from the RHS were selected and initially assigned a radius of zero, so that no edges exist at the start.

**Table 2:** Statistics of bipartite graphs generated from the **Adult** ($\rho = 14$) dataset. For all graphs, $n = 328$.

| Param | value | $(m^-, m^+)$ | avg.LHS | only+Ns | only-Ns | emptyNs | $uni^+$ |
|---|---|---|---|---|---|---|---|
| $k_{\mathrm{max}}$ | 1 | (36,10) | 1.0 | 70 | 258 | 0 | 0 |
| | 2 | (37,12) | 2.0 | 15 | 184 | 0 | 0 |
| | 3 | (37,12) | 3.0 | 4 | 121 | 0 | 0 |
| | 4 | (37,12) | 4.0 | 2 | 86 | 0 | 0 |
| | 5 | (37,12) | 5.0 | 1 | 61 | 0 | 0 |
| | 6 | (37,12) | 6.0 | 0 | 44 | 0 | 0 |
| | 7 | (37,12) | 7.0 | 0 | 33 | 0 | 0 |
| | 8 | (37,12) | 8.0 | 0 | 19 | 0 | 0 |
| | 9 | (37,12) | 9.0 | 0 | 12 | 0 | 0 |
| | 10 | (37,12) | 10.0 | 0 | 10 | 0 | 0 |
| $\ell$ | 4.0 | (36,12) | 12.70 | 6 | 51 | 35 | 0 |
| | 4.5 | (37,12) | 19.55 | 2 | 33 | 21 | 0 |
| | 5.0 | (37,12) | 27.00 | 2 | 20 | 9 | 0 |
| | 5.5 | (37,12) | 33.47 | 0 | 9 | 6 | 0 |
| | 6.0 | (37,12) | 38.95 | 0 | 6 | 2 | 0 |
| | 6.5 | (37,12) | 42.83 | 1 | 4 | 0 | 0 |
| | 7.0 | (37,12) | 45.70 | 0 | 2 | 0 | 0 |
| | 7.5 | (37,12) | 47.35 | 0 | 0 | 0 | 0 |
| | 8.0 | (37,12) | 48.26 | 0 | 0 | 0 | 2 |
| | 8.5 | (37,12) | 48.70 | 0 | 0 | 0 | 6 |
| | 9.0 | (37,12) | 48.87 | 0 | 0 | 0 | 8 |
| | 9.5 | (37,12) | 48.96 | 0 | 0 | 0 | 10 |
| | 10.0 | (37,12) | 48.99 | 0 | 0 | 0 | 12 |

### G.1.3 The Algorithms and Parameters Used

**The standard model.** For each bipartite graph and every budget $K \in \{1, 5\}$, we computed the social welfare returned by classic greedy $F(S_{\mathrm{g}})$, the heuristic greedy $F(S_{\mathrm{hg}})$, random $F(S_{\mathrm{r}})$ (i.e., $K$ targets are chosen uniformly at random), and random heuristic $F(S_{\mathrm{hr}})$ (i.e., modifies lines 4 and 5 of Algorithm 4 to use random algorithm instead). For reference, we also compute social welfare with no budget constraints $F(S_{\mathrm{full}})$, zero budget $F(S_{\mathrm{o}})$, and the welfare returned by bruteforce search (or $K$-step lookahead greedy algorithm) $F(S^\star)$.

Note that for experiments under the standard model, Algorithm 1 is executed without restriction on information disclosure.

**Table 3:** Statistics of bipartite graphs generated from the **Math** ($\rho = 30$) dataset. For all graphs, $n = 206$.

| Param | value | $(m^-, m^+)$ | avg.LHS | only+Ns | only-Ns | emptyNs | $uni^+$ |
|---|---|---|---|---|---|---|---|
| $k_{\max}$ | 1 | (19, 16) | 1.0 | 106 | 100 | 0 | 0 |
| | 2 | (21, 17) | 2.0 | 60 | 52 | 0 | 0 |
| | 3 | (22, 17) | 3.0 | 21 | 30 | 0 | 0 |
| | 4 | (22, 17) | 4.0 | 13 | 14 | 0 | 0 |
| | 5 | (22, 17) | 5.0 | 5 | 8 | 0 | 0 |
| | 6 | (22, 17) | 6.0 | 5 | 6 | 0 | 0 |
| | 7 | (22, 17) | 7.0 | 1 | 2 | 0 | 0 |
| | 8 | (22, 17) | 8.0 | 0 | 1 | 0 | 0 |
| | 9 | (22, 17) | 9.0 | 0 | 0 | 0 | 0 |
| | 10 | (22, 17) | 10.0 | 0 | 0 | 0 | 0 |
| $\ell$ | 4.0 | (4, 5) | 0.09 | 10 | 5 | 190 | 1 |
| | 4.5 | (9, 7) | 0.21 | 14 | 13 | 174 | 0 |
| | 5.0 | (12, 12) | 0.58 | 25 | 19 | 145 | 0 |
| | 5.5 | (17, 15) | 1.71 | 24 | 34 | 97 | 0 |
| | 6.0 | (21, 17) | 3.63 | 21 | 27 | 61 | 0 |
| | 6.5 | (22, 17) | 7.09 | 8 | 22 | 33 | 0 |
| | 7.0 | (22, 17) | 11.81 | 8 | 23 | 9 | 0 |
| | 7.5 | (22, 17) | 17.55 | 2 | 12 | 3 | 0 |
| | 8.0 | (22, 17) | 23.30 | 0 | 8 | 1 | 0 |
| | 8.5 | (22, 17) | 28.29 | 1 | 3 | 0 | 0 |
| | 9.0 | (22, 17) | 32.38 | 1 | 2 | 0 | 0 |
| | 9.5 | (22, 17) | 35.23 | 0 | 1 | 0 | 0 |
| | 10.0 | (23, 17) | 36.98 | 0 | 1 | 0 | 2 |

**Table 4:** Statistics of bipartite graphs generated from the **Portuguese** ($\rho = 30$) dataset. For all graphs, $n = 224$.

| Param | value | $(m^-, m^+)$ | avg.LHS | only+Ns | only-Ns | emptyNs | $uni^+$ |
|---|---|---|---|---|---|---|---|
| $k_{\max}$ | 1 | (26, 20) | 1.0 | 109 | 115 | 0 | 0 |
| | 2 | (28, 20) | 2.0 | 70 | 74 | 0 | 0 |
| | 3 | (28, 20) | 3.0 | 38 | 36 | 0 | 0 |
| | 4 | (28, 21) | 4.0 | 26 | 22 | 0 | 0 |
| | 5 | (28, 22) | 5.0 | 17 | 13 | 0 | 0 |
| | 6 | (28, 22) | 6.0 | 12 | 8 | 0 | 0 |
| | 7 | (28, 22) | 7.0 | 3 | 6 | 0 | 0 |
| | 8 | (28, 22) | 8.0 | 3 | 2 | 0 | 0 |
| | 9 | (28, 22) | 9.0 | 1 | 1 | 0 | 0 |
| | 10 | (28, 22) | 10.0 | 1 | 0 | 0 | 0 |
| $\ell$ | 4.0 | (1, 5) | 0.04 | 4 | 0 | 219 | 2 |
| | 4.5 | (9, 13) | 0.21 | 18 | 6 | 196 | 0 |
| | 5.0 | (14, 18) | 0.67 | 32 | 8 | 169 | 0 |
| | 5.5 | (19, 20) | 1.50 | 30 | 28 | 129 | 0 |
| | 6.0 | (24, 20) | 3.43 | 26 | 40 | 77 | 0 |
| | 6.5 | (27, 21) | 7.12 | 15 | 29 | 41 | 0 |
| | 7.0 | (28, 21) | 12.69 | 6 | 22 | 20 | 0 |
| | 7.5 | (28, 22) | 19.76 | 4 | 16 | 7 | 0 |
| | 8.0 | (28, 22) | 27.27 | 1 | 7 | 3 | 0 |
| | 8.5 | (28, 22) | 34.06 | 0 | 3 | 1 | 0 |
| | 9.0 | (28, 22) | 39.95 | 0 | 2 | 0 | 0 |
| | 9.5 | (28, 22) | 44.29 | 0 | 0 | 0 | 0 |
| | 10.0 | (28, 22) | 47.06 | 0 | 0 | 0 | 1 |

**Table 5:** Statistics of bipartite graphs generated from the **Productivity** ($\rho = 11$) dataset. For all graphs, $n = 112$.

| Param | value | $(m^-, m^+)$ | avg.LHS | only+Ns | only-Ns | emptyNs | $uni^+$ |
|---|---|---|---|---|---|---|---|
| $k_{\max}$ | 1 | (8, 24) | 1.0 | 77 | 35 | 0 | 0 |
| | 2 | (11, 33) | 2.0 | 62 | 11 | 0 | 0 |
| | 3 | (11, 34) | 3.0 | 52 | 0 | 0 | 0 |
| | 4 | (11, 35) | 4.0 | 28 | 0 | 0 | 0 |
| | 5 | (11, 37) | 5.0 | 16 | 0 | 0 | 0 |
| | 6 | (11, 37) | 6.0 | 9 | 0 | 0 | 0 |
| | 7 | (11, 37) | 7.0 | 4 | 0 | 0 | 0 |
| | 8 | (11, 37) | 8.0 | 3 | 0 | 0 | 0 |
| | 9 | (11, 37) | 9.0 | 3 | 0 | 0 | 0 |
| | 10 | (11, 37) | 10.0 | 2 | 0 | 0 | 0 |
| $\ell$ | 4.0 | (11, 39) | 20.33 | 0 | 1 | 7 | 0 |
| | 4.5 | (11, 39) | 25.60 | 0 | 0 | 6 | 0 |
| | 5.0 | (11, 39) | 32.12 | 0 | 0 | 6 | 0 |
| | 5.5 | (11, 39) | 38.80 | 0 | 0 | 6 | 0 |
| | 6.0 | (11, 39) | 44.24 | 0 | 0 | 5 | 1 |
| | 6.5 | (11, 39) | 46.70 | 0 | 0 | 5 | 13 |
| | 7.0 | (11, 39) | 47.49 | 0 | 0 | 5 | 25 |
| | 7.5 | (11, 39) | 47.73 | 0 | 0 | 5 | 35 |
| | 8.0 | (11, 39) | 47.77 | 0 | 0 | 5 | 39 |
| | 8.5 | (11, 39) | 47.84 | 0 | 0 | 4 | 5 |
| | 9.0 | (11, 39) | 48.05 | 0 | 0 | 4 | 25 |
| | 9.5 | (11, 39) | 48.26 | 0 | 0 | 2 | 0 |
| | 10.0 | (11, 39) | 48.71 | 0 | 0 | 2 | 16 |

**The target interventions model.** For each bipartite graph, and for every target reveal budget $K \in \{1, 5\}$ and targeted intervention budget $B \in \{1, 3\}$, we compare the pre-reveal and post-reveal (Algorithms 6 and 7) intervention gains (Eqns. 9 and 10).

**The coverage radius model.** We computed the number of agents covered using Algorithm 8, for each of the 4 graphs and radius budget $R \in \{4, 4.5, 5, 5.5, 6, 6.5, 7, 7.5, 8, 8.5, 9, 9.5, 10\}$.

**The learning setting.** The learning algorithm proceeds in two main stages. First, we apply the budget ($K$) constrained greedy algorithm (Algorithm 1) to the training set graph, where the revealed target set is the learned hypothesis. Next, we assess the performance of this hypothesis on the testing set graph.

### G.1.4   Performance and Evaluation

**Under the standard, targeted intervention, and coverage models.** In both the standard and targeted intervention models, we compare algorithms by the social welfare they produce. The coverage model, in contrast, measures the number of agents that fall within reach after expanding the coverage radius of a selected set of positive targets.

In the standard model, each algorithm is evaluated by the social welfare $F(S)$ it achieves under different target reveal budgets $K$. In the targeted intervention model, we analyze the difference in pre- and post-reveal intervention gains at different target reveal budgets $K$ and intervention budgets $B$. Below is a definition of the intervention gains.

**Pre-reveal intervention gains:** These are computed as the difference in social welfare returned by the Algorithm 6 ($F(S_{\mathrm{ig}})$ and that from Algorithm 1 ($F(S_{\mathrm{g}})$:

$$\Delta_F(\mathrm{ig}, \mathrm{g}) = F(S_{\mathrm{ig}}) - F(S_{\mathrm{g}}). \tag{9}$$

**Post-reveal intervention gains:** These are computed as the difference in social welfare returned by the Algorithm 7 ($F(S_{\mathrm{gi}})$ and that from Algorithm 1 ($F(S_{\mathrm{g}})$:

$$\Delta_F(\mathrm{gi}, \mathrm{g}) = F(S_{\mathrm{gi}}) - F(S_{\mathrm{g}}). \tag{10}$$

**Under the learning setting.** Let $\mathcal{X}_{tr}$ and $\mathcal{X}_{ts}$ denote the training and testing agent sets. Due to notation simplicity, all performance metrics introduced below are defined over the training set $\mathcal{X}_{tr}$, but the same apply

to the testing set $\mathcal{X}_{ts}$. For each agent $x \in \mathcal{X}_{tr}$, let $N(x) \subseteq \mathcal{T}$ denote the set of targets in its neighborhood and define $\delta_x^+ = |\{t \in N(x) : f(t) = +1\}|$ and $\delta_x^- = |\{t \in N(x) : f(t) = -1\}|$ as the number of positive and negative targets in that neighborhood, respectively. To evaluate performance, we consider three kinds of agent subsets.

$$\mathcal{X}_{tr}^{(1)} = \{x \in \mathcal{X}_{tr} : \delta_x^+ > 0\}, \quad \mathcal{X}_{tr}^{(2)} = \mathcal{X}_{tr}, \quad \mathcal{X}_{tr}^{(3)} = \{x \in \mathcal{X}_{tr} : \delta_x^+ > 0 \text{ and } \delta_x^- > 0\}.$$

The first agent set $\mathcal{X}_{tr}^{(1)}$ consists of agents with at least one positive target in their neighborhood, the second $\mathcal{X}_{tr}^{(2)}$ includes all the agents $\mathcal{X}_{tr}$, and the third $\mathcal{X}_{tr}^{(3)}$ includes only *helpable agents* (those with both positive and negative target neighbors). Let $S_{tr} \subseteq \mathcal{T}$ with $|S_{tr}| \leq K$ be the target set revealed by the classic greedy algorithm when run on the train graph $\mathcal{G}_{tr} = (\mathcal{X}_{tr} \cup \mathcal{T}, E)$ at a budget of $K$, and let $F(S_{tr})$ denote the resulting social welfare. The performance measures are defined as

$$\text{Perf}_i = \frac{F(S_{tr})}{|\mathcal{X}_{tr}^{(i)}|} \times 100\%, \ i \in \{1,2\}, \quad \text{Perf}_3 = \frac{F(S_{tr}) - |\{x \in \mathcal{X}_{tr} : \delta_x^+ \geq 1 \text{ and } \delta_x^- = 0\}|}{|\mathcal{X}_{tr}^{(3)}|} \times 100\%.$$

A score of 100 with respect to $\text{Perf}_1$ indicates success on agents with at least one positive target neighbor, excluding agents with empty or all-negative neighborhoods. For $\text{Perf}_2$, a score of 100 indicates success on all helpable agents, including all sampled agents, and a score of 100 in $\text{Perf}_3$ indicates success on all helpable agents, excluding unhelpable ones. Theoretical results use $\text{Perf}_2$, while empirical analysis considers them all.

## G.2 Empirical Results under the Standard Model

For all single-group results under the standard model, $F(S_{\text{full}})$ represents the social welfare without any budget constraints, while $F(S_o)$ corresponds to the social welfare when the budget is zero, and no targets are revealed. Under budget constraints, $F(S_r)$ denotes the social welfare obtained by randomly revealing targets, $F(S_g)$ corresponds that of Algorithm 1, $F(S_{hr})$ and $F(S_{hg})$ represent the social welfare achieved by the heuristic random and heuristic greedy strategies, respectively, and $F(S^*)$ denotes the optimal social welfare computed via bruteforce search.

**General observations.** For all $k$NN generated graphs, when agents have atmost one target neighbor that is either positive or negative (Tables 2–5, $k_{\max=1}$), revealing targets is unnecessary (Figures 9–11 subfigures (**a**–**d**)).

In sparse graphs, particularly when many agents have empty neighborhoods (Tables 3 and 4 where emptyNs $\geq 1$), overall social welfare remains close to zero regardless of the algorithm or budget (Figures 9f,g and 10f,g).

As connectivity increases, especially in threshold-based graphs constructed with larger threshold values, resulting social welfare increases (e.g., Figures 9e,h) because of presence of positive target that are connected to all helpable agents (Tables 2 and 5, $\ell \geq 8$ and $uni^+ \geq 1$).

**Social welfare comparison: Algorithm 1 vs. random selection ($F(S_g)$ vs. $F(S_r)$).** In $k$NN generated graphs, the higher $k_{\max}$ is, the higher the social welfare achieved by the classic greedy algorithm, even at low budget levels (Figures 9a–d). Although higher budgets generally lead to greater social welfare, particularly when using the classic greedy algorithm (Figures 9a–d), and occasionally for the random algorithm as well (Figures 9b,d), in some instances, such as in Figure 9a, the budget appears to have little influence on the random algorithm's performance. Overall, for the same budget $K$, the Algorithm 1 consistently attains social welfare that is at least that achieved by the random algorithm (Figures 9a–d). But, when the random algorithm operates at a higher budget, it can occasionally result in higher social welfare than Algorithm 1 at a lower budget (Figures 9c–d). However, random selection generally performs poorly, and revealing additional targets often yields little to no increase in social welfare. E.g., see Figure 9a for all values of $k_{\max}$ and $K$, and Figure 9c when $k_{\max} \geq 5$ and $K = 1$.

In threshold-based graphs, particularly those constructed with larger threshold values ($\ell$), the advantage of the greedy algorithm over the random baseline becomes more pronounced due to increased connectivity. Even with limited budgets, the greedy approach often attains near-maximal social welfare (Figures 9e–h).

In contrast, the performance of the random algorithm varies substantially. In graphs where most targets are negative (Table 2), random selection yields very low social welfare (Figures 9a,e). However, in graphs with a relatively large number of positive targets connected to nearly all helpable agents, where the probability of revealing one at random is higher (Tables 3–5), the random algorithm performs considerably better (Figures 9f–h).

**Social welfare comparison: heuristic greedy vs. heuristic random ($F(S_{\text{hg}})$ vs. $F(S_{\text{hr}})$).** Under low connectivity, particularly in graphs generated using the $k$NN method, the heuristic random algorithm still achieves a high social welfare but in some cases remains significantly below the heuristic greedy algorithm at a similar budget level (Figures 10a–d). In settings where random algorithm produced low social welfare (Figures 9a,c), the heuristic random algorithm performs markedly better (Figure 10a,c). This improvement arises because the selection becomes localized, and randomly choosing among positive targets is more effective than selecting from all targets. Performance further improves with higher connectivity: the social welfare returned by the heuristic random and heuristic greedy become nearly identical, even with graphs generated with a low threshold value (Figures 10f–h).

**Social welfare comparison: bruteforce search vs. Algorithm 1 vs. heuristic greedy ($F(S^\star)$ vs. $F(S_{\text{g}})$ vs. $F(S_{\text{hg}})$).** Across bipartite graphs constructed via thresholding or $k$NN, bruteforce search, Algorithm 1, and heuristic greedy achieve comparable social welfare across all budgets and datasets (Figure 11). This similarity arises because the optimal target sets mainly consist of positive targets, leading all algorithms to converge on nearly identical solutions at similar budgets. These findings indicate that greedy approaches may still perform well in practice when there are no information disclosure restrictions, likely because graphs based on real-world tend to be well connected and balanced.

### G.3 Empirical Results for Fairness

In the main paper and in Figures 12 and 13, we compared the average social welfare (gain) (i.e., the total group welfare (gain) divided by the number of agents in the group) achieved by the classic greedy algorithm

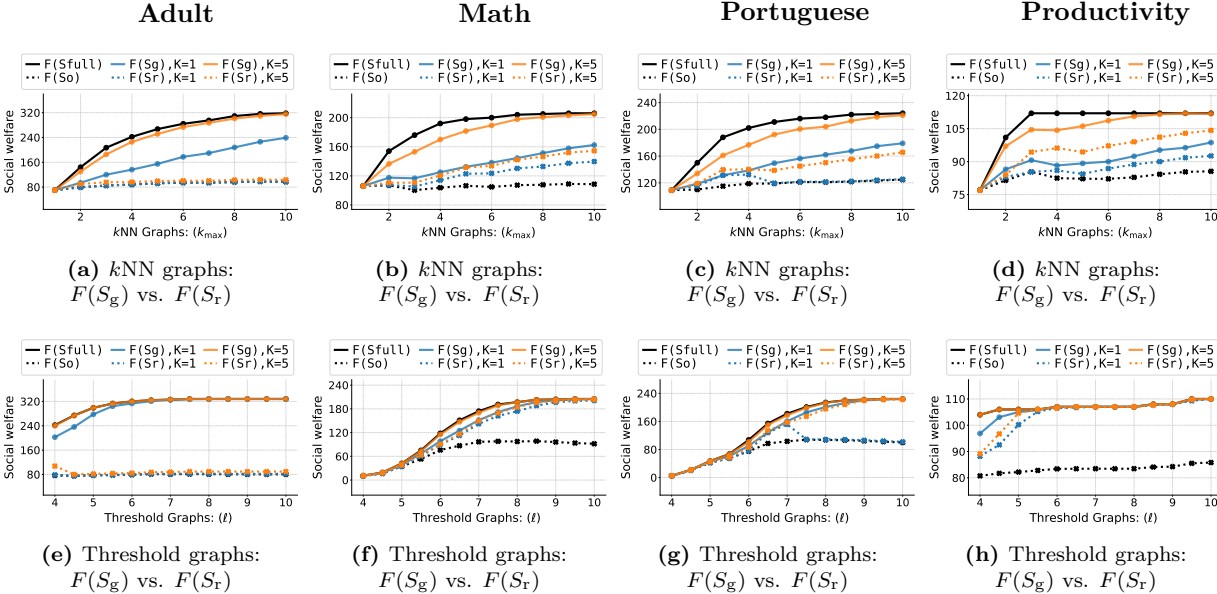

**Figure 9:** Comparative analysis of the social welfare generated from running the random $F(S_{\text{r}})$ and the classic greedy $F(S_{\text{g}})$ algorithms across $k$NN graphs (a–d) and the threshold graphs (e–h) from the 4 datasets (Tables 2–5). Except on the Adult dataset, when the graphs are nearly complete, budgets $K = \{1, 5\}$ have a low effect and social welfare returned by both algorithms is almost equal (f–h). In contrast, in sparser settings, at similar budget levels, $F(S_{\text{g}})$ is consistently higher than $F(S_{\text{r}})$ (a–d).

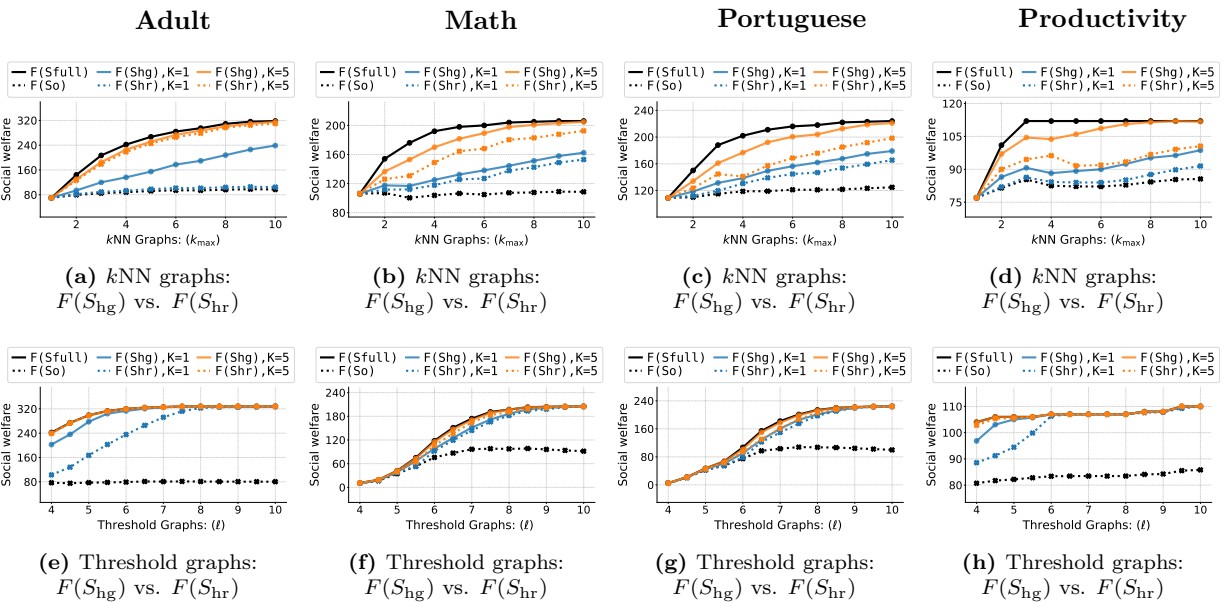

**Figure 10:** Comparative analysis of the social welfare generated from running the heuristic random $F(S_{hr})$ and heuristic greedy $F(S_{hg})$ algorithms across $k$NN graphs (a–d) and the threshold graphs (e–h) from the 4 datasets (Tables 2–5). In general, when the graphs are nearly complete, across budget levels $K = \{1, 5\}$, social welfare returned by both heuristic algorithms is almost equal (e–h). In sparser settings, at similar budget levels, $F(S_{hg})$ is consistently higher than $F(S_{hr})$ (a–d).

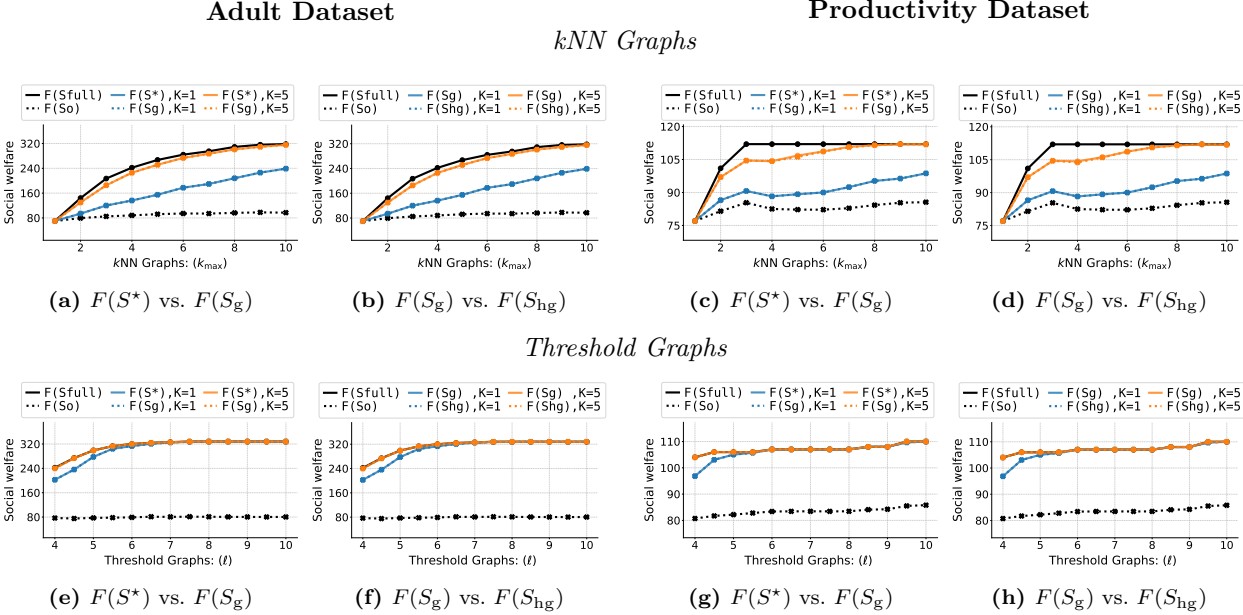

**Figure 11:** Comparison of social welfare obtained via brute-force $F(S^\star)$, classic greedy $F(S_g)$, and heuristic greedy $F(S_{hg})$ on the Adult and Productivity datasets. Top row are $k$NN graphs and bottom row are threshold graphs. Across target reveal budgets $K = \{1, 5\}$, in both datasets and graphs, both greedy variants get exact solutions ($F(S^\star)$). As connectivity increases (cf. Tables 2 and 5), both greedy variants closely approximate the optimum $F(S_{full})$ (Figures 11e–h where $\ell > 6$). Although we show only the Adult and Productivity datasets, we observe similar patterns on the Math and Portuguese datasets.

when applied to the full graph and when applied separately to the male and female bipartite graphs derived from the Adult, Math, and Portuguese datasets. Here, we assess whether the group-prioritized greedy variant described below improves average group welfare and reduces inter-group disparities.

The group-prioritized classic greedy approach proceeds as follows. At each Algorithm 1 iteration, when multiple targets yield the same total marginal gain in social welfare, ties are resolved by selecting the target that maximizes the marginal gain for the prioritized group. For example, consider three targets with identical total marginal gains of 100, but varied gains for the groups. Target $t_1$ yields 50 for the male group and 50 for the female group, $t_2$ yields 51 for the male group and 49 for the female group, and $t_3$ yields 100 for the male group and 0 for the female group. If the female group is prioritized, $t_1$ is selected, and if the male group is prioritized, $t_3$ is selected.

**Group-prioritized classic greedy approach experimental results.** Overall, the group-prioritized greedy variant rarely improves the average group welfare relative to the classic greedy algorithm, except in a few cases highlighted below.

Prioritizing a group can increase its average group welfare at the expense of the other group. On the Math $k$NN graph generated with $k_{\max} = 2$, when classic greedy is run on the graph with a budget of $K = 6$, it achieves an average welfare of 0.6822 and 0.6705 for the female and male groups, respectively. Prioritizing the male group raises their average welfare to 0.6818 and lowers that of the female group (0.6737). Similarly, on the Portuguese $k$NN graph generated with $k_{\max} = 2$, when classic greedy is run on the graph with a budget of $K = 6$, it attains an average welfare of 0.6207 and 0.5926 for the female and male groups, respectively. Prioritizing the female group increases their average welfare to 0.6379 and reduces that of the male group (0.5787) while prioritizing the male group raises their average welfare to 0.5972 and lowers that of the female group (0.6164).

In other cases, prioritization harms the targeted group while benefiting the other. On the Portuguese $k$NN graph generated with $k_{\max} = 4$, when classic greedy is run on the graph with a budget of $K = 6$, it attains an average welfare of 0.8039 and 0.8171 for the female and male groups, respectively. Prioritizing the male

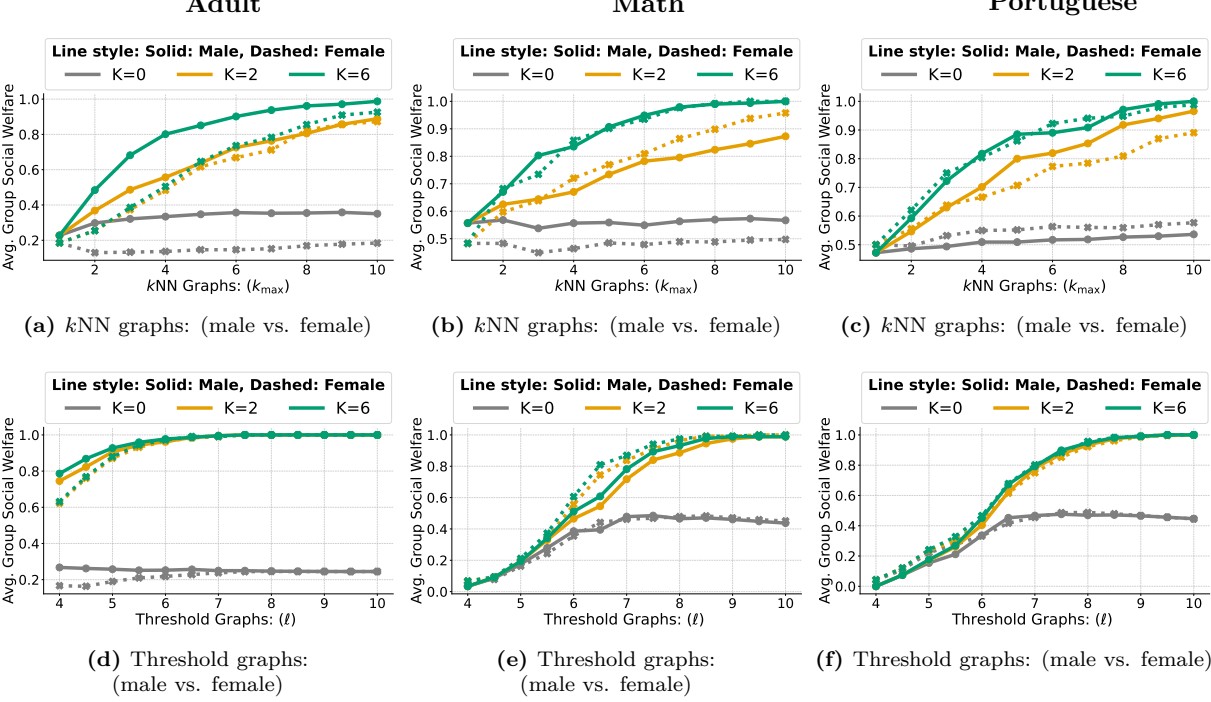

**Figure 12:** Comparative analysis of the average group social welfare generated from running the classic greedy approach across $k$NN graphs (a–c) and the threshold graphs (d–f) from the 3 datasets (Tables 2–4) on the **whole graph**. When the graph connectivity is high, there is no disparity in social welfare generated for the 2 groups.

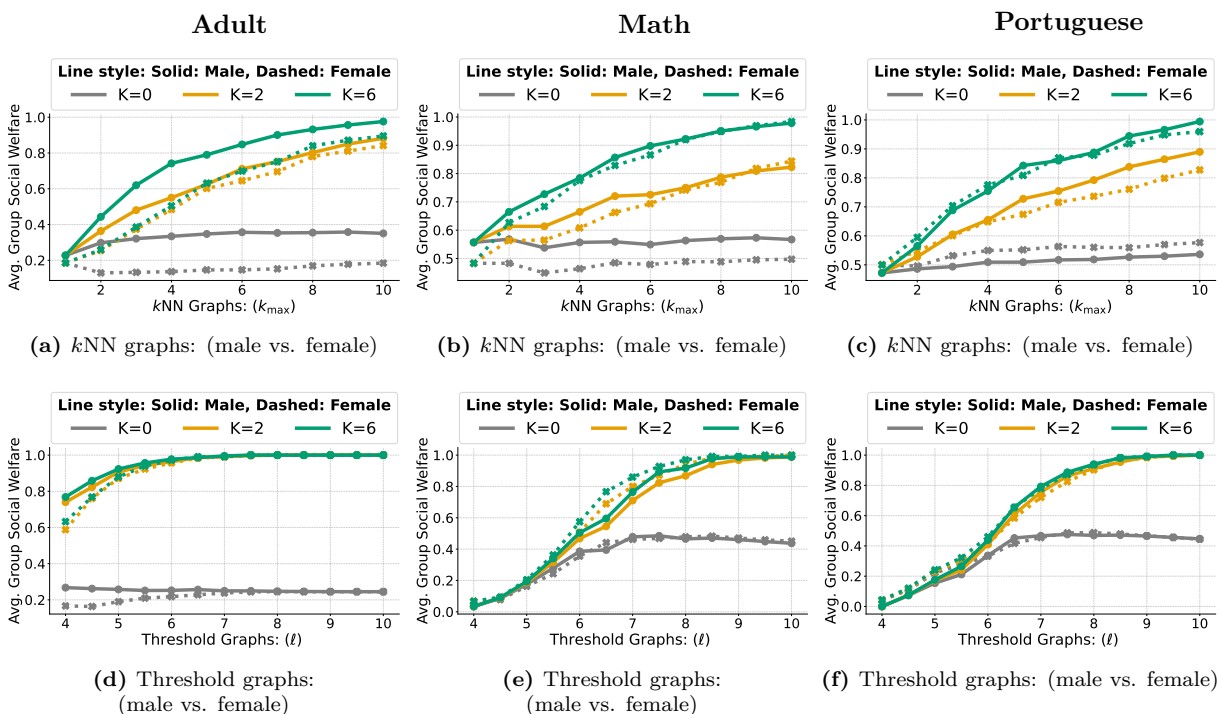

**Figure 13:** Comparative analysis of the average group social welfare generated from running the classic greedy approach across $k$NN graphs (a–c) and the threshold graphs (d–f) from the 3 datasets (Tables 2–4) **separately on each group**. That is, when we run greedy at a budget $K/2$ solely on each group's graph. When the groups' graph connectivity is high, there is no disparity in social welfare generated for the 2 groups.

groups reduces their average welfare to 0.8009 and increases that of the female group (0.8211). Similarly, on the Math $k$NN graph generated with $k_{max} = 2$, when classic greedy is run on the graph with a budget of $K = 6$, prioritizing the female group lowers their average welfare to 0.6737 and raises the male group's average welfare to 0.6761.

These findings show that group prioritization does not consistently improve outcomes over classic greedy and may introduce welfare trade-offs without a clear overall benefit.

### G.4 Empirical Results for Targeted Interventions

In this section, we compare pre- and post-reveal intervention gains and examine the effects of varying the intervention and target reveal budgets $(K, B)$.

**General observations.** Intervention gains are generally upper bounded by the intervention budget $B$ (Figures 14 and 15).

Smaller target reveal budgets $K$ tend to produce higher gains, as most agents may still have low probabilities of emulating a positive target, even after welfare-maximizing subset of targets is revealed. In Figures 14 and 15, intervening when greedy was run with a budget of $K = 1$ (subfigures (**a,b,e,f**)) yields gains at least as large as the those when run with $K = 5$ (subfigures (**c,d,g,h**)).

Intervention gains are highest when multiple agents have empty or all-negative neighborhoods. For instance, in Adult $k$NN-generated graphs, the number of agents with all-negative neighborhoods exceeds the intervention budget, that is, for each graph $i \in [10]$, only-Ns$^{(i)} > B = 4$ (Table 2). Consequently, the pre- and post-reveal intervention gains are capped by $B$ (Figures 14a,c).

Intervention may be redundant or underutilized when many agents have all-positive neighborhoods, very few have all-negative or empty neighborhoods and the label reveal algorithm achieves an optimal solution.

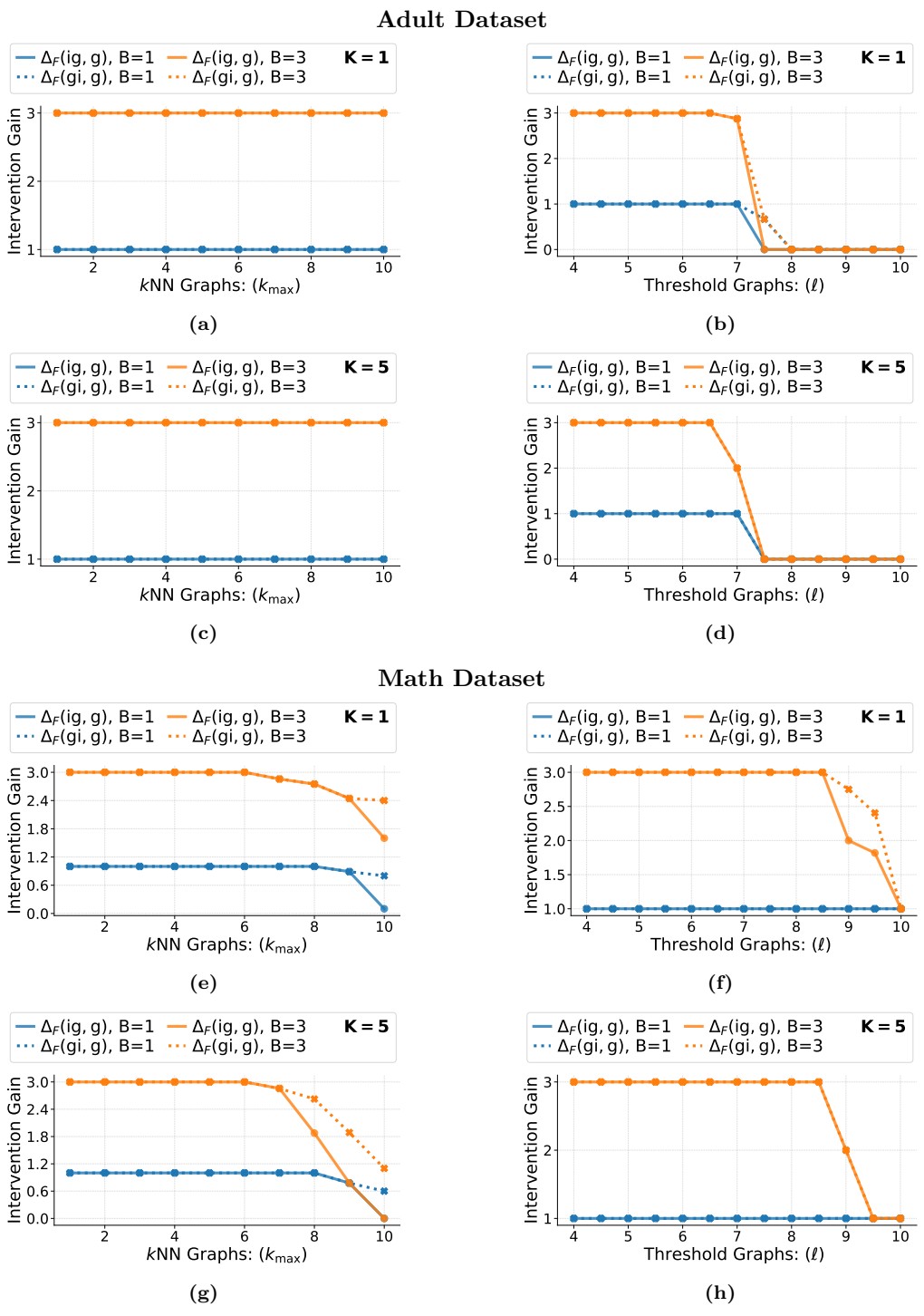

**Figure 14:** Pre- and post-reveal intervention gains ($\Delta_F(ig, g)$ and $\Delta_F(gi, g)$) across datasets, target reveal budgets $K \in \{1, 5\}$, intervention budgets $B \in \{1, 3\}$, and graph generation methods ($k$NN and threshold) (Tables 2 and 3). For both datasets, lower $K$ often yields larger intervention gains. A high $B$ can sometimes be redundant when agents have all-positive neighborhoods, and/or very few agents have empty or all-negative neighborhoods and greedy is optimal (b, d, e, g). When the number of agents with all-negative neighborhoods is at least $B$, both pre- and post-reveal interventions gains $\Delta_F(ig, g)$ and $\Delta_F(gi, g)$ are atmost $B$ (a, c).

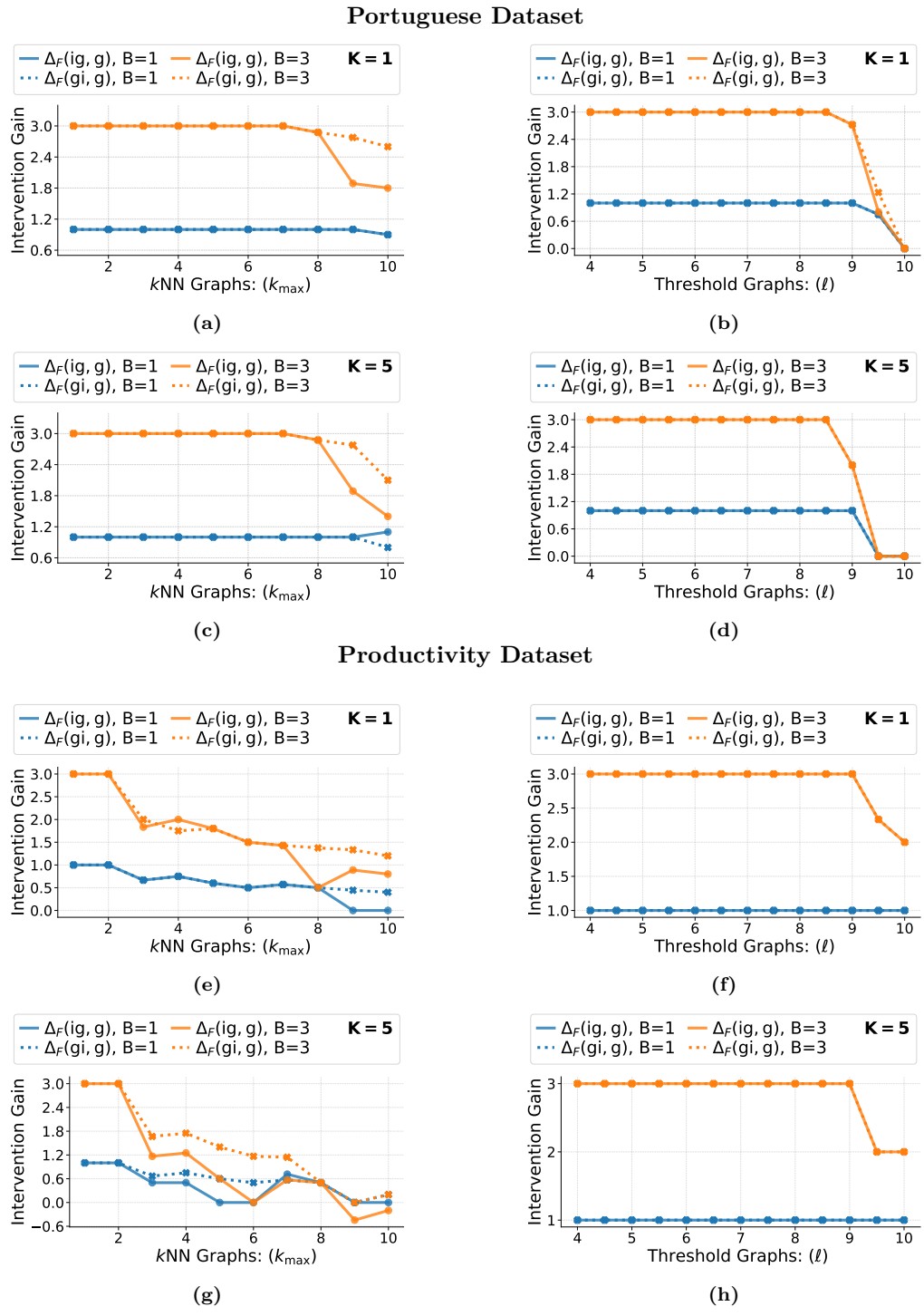

**Figure 15:** Pre- and post-reveal intervention gains ($\Delta_F(ig, g)$ and $\Delta_F(gi, g)$) across datasets, target reveal budgets $K \in \{1, 5\}$, intervention budgets $B \in \{1, 3\}$, and graph generation methods ($k$NN and threshold) (Tables 4 and 5). For both datasets, lower $K$ often yields larger gains. A high $B$ can be redundant when many agents have all-positive neighborhoods (e, b, d). Pre-reveal intervention can yield negative gains (Figure g, $k_{\max} = \{9, 10\}$) but may also sometimes outperform post-reveal intervention (Figure 15c, $k_{\max} = 10$).

For instance, in Productivity threshold generated graphs, greedy is optimal (Table 5; Figure 11g), and intervention applies only to a shrinking set of agents with no neighbors (Figures 15f,h). A similar pattern

is observed in Adult, Math, and Portuguese threshold-generated graphs (Tables 2–4; Figures 9e–g), where the intervention budget is underutilized/redundant (Figures 14 and 15, subfigures (**b,d,f,h**)). In addition, intervention gains are generally small when high-risk agents already have a high probability of emulating a positive target (e.g., in Figures 15e,g).

**Comparison of pre- and post-reveal interventions** Post-reveal interventions consistently produce positive intervention gains, which are most times at least as large as those from pre-reveal intervention (Figures 14 and 15).

Pre-reveal intervention can occasionally lead to negative intervention gains (Figure 15g). If the label reveal algorithm (Algorithm 1) is effective and high-risk agents already have high probabilities of emulating positive targets, removing them before executing Algorithm 1 might distort the graph and lead to the algorithm selecting a target set with lower social welfare than if those agents had remained, resulting in a negative intervention gain $\Delta_F(ig, g)$.

### G.5 Empirical Results under the Coverage Radius Model

For all geometric graphs generated with zero initial target radius $r_i = 0$ for all targets $i$, increasing the radius budget increases the number of agents reached by positive targets (Figure 16). When many agents have several positive targets within a comparable radius, an additional radius provides little to no gain (e.g., on Adult dataset, at $R \geq 8$). When distances to the nearest positive target vary substantially, larger radius budgets lead to broader coverage (e.g., on Math, Portuguese, and Productivity datasets). When agents are densely grouped at roughly the same large distance from positive targets, minor increases in radius produce little change while more substantial expansions broaden coverage, as seen in the Productivity dataset.

### G.6 Empirical Results under the Learning Setting

We analyze the empirical results for the learning setting. Training and testing scores are averaged over 100 independent train-test splits, each constructed with a different random seed. Performance is evaluated using three metrics, $\text{Perf}_1, \text{Perf}_2, \text{Perf}_3 \in [0, 100]$, defined in Appendix G.1.4.

Across $k$NN and threshold graphs for all datasets, training performance is consistently at least as high as testing performance at comparable budget levels (Figures 17 and 18).

For $k$NN graphs in which each agent has at most one neighbor that is positive or negative (Tables 2–5, $k_{\max} = 1$), the performance scores are structure-dependent since there are no revealed targets. Here, $\text{Perf}_1 = 100$ since all agents with atleast one positive target neighbor are catered to, $\text{Perf}_2$ equals the fraction of agents connected to positive targets, and $\text{Perf}_3 = 0$ because no agents are helpable (Figures 17 and 18, subfigures (**a–c, g–i**)).

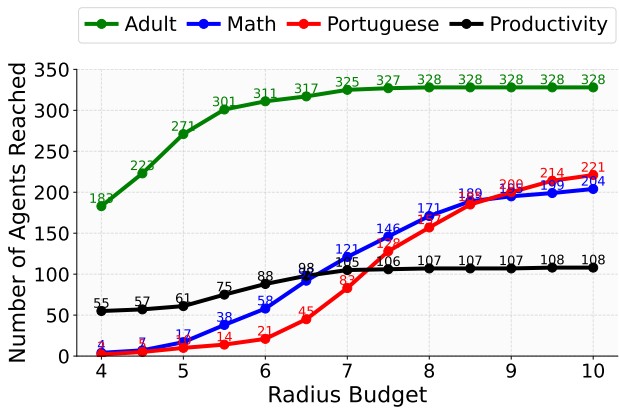

**Figure 16:** Performance of the greedy coverage radius algorithm (Algorithm 8) on the Adult, Math, Portuguese, and Productivity datasets. Number of agents reached increases with the radius budget.

For threshold graphs, particularly those generated with higher thresholds, increased connectivity (Tables 2–5 where $\ell \geq 6.5$) leads training and testing performance to converge to the same score, and the budget levels become less impactful (Figures 17 and 18, subfigures (**d–f, j–l**)). These findings are attributed to an increase in the number of positive targets connected to all helpable agents.

When many agents have empty or all-negative target neighborhoods, $\text{Perf}_2$ is more affected than the other metrics, since those agents are excluded from the evaluation in $\text{Perf}_1$ and $\text{Perf}_3$. For example, in the Productivity threshold graphs (Table 5), even when there is a high number of positive targets connected to all helpable agents, $\text{Perf}_2$ remains well below 100 because some agents are connected exclusively to negative targets (Figure 18k).

Finally, the three metrics can differ markedly within the same graph, particularly when only one agent has both positive and negative targets in its neighborhood. For example, in the Portuguese $\ell = 4$ threshold graph, of 224 agents, 4 have only positive neighbors, 219 only negative neighbors, and 1 has both (Table 4, $\ell = 4.0$). Likewise, in the Math $\ell = 4$ threshold graph, among 206 agents, 10 have only positive neighbors, 5 only negative neighbors, 190 have no neighbors, and 1 has both (Table 3, $\ell = 4.0$). If this single mixed-neighborhood agent appears in the training set, Algorithm 1 can reveal a target that ensures that the agent emulates a positive target with probability one. If it appears in the test set instead, the algorithm reveals no targets during training, yielding zero social welfare for that agent at test time. As a result, $\text{Perf}_1$ may reach 100% on the training set yet be very low on the test set. Overall, $\text{Perf}_1$ is reduced by the large number of agents who cannot be helped by Algorithm 1, whereas $\text{Perf}_3$ captures the average probability of helping the mixed agent across splits (Figures 17j–l and 18d–f).

## H  Limitations and Future Works

In addition to the recommendations outlined in the Section 6, we note several limitations and identify opportunities for future work below.

Currently, we focus on how the proxy social welfare function can help achieve a constant-factor approximation to the true social welfare when the revealed set can include negative targets. An interesting direction for future work is to further investigate our observation that an equal amount of proxy social welfare does not necessarily lead to similar emulation choices, and to examine the implications of this divergence for modeling fairness when agents are partitioned into distinct groups. Additionally, further experiments could investigate the sensitivity of the greedy approach to parameter $c$, and also analyze how the divergence between the proxy and true welfare (through parameter $c$) is reflected in the performance of the greedy algorithm.

We consider a setting in which the social planner has complete knowledge of the graph, or observes an agent's neighborhood upon sampling, when the left side of the graph is replaced with a probability distribution over agents. Future work could explore learning in scenarios where the social planner only has access to partial information. In addition, as captured by the bipartite graph structure, we assume that there are no interactions among agents or among targets. Future work could relax this assumption by examining more general graph structures that allow for richer forms of interaction.

While our experiments are extensive, covering four datasets, multiple greedy variants, and 23 generated bipartite graphs per dataset, a viable next step is to evaluate our models on real-world bipartite networks. For instance, one could examine the performance of the greedy approaches on the social network of mentor-mentee relationships drawn from academic genealogy datasets (David, 2024).

Lastly, our work mainly focuses on a setting where role models or options are classified as either positive or negative. However, the core structure of our model naturally extends to settings with multiple types. For example, if "good" is defined relative to a threshold, a social planner could direct agents toward role models that exceed this benchmark, and during emulation, agents avoid those revealed to be below the desired threshold. For future work, it would be interesting to study the settings in which the role model or option quality is continuous and or can be ranked.

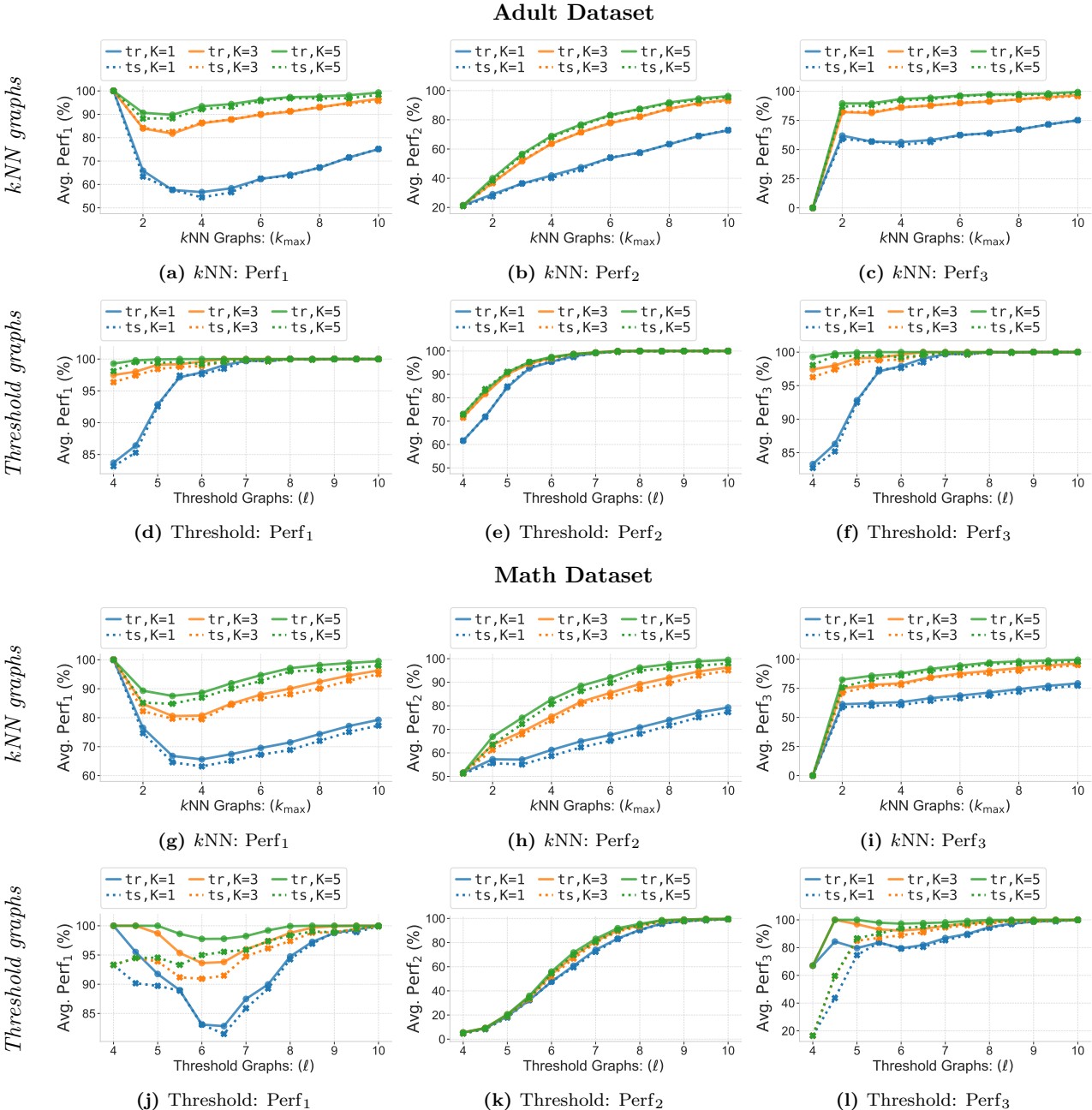

**Figure 17:** Performance of Algorithm 1 in the learning setting on the **Adult** and **Math** datasets under three metrics ($\text{Perf}_1, \text{Perf}_2, \text{Perf}_3$) for $k$NN and threshold generated graphs (Tables 2 and 3). Across both datasets, larger budget $K$ lead to weakly higher performance. Increasing graph connectivity raises overall scores while reducing the performance gap between budgets (e.g., in Figures 17d–f). Compared to Figures 17j and 17l, the performance scores cluster more tightly across budgets in Figure 17k because unlike them, denominator in $\text{Perf}_2$ includes all sampled agents.

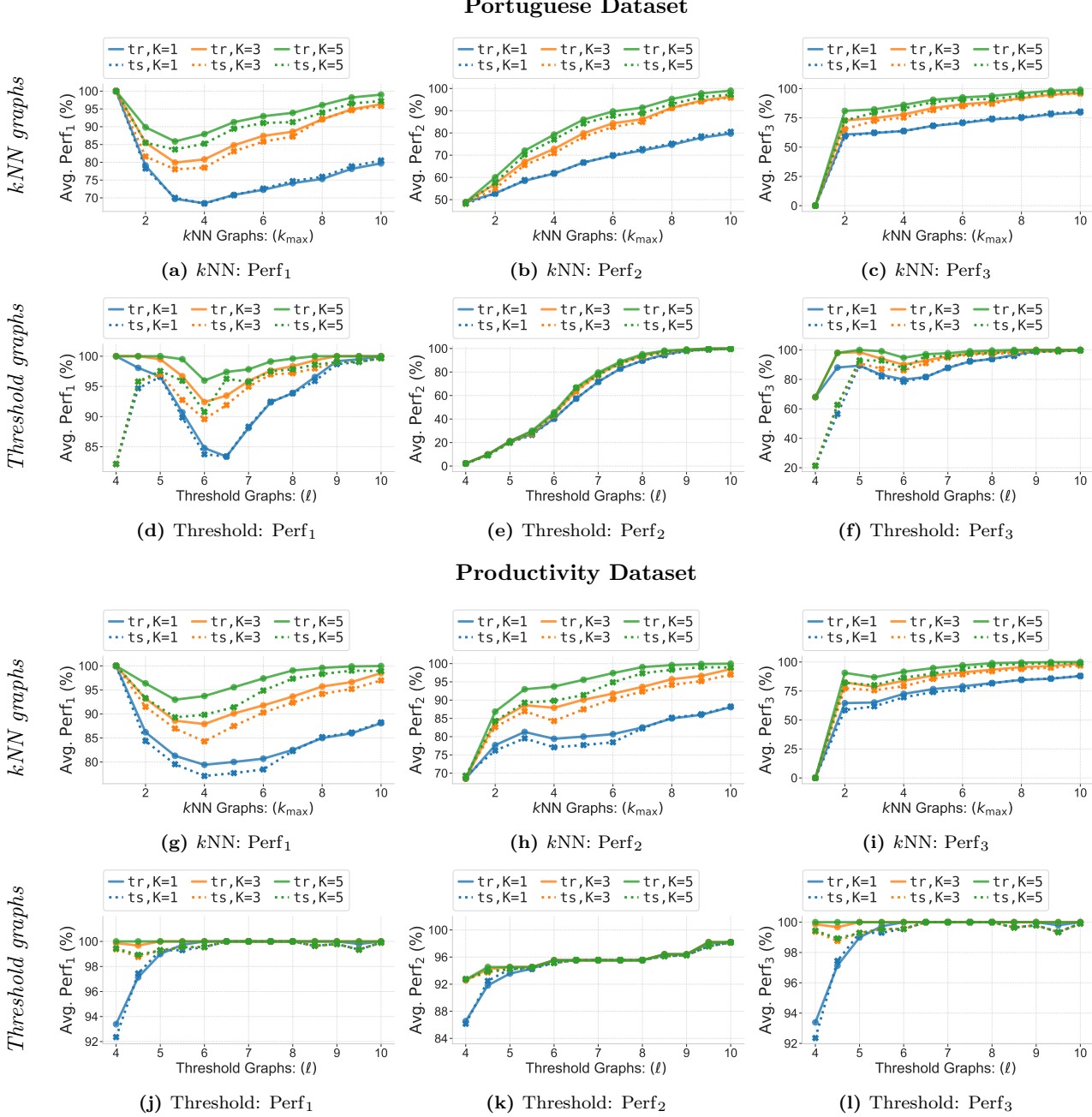

**Figure 18:** Performance of Algorithm 1 in the learning setting on the **Portuguese** and **Productivity** datasets under three metrics ($\mathrm{Perf}_1, \mathrm{Perf}_2, \mathrm{Perf}_3$) for $k$NN and threshold generated graphs (Tables 4 and 5). Across both datasets, larger budget $K$ lead to weakly higher performance. Increasing graph connectivity raises overall scores while reducing the performance gap between budgets (e.g., in Figures 18j–l). Because of presence of unhelpable agents in the graphs (Table 5), $\mathrm{Perf}_2$ can remain below 100 even when the algorithm is optimal (cf. Figure 18k). Lastly, because the Math and Portuguese datasets were curated in a similar manner Cortez (2008), learning setting results on the generated graphs are closely similar.

