# OpenReview forum: "Revealing Positive and Negative Role Models to Help People Make Good Decisions"
_TMLR — Decision pending for TMLR_

### Review · Reviewer_rqzd · 2026-04-29

**Summary Of Contributions:**

The article studies a bipartite problem that maps agents and actions in which a planner knows which targets are positive or negative and can spend a limited budget to reveal some labels so that agents are more likely to choose positive actions. Agents choose uniformly among positive actions if they exist, or among unrevealed neighbors.

The main technical observation is that the welfare function is monotone, and submodular only when only positive targets can be revealed. To handle this case, the authors define a proxy welfare function that is submodular, and then prove that under a suitable c-boundness assumption, this proxy leads to a multiplicative approximation of the optimal solution.

**Audience:**

Yes

**Audience Explanation:**

The paper could be classified into discrete optimization, which has a non-empty intersection with TMLR's audience. I, however, believe that "learning" is not at the core of the article, despite the section 4.3 of the article that presents how to adapt the proposed method to a statistical setting, but which is in essence simply a concentration to the average by Hoeffding's inequality.

**Claims And Evidence:**

Yes

**Claims Explanation:**

In my opinion, the claims made in the article are supported by accurate, convincing, and clear evidence. To begin, the paper is particularly well written, which helps in understanding the claims and the proofs. Then, all the details presented in the article are rigorously proven, and I did not spot any errors in the proofs. Finally, while only supported by semi-synthetic experiments, I did not spot any methodological errors in the experimental section.

**Requested Changes:**

As a note for the review process, I would rate my expertise to review this paper as quite low, as the paper is quite far from my usual areas of research. I classify all my requested changes as "recommendation" rather than critical.

1. The model assumptions are quite strong: the planner knows all target labels, agents follow a one-step uniform choice rule, and agent-to-agent as well as target-to-target interactions are ignored. Can the authors detail further, whenever introduced, if such assumptions are realistic or not?

2. The assumption of $c$-boundness appears quite strong, and I think that it would improve the paper to discuss it further, for instance by adding empirical evidence for realistic values of $c$.

3. We can imagine scenarios where the outcome is not + or - 1, but has some sort of continuous distribution. Is it possible that the presented work extends to this scenario? And if so, it would be interesting to discuss it.

4. I think that the section 3.3 could benefit from a better development on what the fairness measures are, and how they are related to the rest of the paper.

---

> ### Author Response · Authors · 2026-05-01
> **Thank you so much, reviewer rqzd, for the quick and positive review of our work!**
>
> Thank you so much, reviewer rqzd, for the quick and positive review of our work!
> Below, we address the points raised under “Requested Changes.” We have also uploaded a revised version of the manuscript with changes highlighted in blue in response to the reviewer’s suggestions.
> To keep this response concise, we generally avoid repeating the revised text in full here, and instead indicate the edited sections and paragraphs.
>
> > I think that section 3.3 could benefit from a better development on what the fairness measures are, and how they are related to the rest of the paper.
>
> Thank you for pointing this out!  We will rename the section title from “Fairness”  to “Simultaneous Approximate Optimality”.  In addition, we will add a sentence at the beginning of the section saying the following. “In this section, we establish fairness guarantees by demonstrating the existence of a solution set that is simultaneously approximately optimal for all groups, and examine how the disclosure policy impacts this per-group approximation guarantee.”
> We will also slightly change the sentence: “A revealed target set satisfies all groups if every group” to “A revealed target set is simultaneously approximately optimal for all groups (i.e., satisfies all groups), if every group …
>
> > The model assumptions are quite strong: the planner knows all target labels, agents follow a one-step uniform choice rule, and agent-to-agent as well as target-to-target interactions are ignored. Can the authors detail further, whenever introduced, if such assumptions are realistic or not?
>
> While our assumptions are quite strong, they do capture some realistic aspects of how agents make decisions, and can be generalized to other settings. For instance, in the tax filing example, taxpayers tend to follow a simple decision process: they refer to the revealed target set (e.g., booklet from tax body), avoid options explicitly labeled as non-compliant, select a compliant strategy when one is available, and otherwise choose among the remaining options, often with no interaction with other taxpayers.
> That said, changes to these assumptions we make might introduce new interesting directions for future work. We added this paragraph to Appendix H. “We consider a setting in which the social planner has complete knowledge of the graph, or at least observes an agent’s neighborhood upon sampling, when the left side of the graph is replaced with a probability distribution over agents.  Future work could explore learning in scenarios where the social planner only has access to partial information. In addition, as captured by the bipartite graph structure, we assume that there are no interactions among agents or among targets. Future work could relax this assumption by examining more general graph structures that allow for richer forms of interaction.”
>
> > The assumption of c-boundness appears quite strong, and I think that it would improve the paper to discuss it further, for instance by adding empirical evidence for realistic values of c.
>
> The reviewer raises a good point, and we agree that the c-boundness assumption is quite strong in general, and in principle, it would be valuable to further explore empirical estimates of c in various scenarios. That said, we do want to mention, however, that for the experiments where graphs are generated via the kNN approach, the parameter k_{\max} directly controls the number of negative neighbors an agent may have. In that sense, it naturally induces a setting consistent with the c-bound assumption, where effectively c \leq k_{\max}. For instance, when k_{\max} = 1, this corresponds to c = 1, and the results observed in such settings apply. We would also like to note that our current experiments are already quite extensive across datasets, graph generation processes, and intervention settings; therefore, adding a separate investigation of the impact of the c might make it a bit difficult to more easily highlight the main comparisons. We think a thorough analysis of the impact of various c values would be interesting for future work. We will add a discussion of this point to Appendix H.

---

> > ### Author Response · Authors · 2026-05-01
> >
> > > We can imagine scenarios where the outcome is not + or - 1, but has some sort of continuous distribution. Is it possible that the presented work extends to this scenario? And if so, it would be interesting to discuss it.
> >
> > The reviewer raises a good point and a natural extension to consider. We believe that if there are multiple kinds of role models / options, instead of only considering positive and negative, then most of the model setup would still work. For instance, in cases where there is a threshold for “good”, then the social planner could still guide agents towards targets that are above the threshold, and agents would avoid those revealed to be below the desired threshold.
> > In a setting where there are varying levels of goodness, one could discretize levels of quality, which would require corresponding refinements to both the agents’ emulation process and the welfare objective, but would not fundamentally alter the model’s logic. Overall, while we believe that our current setup yields significant theoretical and empirical results, moving beyond binary labeled role models presents an interesting direction for future work. We will include the reviewers’ observations and the insights above in Appendix H.

---

> > > ### Author Response · Authors · 2026-05-14
> > >
> > > Hi Reviewer rqzd, thank you again for the positive review of our work! Please let us know whether we’ve addressed all of your concerns or if there are any additional issues we should address.

---

### Review · Reviewer_U4H9 · 2026-05-29

**Summary Of Contributions:**

The paper focuses on a scenario where a set of individuals must choose between different actions without knowing whether they are good or bad. A central planner reveals information for a subset of actions so that individuals become more likely to pick good ones. The authors introduce a model formalizing this as an interaction over a bipartite graph and analyze properties of the planner’s objective function. They then analyze a simple algorithm for the planner, providing theoretical guarantees and hardness results, depending on whether the planner can reveal only positive, only negative, or both types of actions. Finally, they introduce and analyze several variants of the problem and complement the analysis with semi-synthetic experimental results.

**Audience:**

No

**Audience Explanation:**

This paper studies a principal-agent problem, that is, a scenario where a social planner solves an optimization problem to induce beneficial behavior in individuals. Such problems are of interest to several members of the broader TMLR community, so I believe the paper has potential. However, as I mentioned above, a big limitation affecting whether the paper would be of interest to the TMLR audience is that it needs better motivation and clearer communication of its key ideas.

**Broader Impact Concerns:**

The work has no ethical implications requiring an impact statement.

**Claims And Evidence:**

No

**Claims Explanation:**

My overall take on the paper is that the technical results are mostly accurate, though I am not entirely convinced by them. The clarity and presentation could also be significantly improved. On the positive side, the paper addresses a problem that, to my knowledge, has not been studied in the literature. I have selectively reviewed some of the proofs and believe the technical content is correct. On the negative side, the modeling choices and the variants of the problem being studied lack motivation and real-world grounding, making it unclear whether the results have practical relevance. Additionally, the paper needs major restructuring for clarity. Currently, it feels like it introduces too many ideas without going deeply into any of them, leaving the reader with no clear takeaway. I elaborate more under "Requested Changes" but my main recommendation to the authors as they revise the paper would be to ask themselves: "What is the main result? What do we want the reader to remember from our paper?"

**Requested Changes:**

Major:

* My most important concern is that the motivation for the problem the authors study seems confined to the first paragraph of the introduction. The paper then makes several modeling choices and assumptions without providing further motivation or real-world grounding. More worryingly, the model ends up disconnected from the motivational example in the introduction. I am particularly concerned about the behavior of the agents. The authors make two assumptions that, from my perspective, directly clash with the tax-filing example they use as motivation. In the model, agents (i) have no information about whether a target is good or bad, and (ii) once they gain information about the targets, they always prefer to pick a positive target rather than a negative one. Using the paper’s example, this implies that a taxpayer (i) cannot distinguish that disclosing transfers is a positive action and misclassifying gifts to avoid taxes is a negative one, and (ii) once the planner tells them what is positive or negative, they blindly follow the suggestion, even if a "negative" action would allow them to avoid taxes. In other words, the model does not treat agents as strategic actors with their own incentives but rather as obedient followers of the planner. This makes the model easier to analyze, but I cannot see how it is relevant in practice. As a sidenote, one might expect a more fine-grained action space, rather than a binary distinction between "good" and "bad" actions where agents treat all actions of each category equally and randomize, but this is a secondary concern.
* Another aspect of the paper that I find unmotivated is its heavy focus on the greedy algorithm. The authors establish that, when the planner uses only positive examples, their formal expression of social welfare is submodular, making the greedy algorithm a natural choice due to its well-known approximation guarantees. However, when the planner also uses negative examples, the authors show that submodularity breaks. First, I would expect the authors to provide a numerical example demonstrating that including negative examples allows the planner to achieve higher social welfare than using only positive examples. Without this, I see no reason to devote so much space to analyzing this scenario, as it seems to only add complexity without any clear benefit. Second, since welfare is not submodular when negative examples are included, I would expect the authors to explore alternative algorithms that leverage other structural properties of the problem, rather than relying on assumptions that lead to theoretical guarantees for an algorithm that is tied to the submodular case. At the very least, it would be interesting to investigate whether the welfare satisfies some notion of weak submodularity, for which other algorithms have been proposed in the literature.
* I found Section 4 very rushed and unmotivated. The authors introduce several variants of the problem, but I do not see what this section adds to the paper. The mere fact that these variants *can* be analyzed does not mean they *must* be analyzed. For example, in the targeted interventions model, the authors assume that the planner sets $Q^{S_0}(x) = 1$, which seems to imply that the planner dictates to the agent to take a good action. What realistic scenario does this model? In the coverage radius model, edges have a distance-based interpretation, and the planner can expand the "visibility" or "radius" of the agents. What motivating applications do the authors have in mind? I cannot see any connection to the tax-filing scenario. Finally, I think the authors need to revisit the formalization of the "learning setting" (Section 4.3). They state that they sample agents $x$ from a distribution $D$, which leads to a (finite) "train graph," but this is somewhat vague. Strictly speaking, what the authors are doing is fixing the size $|\mathcal{X}|$, then drawing *edges* for each agent in $\mathcal{X}$ from a distribution. Their subsequent analysis may be fine as it varies the size of $|\mathcal{X}|$, but the formalization could be more precise.

Minor:

* I was confused by the use of the term "standard model" in the introduction, as it seems to imply that this version of the model is established in prior work. I would suggest using a different term, such as "basic model."
* I would suggest restructuring the order in which the results are presented. From my perspective, it seems unnatural to first introduce the submodularity of the welfare and the greedy approximation algorithm, and only afterward present the hardness results. The hardness results essentially motivate why the paper focuses on approximation algorithms in the first place. Without introducing hardness first, the reader might assume that the problem can be solved optimally, making it unclear why submodularity and this specific greedy algorithm are introduced.
* In the contributions, it would be helpful to separate the hardness results from the variants introduced in Section 4, as these are very different contributions.
* Although the discussion of related work seems satisfactory, it would be relevant to also discuss work focusing on scenarios where a planner provides global counterfactual explanations (e.g., see [1]) and uses them to incentivize beneficial agent behavior (e.g., see [2]). The latter, in particular, shares several conceptual and technical similarities with the current work.

  [1] Kavouras, Loukas, et al. "GLANCE: global actions in a nutshell for counterfactual explainability." *Proceedings of the AAAI Conference on Artificial Intelligence*. Vol. 40. No. 27. 2026.

  [2] Tsirtsis, Stratis, and Manuel Gomez Rodriguez. "Decisions, counterfactual explanations and strategic behavior." *Advances in Neural Information Processing Systems* 33 (2020): 16749-16760.

* I believe the proof of Proposition 3 needs to be more precise. The proposition itself addresses a scenario where the planner *includes* negative targets, yet the proof focuses on a scenario where the planner is *restricted* to negative targets. While the proof is correct (since the counterexample suffices to show that submodularity breaks when negative targets are included) it is not currently phrased as a counterexample. Some rephrasing is needed to align the proof with the proposition. Additionally, I found the proof sketch below it hard to follow and disconnected from the actual proof. I would suggest rephrasing it.
* The introduction of Section 3 (before 3.1) is quite long and, from my perspective, somewhat vague. I would suggest condensing it.
* In Section 3.3, it was not clear to me why the authors use big O notation. For example, I think writing "no group can achieve more than $OPT^K_a$ social welfare" would be sufficient and correct. Big O notation implies some sort of asymptotic behavior, which I do not believe is relevant in this context.
* As a general comment, I found it problematic that several aspects of the work are introduced in the main body of the paper but not sufficiently discussed, with all relevant results deferred to the Appendix. For example, Section 3.3 introduces a variant of the problem but presents no technical results. I would strongly encourage the authors to think whether they consider this section important. If so, the relevant results should be presented in full in the main body of the paper. If not, the entire subsection could be moved to the appendix. The same comment applies to Section 4. It is unclear what the authors consider important in that section. It reads more like a lengthy, high-level overview of setups the authors considered, with all results placed in the appendix.
* I believe the experiments could be expanded. For example, in Figure 2, the authors compare the greedy algorithm with a random algorithm, but there is no comparison with the optimal solution. This makes it unclear whether the greedy algorithm performs well overall, especially since its guarantees weaken when the planner reveals negative targets as well. In Figure 4, I expected to see an analysis of generalization, that is, how the method performs as the number of samples changes. The current figure is not particularly informative.

---

> ### Author Response · Authors · 2026-06-01
>
> Thank you, reviewer U4H9, for taking the time to review our work! Below, we address the points raised under “Requested Changes.” We have also uploaded a revised version of the manuscript with changes in response to the reviewer’s suggestions.
>
>
> > This paper studies a principal-agent problem … In other words, the model does not treat agents as strategic actors with their own incentives but rather as obedient followers of the planner. This makes the model easier to analyze, but I cannot see how it is relevant in practice. As a sidenote, one might expect a more fine-grained action space, rather than a binary distinction between "good" and "bad" … but this is a secondary concern.
>
> Our model captures settings in which agents rely on bad/negative and good/positive options, social models, or exemplars when making decisions. Additional motivating applications are discussed in the appendix. For example, comprehensively labeling all social media content sources as either legitimate/bad or illegitimate/good is very costly. As a result, content moderators may choose to label only a subset of content so as to reduce misinformation. A social media user will choose to rely on a “good” source of information if one they already regularly consume is revealed as positive, but if a “bad” one is revealed, they are left uncertain about which other sources to rely on and would then choose randomly among those they consume.
>
> Our setting is not necessarily a model of a principal-agent problem. However, in settings such as imitative strategic learning, where agents rely on social models to form accurate beliefs about the decision-making model so as to act optimally, our framework can be interpreted as a model of how agents choose which social models to imitate. An interesting direction for future research would be to extend our model to strategic learning environments. As noted in the conclusion, one possibility is to reinterpret emulation of positive and negative targets as improvement and gaming, respectively. Another promising extension would be to move beyond binary target labels and consider settings in which target quality is continuous and or ranked.
>
> > I would expect the authors to provide a numerical example demonstrating that including negative examples allows the planner to achieve higher social welfare than using only positive examples
>
> We do include examples where revealing negative targets instead of positive targets would result in higher social welfare. For example, in the bipartite graph shown in Figure 6, at a budget of $K=\kappa +1$, the optimal solution would be to choose negative  $t_{1}^{-'}$ to $t_{\kappa + 1}^{-'}$. Same thing for Figure 7.  We also ran empirical experiments on big and small graphs that demonstrated that including negative examples can result in higher social welfare than using only positive examples. E.g., consider the graph  below, with 4 agents ($x_0$ to $x_3$), with 4 positive targets and 3 negative targets.\
>  $x_0:$  $t_0^{+}$, $t_4^{-}$, $t_5^{-}$, $t_6^{-}$ \
>  $x_1:$  $t_1^{+}$, $t_4^{-}$, $t_5^{-}$, $t_6^{-}$ \
>  $x_2:$  $t_2^{+}$, $t_4^{-}$, $t_5^{-}$, $t_6^{-}$ \
> $x_3:$  $t_3^{+}$, $t_4^{-}$, $t_5^{-}$, $t_6^{-}$ \
> When K=3, the Greedy solution set is S=\{0, 1, 2\}, and the resulting social welfare is 3.25. On the other hand, the optimal solution set is \{4, 5, 6\} and the social welfare is 4.0. Additionally, in cases where agents are connected to both positive and negative targets, especially cases where negative targets are more common and shared than positive targets, the optimal solution set can be exclusively negative targets or a mix of positive and negative targets.  These exploratory empirical experiments were excluded for clarity.
>
> As such, we believe that it’s important to consider cases where a solution set might include negative targets or being exclusively negative targets, because there are so many scenarios where this could happen.

---

> > ### Author Response · Authors · 2026-06-01
> >
> > > Since welfare is not submodular when negative examples are included, I would expect the authors to explore alternative algorithms that leverage other structural properties of the problem, rather than relying on assumptions that lead to theoretical guarantees for an algorithm that is tied to the submodular case. At the very least, it would be interesting to investigate whether the welfare satisfies some notion of weak submodularity, for which other algorithms have been proposed in the literature.
> >
> > We do explore several alternative greedy strategies, including d-step lookahead and heuristic variants, both theoretically and empirically (see Appendices C.6 and G.2). In addition, our comparison of random selection against the standard greedy approach showed that greedy is better. Our results show that the inclusion of negative targets can significantly distort the social welfare objective, causing alternative greedy strategies to perform arbitrarily poorly in certain instances. \
> > To address this challenge, we introduce a proxy welfare function, a lower bound of the true welfare function, that remains submodular even when negative targets are included in the solution set. This allows us to obtain a constant-factor approximation to the true social welfare (see Section 3.2).
> >
> > > The authors introduce several variants of the problem, but I do not see what this section adds to the paper. ..
> >
> > These were, in principle, extensions of the main model.  In some settings, agents may simply have poor neighborhoods, making target disclosure alone insufficient. For example, an agent may be connected exclusively to negative targets, e.g., a taxpayer may only have access to non-compliant strategies, or, in the content moderation setting, a user may only be exposed to illegitimate or low-quality sources. In such cases, revealing target labels cannot increase the likelihood of selecting a positive target since none are available within the agent's choice set. In this case, targeted intervention is needed to explicitly connect high-risk agents to positive targets.
> >
> > A related challenge arises when positive targets exist but are not visible to agents (e.g., a user might be unaware of a good source within their interests). In these cases, interventions that increase the coverage or visibility of positive targets can improve outcomes by ensuring that agents are aware of adjacent positive alternatives.
> >
> > The extensions revealed several interesting findings. For example, we observed meaningful differences between interventions implemented before versus after the disclosure of target information, along with several other noteworthy patterns. The learning setting was analyzed both theoretically and empirically, whereas the interventional settings were studied primarily through empirical evaluation. While space constraints prevented a comprehensive discussion of all results in the main text, we included the most important insights there to ensure that the central takeaways remain readily accessible.  Lastly,  we note that these extensions also open several promising directions for future research, both theoretical and empirical.
> >
> > > I was confused by the use of the term "standard model" in the introduction ..
> >
> > We first consider the standard setup and the corresponding algorithm, which greedily reveals the targets that maximize the expected number of agents selecting positive targets. This provides a clear point of contrast with the interventional setting. In addition, in influence-style settings, the classic greedy algorithm is the canonical approach for identifying the most influential nodes, namely those that can most effectively shape agents' behavior or beliefs. So we saw it as a fitting phrase.

---

> ### Author Response · Authors · 2026-06-01
>
> > From my perspective, it seems unnatural to first introduce the submodularity of the welfare and the greedy approximation algorithm, and only afterward present the hardness results.
>
> We follow the reasoning outlined herein. The disclosure policy influences whether the objective function remains submodular, and this property directly determines the performance guarantees of the greedy algorithm. After characterizing how the disclosure policy affects submodularity and, in turn, the resulting approximation guarantees, we turn to NP-hardness to assess the potential for stronger algorithmic results and to demonstrate that the current guarantees are essentially tight.
>
> > In the contributions, it would be helpful to separate the hardness results from the variants introduced in Section 4, as these are very different contributions.
>
> Thanks, we revised and updated the manuscript!
>
> > More related works. Although the discussion of related work seems satisfactory, it would be relevant to also discuss work focusing on scenarios …
>
> Thanks, we revised and updated the manuscript!
>
> > I believe the proof of Proposition 3 needs to be more precise. The proposition itself addresses a scenario where the planner includes negative targets, yet the proof focuses on a scenario where the planner is restricted to negative targets.
>
> Thanks, we revised and updated the manuscript!
>
> > I believe the experiments could be expanded. For example, in Figure 2, the authors compare the greedy algorithm with a random algorithm, but there is no comparison with the optimal solution. …
>
> Figure 10 compares the optimal solution obtained by the brute force approach with the classic greedy and heuristic greedy approaches. Additional experimental results were omitted for clarity and to avoid visual clutter. See Appendix G for the complete set of experimental results.
>
> ***
> We have addressed the authors’ major and minor concerns and hope our responses clarify the misunderstandings and provide the necessary context for our work. We believe that our work offers value to  various TMLR communities (e.g., in discrete optimization, strategic learning, and social welfare), and that our claims are supported by accurate, convincing, clear, and rigorous theoretical and empirical evidence. *Please let us know if, after reviewing our responses and revised manuscript, you still have concerns or questions that would negatively affect your evaluation of our work. We would appreciate the opportunity to address them further.*

---

> > ### Comment · Reviewer_U4H9 · 2026-06-09
> >
> > I would like to thank the authors for their response. Although some of the minor points I raised in my review have been addressed, I still have concerns about this paper in its current form, and I don't think the revision addressed any of the major points I raised.
> >
> > **Motivation of the model and its technical assumptions:** I noticed that this is a point raised by the other reviewers as well. If I understand correctly, the revised version includes no additional discussion justifying the modeling choices, which still seem disconnected from the tax filing scenario used as motivation. To be clear, I am not asking for the authors to provide *more* examples/application areas, but to focus on *one* example, evaluate the realism of each of their modeling choices based on that, and incorporate this discussion in the technical description of their model. Without this, it is very hard for a reader to see the value of the paper. Running the risk of repeating myself, I think the assumptions about the agent specifically are too strong; the paper assumes a taxpayer who does not care about paying less taxes, has no idea about which tax filing practices are good or bad, and they obediently follow the suggestions of the tax authority. In my view, there is simply a large mismatch between the example and the proposed model, and the paper should either improve its modeling or find a different motivating example. That said, I do not think that labeling social media content is a good example either. I would argue that social media content is information that a user absorbs and combines with their existing information, not an example they emulate, and it is far too multi-dimensional to simply reduce it to a binary distinction between good/legitimate - bad/illegitimate. Also, there is a vast literature on fake news detection/content moderation, that the paper currently does not engage with.
> >
> > **Presentation and structure of the paper:** I appreciate the authors' example highlighting the value of negative targets. I would just encourage the authors to include that example early in the main body of the paper. In my view, this is the main motivation for introducing technical results involving negative targets and, hence, does not belong to an appendix (where Figures 6,7 currently are). Without such examples, it is not clear why an algorithm designer (who always has the flexibility to decide whether to reveal positive and/or negative targets) would care about any of the results involving negative targets.
> >
> > **Section 4:** Unfortunately section 4 remains a high-level overview of different setups lacking concrete technical content, and the authors' response does not sufficiently motivate the choice to focus on these particular extensions of the base model. The tax filing/content moderation examples are again rather weak and seem more like an afterthought to justify the modeling choices rather than the other way around. Unless I misunderstand something, the example of a taxpayer who only has access to non-compliant strategies means that a taxpayer can only file their taxes in a wrong way and the "intervention" by the authority forces them to file them correctly. I don't see how this scenario is even remotely realistic. In general, I would strongly encourage the authors to think more about the scenarios where they would envision their methods being applied and provide stronger arguments on why the extensions they propose deserve the readers' attention.

---

> > > ### Author Response · Authors · 2026-06-11
> > >
> > > Thank you, Reviewer U4H9!
> > >
> > > > Motivation of the model and its technical assumptions … Section 4
> > >
> > > We have edited the tax-filing example to motivate why the agents have incentives to follow revealed positive targets and to illustrate why agents are unaware of the labels of adjacent targets.
> > >
> > > For section 4, we chose to use the school counselor example to motivate the targeted intervention model, where the social planner directly connects agents prone to emulating negative targets (those with say 0 adjacent positive targets), to positive targets, and to motivate the coverage radius model, where agents are unaware of adjacent positive targets, and the social planner intervenes by increasing the visibility of positive targets.
> > >
> > > > Presentation and structure of the paper:
> > >
> > > We have added Figure 2 on page 6 to highlight different optimal set compositions.
> > >
> > > *Thank you again for your feedback. Please let us know whether we’ve addressed all of your concerns.*

---

### Review · Reviewer_82e6 · 2026-06-08

**Summary Of Contributions:**

The paper studies a budgeted label-revelation problem on a bipartite graph of agents and role models/options. The planner knows which targets are positive or negative and chooses a limited set of labels to reveal, with the goal of increasing the number of agents who emulate positive targets. The main technical contribution is showing that revealing only positive targets gives a monotone submodular objective, while allowing negative targets breaks submodularity. The authors then introduce a submodular proxy objective and prove constant-factor guarantees under a bounded-negative-neighbor assumption. They also provide hardness results, group-level guarantees, intervention extensions, a learning/sample-complexity result, and experiments.

The main strength, in my view, is that the paper identifies a clear technical issue: negative examples can be useful, but they create complementarities that make the objective harder to optimize. The paper also gives a fairly complete theoretical analysis around this issue.

The main limitation could be that the practical significance of the main assumption and proxy objective could be better justified. The bounded-negative-neighbor condition is mathematically useful and may be reasonable in some domains, but the paper should make clearer when it is expected to hold. Similarly, the proxy objective enables approximation guarantees, but its connection to the actual behavioral model is somewhat indirect. This does not invalidate the theoretical contribution, but it affects how compelling the results are as a model of real decision-support systems.

**Additional Comments:**

Disclaimer: I did not go through the appendix carefully. Thank you.

**Audience:**

Yes

**Audience Explanation:**

The paper should be of interest, especially regarding algorithmic decision support, strategic behavior, and fairness, and as well as some optimization problems. The main finding is a clean technical insight, and the proposed proxy-based guarantees provide a useful way to reason about this setting.

**Broader Impact Concerns:**

I do not see major broader impact concerns. The paper studies interventions intended to improve decision-making, but such methods could be misused if a planner selectively reveals labels to steer behavior in biased or paternalistic ways. The authors could briefly discuss transparency, accountability, and potential harms from incorrect or manipulative labeling.

**Claims And Evidence:**

Yes

**Claims Explanation:**

The paper proves the submodularity result for positive-only revelation, shows why negative revelation breaks it, and gives approximation guarantees via the proxy objective under stated assumptions. The empirical results also support the main qualitative claim that greedy performs well on the tested semi-synthetic graphs. In my view, the evidence is convincing for the main technical contributions, though the practical applicability of the modeling assumptions could be discussed more.

**Requested Changes:**

- Clarify the practical interpretation of the bounded-negative-neighbor assumption, including examples of domains where it is expected to hold.
- Explain more explicitly why the proxy welfare objective is a natural surrogate for the true welfare objective, beyond its usefulness for proving submodularity.
- Improve the presentation of the main results by clearly separating results for welfare (F) versus welfare gain (G), since these lead to different interpretations.
- Strengthen the discussion of the behavioral model, especially the assumptions that agents always follow revealed positive targets and choose uniformly among unrevealed alternatives.
- Better connect the experimental setup to the theoretical assumptions, for example by reporting how often the generated graphs satisfy or violate the (c)-bounded condition.
- Consider streamlining or better motivating the additional extensions, particularly the intervention and coverage-radius models, so their role in the overall contribution is clearer.

---

> ### Author Response · Authors · 2026-06-08
>
> Thank you, reviewer 82e6, for taking the time to review our work! Below, we address the points the reviewer raised under “Requested Changes.” We have also uploaded a revised version of the manuscript with changes in response to the reviewer’s suggestions.
>
>
> > Clarify the practical interpretation of the bounded-negative-neighbor assumption, including examples of domains where it is expected to hold.
>
> The idea is that each agent can only observe a bounded number of adjacent negative examples or social models within their local network. The c-boundness assumption is most likely to hold when the structure of the network and the agent’s neighborhood is inherently constrained. For example, when filing taxes, there is only a finite “group” of non-compliant strategies that a taxpayer is likely to encounter and potentially emulate through their social network.
>
> > Explain more explicitly why the proxy welfare objective is a natural surrogate for the true welfare objective, beyond its usefulness for proving submodularity.
>
> The proxy social welfare objective is just a lower bound on the true social welfare objective. The agents still behave the same.  Therefore, it serves as a natural surrogate for the true objective, beyond its usefulness for proving submodularity. In particular, the proxy does not change the underlying problem statement but approximates the welfare function by increasing the proxy by the amount that revealing the first negative neighbor helped.
>
> > Improve the presentation of the main results by clearly separating results for welfare (F) versus welfare gain (G), since these lead to different interpretations.
>
> In a setting where the social planner can reveal both positive and negative examples (Section 3.2.3 ), we show that a $((1+e)/c)$-factor approximation to the true social welfare gain can be achieved (Theorem 2). However, for any solution set, the proxy achieves $(1/c)$-factor approximation to the true optimal welfare (Remark 5).
> While section 3.2.3 analyzes both the social welfare and social welfare gain, section 3.3 focuses on the social welfare gain, as illustrated by equation 6 and the full proofs in Appendix C.4.
>
> Thank you for the suggestion! We have revised the manuscript accordingly to indicate when the analysis is specific to social welfare and to social welfare gain.

---

> ### Author Response · Authors · 2026-06-08
>
> > Strengthen the discussion of the behavioral model, especially the assumptions that agents always follow revealed positive targets and choose uniformly among unrevealed alternatives.
>
> Our model captures any setting in which agents rely on adjacent options/exemplars that could be positive/good or negative/bad, and can't distinguish them. Because agents rely on these social models to take actions that affect consequential outcomes and, ultimately, their livelihoods, they would choose to emulate a positive exemplar whenever one is identified. For example, in the tax-filing scenario discussed in the motivating examples, the penalties associated with incorrect tax filing create a strong incentive for taxpayers to follow the revealed positive strategy in their network. If multiple positive strategies are revealed, agents may uniformly at random select one among them.
>
> In practice, however, factors such as the cost of emulation may make some positive exemplars easier to emulate than others. In such cases, the choice among positive targets would no longer be uniformly random. Extending our setup to incorporate heterogeneous emulation probabilities across positive exemplars is a promising direction for future research.
>
> > Better connect the experimental setup to the theoretical assumptions, for example, by reporting how often the generated graphs satisfy or violate the (c)-bounded condition.
>
> We conduct extensive experiments to evaluate a range of model settings, and for the standard model, we compare the performance of the classical greedy algorithm against random selection, the optimal solution, and several alternative greedy strategies. Beyond these comparisons, there are several interesting empirical questions concerning the role of the parameter (c) and its interaction with other aspects of the model. For instance, the example setup pointed out by the reviewer, and other settings, e.g., how the gap between greedy and proxy greedy performance varies with c, how it affects group fairness, among others. We leave this empirical study for future work.
>
> > Consider streamlining or better motivating the additional extensions, particularly the intervention and coverage-radius models, so their role in the overall contribution is clearer.
>
> Thank you for the suggestion! We have revised both the contribution section in the Introduction and Section 5 to more clearly articulate the motivation for the proposed extensions to the standard model and to highlight the additional insights these extensions provide.
>
> *Thank you again for the positive review of our work! Please let us know whether we’ve addressed all of your concerns*